# Role of Locality and Weight Sharing in Image-Based Tasks: A Sample Complexity Separation between CNNs, LCNs, and FCNs

**Aakash Lahoti**[1], **Stefani Karp**[1,2], **Ezra Winston**[1], **Aarti Singh**[1] **& Yuanzhi Li**[1]
[1]Machine Learning Department, Carnegie Mellon University, [2]Google Research
`{alahoti, shkarp, ewinston, aarti, yuanzhil}@andrew.cmu.edu`

## Abstract

Vision tasks are characterized by the properties of locality and translation invariance. The superior performance of convolutional neural networks (CNNs) on these tasks is widely attributed to the inductive bias of locality and weight sharing baked into their architecture. Existing attempts to quantify the statistical benefits of these biases in CNNs over locally connected convolutional neural networks (LCNs) and fully connected neural networks (FCNs) fall into one of the following categories: either they disregard the optimizer and only provide uniform convergence upper bounds with no separating lower bounds, or they consider simplistic tasks that do not truly mirror the locality and translation invariance as found in real-world vision tasks. To address these deficiencies, we introduce the Dynamic Signal Distribution (DSD) classification task that models an image as consisting of $k$ patches, each of dimension $d$, and the label is determined by a $d$-sparse signal vector that can freely appear in any one of the $k$ patches. On this task, for any orthogonally equivariant algorithm like gradient descent, we prove that CNNs require $\tilde{O}(k+d)$ samples, whereas LCNs require $\Omega(kd)$ samples, establishing the statistical advantages of weight sharing in translation invariant tasks. Furthermore, LCNs need $\tilde{O}(k(k+d))$ samples, compared to $\Omega(k^2d)$ samples for FCNs, showcasing the benefits of locality in local tasks. Additionally, we develop information theoretic tools for analyzing randomized algorithms, which may be of interest for statistical research.

## 1 Introduction

Convolutional Neural Networks (CNNs) exhibit state-of-the-art performance across computer vision tasks, including Image Classification, Object Detection, and Out of Distribution Detection (Liu et al. (2022); Fang et al. (2022); Wang et al. (2022)). This efficacy is commonly attributed to the biases of locality and weight sharing encoded into CNNs' short convolutions. The rationale is that these biases align with the properties of vision tasks, where local and mobile signals determine the output (Gens & Domingos (2014); Marcus (2018)). In contrast, Locally Connected Neural Networks (LCNs) encode only locality, while Fully Connected Neural Networks (FCNs) encode neither locality nor weight sharing, thus resulting in a larger sample complexity compared to CNNs.

Previous works have attempted to quantify the statistical benefit of these architectural biases in CNNs. For example, Vardi et al. (2022), Du et al. (2018) and Long & Sedghi (2020) derived Empirical Risk Minimization (ERM) bounds for CNNs which are tighter than that for FCNs. However, they do not provide separating lower bounds for FCNs on the same task, and cannot rule out the possibility that FCNs can adaptively yield better bounds when the input satisfies locality and translation invariance. In fact, as noted in Li et al. (2021), without taking the training algorithm into consideration, standard lower bound techniques cannot be used to show a separation between the three models. This is because an algorithm can simulate CNNs and LCNs within FCNs. Thus, if the algorithm is unconstrained, the minimax lower bound for FCNs cannot be greater than any upper bound for CNNs or LCNs.

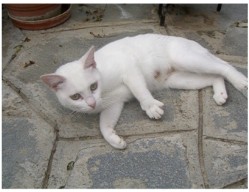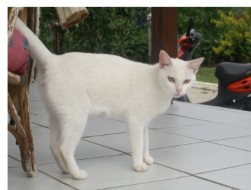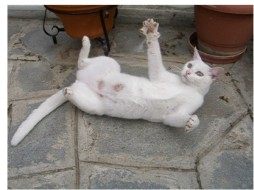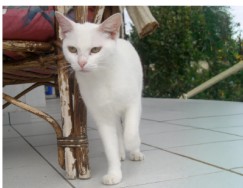

Figure 1: From the Cats Dataset Zhang et al. (2008). The cat, which is the class-determining signal, varies in position across images, showing the translation property amidst background noise.

Recently, Li et al. (2021) established a sample complexity separation between CNNs and FCNs that were trained on the restricted class of equivariant algorithms like gradient descent [1]. Wang & Wu (2023) further extended this line of work to show a separation between FCNs, LCNs and CNNs. However, the data models employed in these works are not truly reflective of the locality and translation invariance of vision tasks. Typically in such tasks, the output is determined by some local pattern, also known as "signal". For example, a cat within images labeled "cat". Often, this signal is embedded within uninformative background, also known as "noise", and can freely translate within the image, i.e. it can appear in any patch within the image, without changing the label (as illustrated in figure 1). In contrast, in both Li et al. (2021) and Wang & Wu (2023), the data model considered is as follows: the input $\mathbf{x} \sim \mathcal{N}(0, I_{4d})$, and the label is given by $f(\boldsymbol{x})$, and $g(\boldsymbol{x})$ respectively,

$$f(\mathbf{x}) = \sum_{i=1}^{2d} x_i^2 - \sum_{i=2d+1}^{4d} x_i^2, \qquad g(\mathbf{x}) = (\sum_{i=1}^{d} x_{2i}^2 - x_{2i+1}^2)(\sum_{i=d+1}^{2d} x_{2i}^2 - x_{2i+1}^2). \tag{1}$$

Both of these data models fail to capture the aforementioned desiderata of a model for a vision task. Additionally, they lack the requisite structure to demonstrate how sample complexity varies with the "degree" of locality and translation invariance within the input, or establish conditions on the input under which the differences between CNNs, LCNs, and FCNs are more pronounced.

Furthermore, it is worth noting that the driving force for their separation results is the interaction between two halves of the input. Specifically, their lower bound selects "hard instances" from the class of functions $\mathcal{H} = \{\boldsymbol{x}_{1:d}^\top \mathbf{U} \boldsymbol{x}_{d+1:2d}\}$, where $\mathbf{U}$ is a $d \times d$ orthonormal matrix, learning which results in a lower bound of $\Omega(d^2)$. While the interaction between the patches is an interesting phenomenon, it is not the primary characteristic of locality and translation invariance found in images.

We introduce the Dynamic Signal Distribution (DSD) task, which is inspired by the setting in Karp et al. (2021), as our data model for vision tasks. The input $\boldsymbol{x} \in \mathbb{R}^{kd}$ is comprised of $k$ consecutive patches, each of dimension $d$. From amongst these $k$ patches, one of them is randomly filled with a noisy signed signal. The remaining patches are filled with isotropic Gaussian noise of variance $\sigma^2$. The binary label is set as the sign of the signal, so that all images with the same signal in any one of the patches have the same label. By encapsulating concepts of signal, noise, locality, and translation invariance, the DSD task offers a higher fidelity to the complexities found in real-world vision tasks.

On this task, we establish a sample complexity separation of $\Omega(\sigma^2 k^2 d)$ vs $\tilde{O}(\sigma^2 k(k+d))$ samples between FCNs and LCNs, as well as a separation of $\Omega(\sigma^2 kd)$ vs $\tilde{O}(\sigma^2(k+d))$ samples between LCNs and CNNs. Our analysis indicates that due to no architectural biases, FCNs incur a multiplicative cost factor of $k$ for each of the two reasons: identifying the location of the $k$ patches, and learning the signal vector for each patch. The factor of $d$ arises due to learning the signal which is $d$ dimensional. For LCNs, we can eliminate the $k$ cost for identification of the patches since the location of all the patches is baked into the architecture. Finally for CNNs both these costs are removed as the architecture not only localizes all the patches, it also allows the signal to be jointly learnt across all patches via weight sharing. It is noteworthy that both the LCN and the CNN upper bound feature a $k+d$ factor instead of the expected factor of $d$. This is an artifact of the gradient descent analysis, and is suggestive of being a potential cost for the algorithmic efficiency of gradient descent.

---

[1]Formally, equivariance is defined on the pair of the network architecture and the training algorithm. For brevity, we may refer to an algorithm as equivariant, when the underlying network(s) are clear from the context.

Our approach diverges from Li et al. (2021); Wang & Wu (2023), because in our task, the marginal over the input is not a $\mathbf{0}$-mean Gaussian, but a mixture of $k$ Gaussians, which is not an orthogonally invariant distribution. As a consequence, for deriving lower bounds, we cannot apply the Benedek-Itai bound from Benedek & Itai (1991) as done in Li et al. (2021), nor can we directly use Fano's Theorem as done in Wang & Wu (2023) owing to the absence of the semi-metricness of $l_2$ loss under an invariant distribution, and analyzing the expected risk under a mixture of Gaussians is analytically difficult. Instead, we utilize a novel technique that leverages the randomness of the training algorithm to break the original minimax lower-bound problem into $k$ simpler problems using a simulation-style argument. In case of FCNs, we prove sample complexity lower bounds for the $k$ the simpler problems using a novel boosting technique to derive a reduction to the Gaussian mean estimation problem on the unit sphere. To prove sample complexity lower bounds for the simpler problems in the case of LCNs, we prove a variant of Fano's Theorem that can be used for randomized algorithms. Distinctively, our variant does not require the semi-metric property to hold on the entire space of output functions, as is needed in the "Fano's Theorem for Random Estimators" developed in Wang & Wu (2023).

Our sample complexity upper bounds depend on the analysis of an equivariant gradient descent style algorithm on LCNs and CNNs. This is unlike the separation proved in Wang & Wu (2023), where they use covering number-based arguments for ERM analysis. The advantage of doing a gradient descent analysis over an ERM analysis is two fold: First, it demonstrates a sample complexity separation for computationally-efficient (poly time) equivariant algorithms. This distinction is crucial because while a separation may exist for computationally inefficient algorithms, the separation might disappear under constraints of computational efficiency. Second, for a valid separation, it is important to ensure that both the upper and lower bounds are derived for equivariant algorithms since non-equivariant algorithms could potentially be more sample-efficient than their equivariant counterparts. Furthermore, our approach differs significantly from Karp et al. (2021), which analyzes population gradients by assuming enough ($\text{poly}(k, d)$) samples at each iteration to yield a representational gap between CNNs and CNTKs (Convolutional Neural Tangent Kernels). Since we are interested in sample complexity separation, we adopt a more direct analysis of empirical gradients.

## 2 OTHER RELATED WORKS

We already discussed some of the most relevant works, including Li et al. (2021); Wang & Wu (2023); Karp et al. (2021); Vardi et al. (2022) in the introduction. Here, we will highlight a couple of additional works.

Another work, Malach & Shalev-Shwartz (2020), proved a computational separation between FCNs and CNNs on a "$k$-pattern" classification task. In the task, the inputs are from the hypercube $\{-1, 1\}^n$, and the label is based on a set of $k$ consecutive coordinates. They employ random-feature analysis to establish that CNNs, with $2^k$ hidden nodes, can learn this task in $O(2^k n)$ samples. In contrast, we only require $O(k)$ nodes and samples. Furthermore, they do not provide lower bounds for FCNs, and instead argue that the gradient is too small for a finite precision machine. Additionally, since their task does not encode translation invariance, they cannot prove a separation between LCNs and CNNs.

## 3 NOTATION

**Vector and Matrix Notation:** We use bold lowercase letters, such as $\boldsymbol{x}, \boldsymbol{y}$, to represent vectors, and bold uppercase letters, such as $\mathbf{U}, \mathbf{V}$, to represent matrices. Let $[n]$ denote the set $\{1, \ldots, n\}$. We denote the standard basis of $\mathbb{R}^n$ by $\mathcal{B}_n$ and the individual basis vectors by $\boldsymbol{e}_i$. We define the function $\text{idx}_n \colon \mathcal{B}_n \to [n]$, $\text{idx}_n(\boldsymbol{e}_l) = l$, for all $l \in [n]$. For any $\boldsymbol{x}$, indexed from 1, we use $\boldsymbol{x}[i:j] \in \mathbb{R}^{j-i+1}$ to represent a slice from its $i$-th to its $j$-th entry. For a set $\{\boldsymbol{x}_i\}_{i=1}^n$, we employ $(\boldsymbol{x}_1, \ldots, \boldsymbol{x}_n)$ to denote the sequential length-wise concatenation of the vectors, and $(\boldsymbol{x}_1; \ldots; \boldsymbol{x}_n)$ to denote the sequential row-wise stacking of the vector transposes into a matrix. Conversely, for any $\boldsymbol{x}$ constructed via ( ; ) or ( , ) notation, we denote its $i^{\text{th}}$ component vector by $\boldsymbol{x}^{(i)}$. We use $\mathbf{U} = \text{Block}(\{\mathbf{U}_1, \ldots \mathbf{U}_n\})$ to be the matrix having diagonal blocks of $\mathbf{U}_i$'s in-sequence, with other entries set to zero. Conversely, for any $\mathbf{U}$ constructed via $\text{Block}(\cdot)$, we denote its $i^{\text{th}}$ component matrix by $\mathbf{U}^{(i)}$. The Euclidean norm for vectors and the spectral norm for matrices are both denoted by $\|\cdot\|$.

**Group Notation:** Let $\mathcal{U}_1$, and $\mathcal{U}_2$ be any two subgroups of $\text{GL}(n, \mathbb{R})$. Then, we define then binary operation $\star$ such that $\mathcal{U}_1 \star \mathcal{U}_2 = \{\mathbf{U}_1 \mathbf{U}_2 .. \mathbf{U}_n \mid \mathbf{U}_i \in \mathcal{U}_1 \cup \mathcal{U}_2, n \in \mathbb{N}\}$. It is easy to see that $\mathcal{U}_1 \star \mathcal{U}_2$

is also a subgroup of $\mathrm{GL}(n, \mathbb{R})$. We denote $\mathcal{O}(n)$ to be the group of orthonormal matrices on $\mathbb{R}^{n \times n}$ and $\mathcal{O}_p(n)$ be the group of permutation matrices on $\mathbb{R}^{n \times n}$.

**Task Notation:** Let $\mathcal{X} \subseteq \mathbb{R}^p$, and $\mathcal{Y} \subseteq \mathbb{R}$ denote the input and output space of a $p$ dimensional problem. Let $P$ be any distribution over $(\mathcal{X}, \mathcal{Y})$ and $\tau \colon \mathcal{X} \to \mathcal{X}$ be any function, then we define the distribution $\tau \circ P$ over $(\mathcal{X}, \mathcal{Y})$ by sampling $(\boldsymbol{x}, y) \sim P$ and returning $(\tau(\boldsymbol{x}), y)$. Let $\mathcal{P}$ be a set of distributions over $(\mathcal{X}, \mathcal{Y})$, then we define the set $\tau \circ \mathcal{P} := \{\tau \circ P \mid P \in \mathcal{P}\}$. Alternatively, let $T$ be a set of functions from $\mathcal{X} \to \mathcal{X}$, then we define $T \circ P := \{\tau_i \circ P \mid \tau_i \in T\}$.

**Model Notation:** We denote a parametric model by $\mathcal{M}$ and its parameter set by $\mathcal{W}$. The model along with its parameter is a function from $\mathcal{X}$ to $\mathbb{R}$. Specifically, $\forall \boldsymbol{w} \in \mathcal{W}, \mathcal{M}[\boldsymbol{w}] \colon \mathcal{X} \to \mathbb{R}$.

We will use $O(\cdot)$, $\Omega(\cdot)$, and $\Theta(\cdot)$ as the Big-O, Big-Omega, and Big-Theta notation respectively. The notation $\tilde{O}(\cdot)$, $\tilde{\Omega}(\cdot)$, and $\tilde{\Theta}(\cdot)$ hides logarithmic factors.

# 4 OUR SETTING

We introduce the Dynamic Signal Distribution, an image-like task which is inspired from Karp et al. (2021). We also specify the FCN, LCN, and CNN architectures that we consider for our analysis.

## 4.1 DYNAMIC SIGNAL DISTRIBUTION (DSD)

In many vision-based tasks, the output often relies on a local "signal" in the image, a property referred to as locality. Often, this signal is enveloped in random noise, and satisfies translation invariance, that is its movement within the image does not alter the output. The DSD task is designed to capture the both locality and translation invariance properties into an analyzable task.

We define the input space as $\mathcal{X} = \mathbb{R}^{kd}$ and the output space as $\mathcal{Y} = \mathbb{R}$. Any input vector $\boldsymbol{x} \in \mathcal{X}$ is structured as $(\boldsymbol{x}^{(1)}, .., \boldsymbol{x}^{(k)})$, with each $\boldsymbol{x}^{(i)}$ being a vector in $\mathbb{R}^d$, and representing the $i^{\text{th}}$ patch of $\boldsymbol{x}$. Thus, each input consists of $k$ consecutive patches of dimension $d$. To model the local signal, we employ an unknown unit vector $\boldsymbol{w}^\star \in \mathbb{R}^d$, with $\|\boldsymbol{w}^\star\| = 1$. To include translation invariance in the task, this signal $\boldsymbol{w}^\star$ can reside within any one of the $k$ patch locations, described above. Specifically, for each $i$ in $[k]$, we define a $d$-sparse mean vector $\boldsymbol{\mu}_i \in \mathbb{R}^{kd}$, such that $\boldsymbol{\mu}_i[(i-1)d+1 : id] = \boldsymbol{w}^\star$ and all its other entries are zero. The noise is chosen to be isotropic Gaussian, with variance $\sigma^2 \in \mathbb{R}_+$.

Formally, DSD is a distribution over $(\mathcal{X}, \mathcal{Y})$ with the generative story: sample the index $i \sim \mathrm{Unif}([k])$, and the label $y \sim \mathrm{Unif}(\{-1, 1\})$. Then, sample the input data as $\boldsymbol{x}|(y, i) \sim \mathcal{N}(y\boldsymbol{\mu}_i, \sigma^2 \mathbf{I}_{kd})$. Observe that the probability density function (pdf) of DSD is,

$$p(\boldsymbol{x}, y) = \frac{1}{2k(\sqrt{2\pi\sigma^2})^{kd}} \sum_{i=1}^{k} \exp\left(-\frac{\|\boldsymbol{x} - y\boldsymbol{\mu}_i\|^2}{2\sigma^2}\right). \tag{2}$$

We also define the Static Signal Distribution ($\mathrm{SSD}_t$), which is the conditional distribution of DSD when the index parameter is fixed at $i = t$. Specifically, the label is chosen as $y \sim \mathrm{Unif}(\{-1, 1\})$, and then input data is sampled as $\boldsymbol{x}|y \sim \mathcal{N}(y\boldsymbol{\mu}_t, \sigma^2 \mathbf{I}_{kd})$. We will use this distribution in proving the lower bounds in theorem 6.1, 7.1, by reducing the problem of learning DSD to learning each $\mathrm{SSD}_t$.

## 4.2 NEURAL NETWORK ARCHITECTURES

We now introduce the model architectures that we consider for our analysis. We adopt the Local Signal Adaptivity (LSA) activation function, first introduced in Karp et al. (2021), for all models,

$$\phi_b(x) : \mathbb{R} \to \mathbb{R} := \mathrm{ReLU}(x - b) - \mathrm{ReLU}(-x - b), \tag{3}$$

where $b \in \mathbb{R}_+$ is the trainable bias parameter. The rationale for choosing $\phi_b(x)$ is its capability to 'filter out' noise below the magnitude of $b$, while letting signals of magnitude larger than $b$ to propagate through the network. This denoising helps the network learn the signal with fewer samples. We also note that the LSA activation function, also known as the "soft-thresholding function" , is extensively used in high-dimensional sparse recovery problems (Section 18.2 Hastie et al. (2009)). Since our task involves recovering the sparse mean vector, it futher justifies its use for the DSD task.

**FCN**: We consider a one-hidden-layer network with $k$ hidden nodes. Each hidden node $i$, is associated with a parameter vector $\boldsymbol{w}_i \in \mathbb{R}^{kd}$, such that $\|\boldsymbol{w}_i\| \leq 1$. The complete model parameter vector is given by $\boldsymbol{v} = [\boldsymbol{w}_1, .., \boldsymbol{w}_k, b] \in \mathcal{W}$, where $\mathcal{W} = \mathbb{R}^{k^2 d} \times \mathbb{R}_+$. The function form for FCN is,

$$\mathcal{M}_F[\boldsymbol{v}](\boldsymbol{x})\colon \mathcal{X} \to \mathbb{R} \coloneqq \sum_{i=1}^k \phi_b(\boldsymbol{w}_i^T \boldsymbol{x}). \tag{4}$$

**LCN:** Similar to FCN, we consider a one-hidden-layer network featuring $k$ hidden nodes. The $i$-th node is associated with the parameter vector $\boldsymbol{w}_i \in \mathbb{R}^d, \|\boldsymbol{w}_i\| \leq 1$. The complete model parameter vector is given by $\boldsymbol{v} = [\boldsymbol{w}_1, .., \boldsymbol{w}_k, b]$, and $\mathcal{W} = \mathbb{R}^{kd} \times \mathbb{R}_+$. The function form for LCN is,

$$\mathcal{M}_L[\boldsymbol{v}](\boldsymbol{x})\colon \mathcal{W} \to \mathbb{R} \coloneqq \sum_{i=1}^k \phi_b(\boldsymbol{w}_i^T \boldsymbol{x}^{(i)}) \tag{5}$$

**CNNs:** We consider a one hidden-layer CNN has $k$ hidden nodes. The parameter $\boldsymbol{w} \in \mathbb{R}^d, \|\boldsymbol{w}\| \leq 1$ is the shared across all nodes. The composite vector $\boldsymbol{v} = [\boldsymbol{w}, b]$ is our complete model parameter vector, and $\mathcal{W} = \mathbb{R}^d \times \mathbb{R}_+$. The function form for CNN is,

$$\mathcal{M}_C[\boldsymbol{v}](\boldsymbol{x})\colon \mathcal{W} \to \mathbb{R} \coloneqq \sum_{i=1}^k \phi_b(\boldsymbol{w}^T \boldsymbol{x}^{(i)})$$

The subscripts $F$, $L$, and $C$ denotes that the model corresponds to a FCN, LCN, and CNN respectively.

## 5 MATHEMATICAL BACKGROUND

### 5.1 TECHNICAL DEFINITIONS

**Definition 1** (Loss Function). *We define the loss function for our task as* $err\colon (\mathcal{Y}, \mathcal{Y}) \to \mathbb{R}_+$,

$$err(\bar{y}, y) = (\bar{y} - y)^2. \tag{6}$$

**Definition 2** (Risk). *Let* $\mathcal{F} = \mathcal{Y}^{\mathcal{X}}$, *and let* $\mathcal{P}$ *be the set of all distributions over* $(\mathcal{X}, \mathcal{Y})$. *Then, we define the risk* $R\colon (\mathcal{F}, \mathcal{P}) \to \mathbb{R}_+$ *of a function* $f \in \mathcal{F}$ *with respect to the distribution* $P \in \mathcal{P}$ *as,*

$$R(f, P) = \mathbb{E}_{(\boldsymbol{x}, y) \sim P} \left[ err(f(\boldsymbol{x}), y) \right]. \tag{7}$$

**Definition 3** (Algorithm). *Let* $\mathcal{F} \subseteq \mathcal{Y}^{\mathcal{X}}$, $\Xi$ *be the sample space that encapsulates all algorithmic randomness, and* $P_\Xi$ *be some fixed distribution over* $\Xi$. *Then, a randomized algorithm denoted by* $\theta\colon ((\mathcal{X}, \mathcal{Y})^n, \Xi) \to \mathcal{F}$, *is a function defined from the product space of input data and randomness to the space of possible functions. The randomness is realized by sampling from the distribution* $P_\Xi$.

*We may omit* $(\mathcal{X}, \mathcal{Y})$ *and* $\Xi$ *from the notation when they are clear from the context and use the random variable notation* $\theta_n$ *instead, where* $n$ *denotes the number of samples.*

**Definition 4** (Iterative (Randomized) Algorithm). *Consider a parametric model* $\mathcal{M}$, *and its parameter set* $\mathcal{W}$, *such that for any* $\boldsymbol{w} \in \mathcal{W}$, $\mathcal{M}[\boldsymbol{w}]$ *is a maps from the input space* $\mathcal{X}$ *to the output space* $\mathcal{Y}$. *Let* $\mathcal{F} = \{\mathcal{M}[\boldsymbol{w}] \mid \boldsymbol{w} \in \mathcal{W}\}$. *Let the model parameters be initialized via a distribution* $W$ *over* $\mathcal{W}$, $\boldsymbol{w}^0 \sim W$. *Let* $T$ *be the number of iterations and* $F^t : (\mathcal{W}, S^n) \to \mathcal{W}$ *be the update functions for each iteration* $t$. *Then the function* $\theta\colon ((\mathcal{X}, \mathcal{Y})^n, \mathcal{W}; \mathcal{M}[\mathcal{W}], \{F^t\}_t) \to \mathcal{F}$ [2] *is an iterative algorithm if it adheres to the procedure 1.*

**Definition 5** (Sample Complexity). *Let* $P$ *be a distribution over* $(\mathcal{X}, \mathcal{Y})$ *and* $\theta_n$ *be a randomized algorithm as defined in 3. Let* $S^n \sim P^n$ *be* $n$ *i.i.d. data points sampled from* $P$. *For any* $\delta \in [0, 1]$, *we define the* $\delta$-*sample complexity of* $\theta_n$ *as,*

$$n_\delta(\theta_n, P) = \min_{n \in \mathbb{N}} \left\{ n \in \mathbb{N} \mid \mathbb{E}\left[ R(\theta_n, P) \right] \leq \delta \right\}, \tag{8}$$

*where the expectation is over the input data* $S^n$, *and the algorithmic randomization.*

*We may omit the distribution* $P$ *from the sample complexity notation* $n_\delta(\theta_n, P)$ *and use the shorthand* $n_\delta(\theta_n)$ *instead, when* $P$ *is clear from the context.*

---

[2]In the proofs, we will employ a generalization of this definition, wherein the parameter $\mathcal{W}$ will be replaced by a general space $\Xi$ and an associated fixed, data-independent distribution $P_\Xi$. $\Xi$ includes parameters as well as other random quantities. All definitions presented henceforth also hold for this generalization.

---

**Algorithm 1** Iterative Algorithm

---

**Require:** Update functions $\{F^t\}_T$, Set of $n$ i.i.d. data samples $S^n$, Parameter initialization $\boldsymbol{w}^0$,
    $t \leftarrow 1$
    **while** $t \leq T$ **do**
        $\boldsymbol{w}^t \leftarrow F^t(\boldsymbol{w}^{t-1}, S^n)$
        $t \leftarrow t + 1$
    **end while**
    **return** $\boldsymbol{w}^T$

---

## 5.2 Equivariant Algorithms

We introduce the concept of equivariant algorithms, originally presented in Li et al. (2021). To keep it concise, we provide a simplified version which is sufficient for our purposes.

To motivate the definition of equivariant algorithms, we review the following thought experiment. Consider a neural network parameterized as $f(\mathbf{A}\boldsymbol{x}, \boldsymbol{b})$, where $\mathbf{A} \in \mathbb{R}^{q \times p}$ is the parameter of the first linear layer, while $\boldsymbol{b} \in \mathbb{R}^q$ encapsulates the remaining parameters. We initialize the parameters as $(\mathbf{A}^0, \boldsymbol{b}^0)$ and use gradient descent, with learning rate $\eta$, to train the network on the dataset $\{\boldsymbol{x}_i, y_i\}_n$. In parallel, we train another network initialized as $(\mathbf{A}^0\mathbf{U}^T, \boldsymbol{b}^0)$, with the dataset $\{\mathbf{U}\boldsymbol{x}_i, y_i\}_n$. Here, $\mathbf{U} \in \mathcal{O}(p)$ such that $\mathbf{A}^0\mathbf{U}^T$ and $\mathbf{A}^0$ are identically distributed.

Observe that at the first iteration, the output of the first hidden layer for both networks is the same, $\mathbf{A}^0\mathbf{U}^T\mathbf{U}\boldsymbol{x} = \mathbf{A}^0\boldsymbol{x}$. This implies that the gradients with respect to the pre-activations of the first layer are also equal. Consequently, the gradients with respect to the matrix parameters satisfy the relation, $\frac{d}{d\mathbf{A}^0}\text{loss}(\mathbf{A}^0)\mathbf{U}^T = \frac{d}{d\mathbf{A}^0\mathbf{U}^T}\text{loss}(\mathbf{A}^0\mathbf{U}^T) := \Delta\mathbf{U}^T$. Thus, after the first iteration, the parameter sets for the two neural networks are,

$$(\mathbf{A}^1, \boldsymbol{b}^1) = (\mathbf{A}^0 - \eta\Delta, \boldsymbol{b}^1), \qquad (\mathbf{A}^1\mathbf{U}^T, \boldsymbol{b}^1) = (\mathbf{A}^0\mathbf{U}^T - \eta\Delta\mathbf{U}^T, \boldsymbol{b}^1), \qquad (9)$$

respectively. By induction, this property is preserved across all iterations $t$, resulting in the parameters for the two neural networks being $(\mathbf{A}^t, \boldsymbol{b}^t)$ and $(\mathbf{A}^t\mathbf{U}^T, \boldsymbol{b}^t)$, respectively.

The key idea is that the risk of a network parameterized as $(\mathbf{A}^t, \boldsymbol{b}^t)$ on any data $\{\boldsymbol{x}, y\}$ is the same as its counterpart with parameters $(\mathbf{A}^t\mathbf{U}^T, \boldsymbol{b}^t)$ on the transformed data $\{\mathbf{U}\boldsymbol{x}, y\}$. Now since $\mathbf{A}^0\mathbf{U}^T$ and $\mathbf{A}^0$ have the same distribution, we can infer that the expected risk of this network trained with gradient descent is invariant to the transformation $\mathbf{U}$ of the input distribution. In other words, the network learns the original distribution and the transformed distribution equally well. Formally,

**Definition 6** ($\mathcal{U}$-equivariant algorithm). *Under the notation established in definition 4, let the input space $\mathcal{X} \subseteq \mathbb{R}^p$, the output space $\mathcal{Y} \subseteq \mathbb{R}$, and the parameter set $\mathcal{W} \subseteq \mathbb{R}^m$. Let $\mathcal{U} \subseteq \mathcal{O}(p)$, then an iterative algorithm $\bar{\theta}_n$ is is $\mathcal{U}$-equivariant if there exists a set $\mathcal{V} \subseteq \mathcal{O}(m)$, such that,*

1. *For all $\mathbf{U} \in \mathcal{U}$, there exists $\mathbf{V} \in \mathcal{V}$ such that for all $\boldsymbol{x} \in \mathcal{X}$, and $\boldsymbol{w} \in \mathcal{W}$,*
   *$\mathcal{M}[\boldsymbol{w}](\boldsymbol{x}) = \mathcal{M}[\mathbf{V}\boldsymbol{w}](\mathbf{U}\boldsymbol{x})$.*

2. *For all $\mathbf{U} \in \mathcal{U}$, the same $\mathbf{V} \in \mathcal{V}$ as defined in (1) satisfies $\forall\{\boldsymbol{x}_i, y_i\}_n \in (\mathcal{X}, \mathcal{Y})^n$, $\forall t \in [T]$, and $\boldsymbol{w} \in \mathcal{W}$, $\mathbf{V}F^t(\boldsymbol{w}, \{\boldsymbol{x}_i, y_i\}_n) = F^t(\mathbf{V}\boldsymbol{w}, \{\mathbf{U}\boldsymbol{x}_i, y_i\}_n)$*

3. *If $\boldsymbol{w} \sim W$, then for all $\mathbf{V} \in \mathcal{V}$, $\mathbf{V}\boldsymbol{w} \stackrel{d}{=} \boldsymbol{w}$.*

And, equivariant algorithms satisfy the following property,

**Lemma 5.1.** *(Section 4.1 Li et al. (2021)) If $\bar{\theta}_n$ is a $\mathcal{U}$-equivariant algorithm, then $\forall \boldsymbol{x} \in \mathcal{X}, \mathbf{U} \in \mathcal{U}$,*

$$\bar{\theta}(\{\boldsymbol{x}_i, y_i\}_n)(\boldsymbol{x}) \stackrel{d}{=} \bar{\theta}(\{\mathbf{U}\boldsymbol{x}_i, y_i\}_n)(\mathbf{U}\boldsymbol{x}), \qquad (10)$$

*where the randomness is over initialization.*

This property formalizes the conclusion drawn in the thought experiment. That is the performance of an equivariant algorithm when trained on $n$ i.i.d. samples from $P_1$ and tested on $P_2$ would be the same, in distribution, had it been trained on $n$ i.i.d. samples from $\mathbf{U} \circ P_1$ and tested on $\mathbf{U} \circ P_2$.

## 5.3 Minimax Framework

We present the minimax framework by closely following the notation established in Duchi (2021). Let $\mathcal{P}$ denote a set of distributions over $(\mathcal{X}, \mathcal{Y})$ and $\mathcal{F} \subseteq \mathcal{Y}^{\mathcal{X}}$ represent a set of functions from $\mathcal{X}$ to $\mathcal{Y}$. Let $\theta^{\star} \colon \mathcal{P} \to \mathcal{F}$ be some unknown target mapping, and let $\Theta = \{\theta \mid \theta \colon ((\mathcal{X}, \mathcal{Y})^n, \Xi) \to \mathcal{F}\}$ be a set of algorithms with a common distribution $P_{\Xi}$ over the sample space $\Xi$ that encapsulates randomness. Let $\rho \colon \mathcal{F} \times \mathcal{F} \to \mathbb{R}_+$ be some symmetric positive function.

**Definition 7** (Minimax Risk). *Under the notation from above, we define the minimax risk of learning the set of tasks $\mathcal{P}$ using the set of algorithms $\Theta$ as,*

$$\mathfrak{M}_n(\Theta, \mathcal{P}) := \inf_{\theta_n \in \Theta} \sup_{P \in \mathcal{P}} \mathbb{E}\left[\rho(\theta_n, \theta^{\star}(P))\right]. \tag{11}$$

For brevity, we may omit $\mathcal{P}$ from the notation, when it is clear form context. The primary change in our adaptation of the minimax framework is that we allow for randomized algorithms, whose randomness is independent of the input data distribution. In contrast, the original framework is only applicable to deterministic algorithms, typically referred to as estimators.

We now present our Fano's Theorem for Randomized Algorithms to lower bound the minimax risk 11. In this variant, we relax the constraint that $\rho$ is a semi-metric on the space $\mathcal{F}$. Specifically, given a set of "hard problem" instances $\mathcal{P}_{\mathcal{V}}$, and their associated target functions $\mathcal{F}_{\mathcal{V}}$, we only require that if a function $f \in \mathcal{F}$ is "close enough", in $\rho$, to any $g \in \mathcal{F}_{\mathcal{V}}$, then it is "far enough", in $\rho$, to all $\mathcal{F}_{\mathcal{V}} \setminus \{g\}$. This relaxation helps us prove lower bounds when the stronger semi-metric property does not hold.

**Theorem 5.1** (Fano's Theorem for Randomized Algorithms). *Under the notation established above, let $\mathcal{V}$ be an index set of finite cardinality of some chosen subset of $\mathcal{P}$. Then, we define $\mathcal{P}_{\mathcal{V}} := \{P_v \mid \forall v \in \mathcal{V}\}$, and $\mathcal{F}_{\mathcal{V}} := \{\theta^{\star}(P_v) \mid \forall v \in \mathcal{V}\}$. For some fixed parameter $\delta > 0$, let $\rho$ satisfy the condition that, for all $f_u \neq f_v \in \mathcal{F}_{\mathcal{V}}$ and $f \in \mathcal{F}$, if $\rho(f, f_u) < \delta$, then $\rho(f, f_v) > \delta$. And, for all $P_u, P_v \in \mathcal{P}_{\mathcal{V}}$, $u \neq v$, let the KL divergence satisfy $KL(P_u \parallel P_v) \leq D$ for some $D > 0$. Then,*

$$\mathfrak{M}_n(\Theta) \geq \delta \left(1 - \frac{nD + \ln(2)}{\ln(|\mathcal{V}|)}\right).$$

The proof of this theorem is presented in appendix B.

**Remark 1.** *We only need to define $\rho$ on the subset $\mathcal{F} \times \mathcal{F}_{\mathcal{V}}$ of its domain $\mathcal{F} \times \mathcal{F}$ and $\theta^{\star}$ on the subset $\mathcal{P}_{\mathcal{V}}$ of its domain $\mathcal{P}$ to apply the above theorem.*

## 6 FCNs vs LCNs Separation Results

We now present the separation result between FCNs and LCNs, along with an outline of the proof. Specifically, we establish that FCNs, when trained with any equivariant algorithm, require $\Omega(\sigma^2 k^2 d)$ samples to learn DSD upto some constant risk $\delta$. Conversely, there exists an equivariant algorithm that can train LCNs with $\tilde{O}(\sigma^2 k(k + d))$ samples, to achieve a risk less than $\delta$.

**Theorem 6.1** (Sketched). *Consider the group $\mathcal{U} = \mathcal{O}(kd)$, then any $\mathcal{U}$-equivariant algorithm that is used to train FCNs, requires $\Omega(\sigma^2 k^2 d)$ samples to achieve some constant risk $\delta$.*

*Proof.* We justify the choice of $\mathcal{U} = \mathcal{O}(kd)$, in light of the intuition for equivariance presented in section 5.2. Note that the parameter of the first layer of FCNs, $\mathbf{A} \in \mathbb{R}^{k \times kd}$, is given by $(\boldsymbol{w}_1; \ldots; \boldsymbol{w}_k)$. We establish equivariance if, for every transformation $\mathbf{U} \in \mathcal{U}$, $\mathbf{A}\mathbf{U}^T$ corresponds to a valid FCN, and if there exists an initialization such that $\mathbf{A}\mathbf{U}^T$ and $\mathbf{A}$ are identically distributed. Indeed, $\mathbf{A}\mathbf{U}^T = (\mathbf{U}\boldsymbol{w}_1; \ldots; \mathbf{U}\boldsymbol{w}_k)$, corresponds to a FCN with the parameter vectors $\mathbf{U}\boldsymbol{w}_1, \ldots, \mathbf{U}\boldsymbol{w}_k$. And, if each $\boldsymbol{w}_i$ is initialized as $\boldsymbol{w}_i \sim \mathcal{N}(\mathbf{0}, \mathbf{I}_{kd})$, then $\mathbf{A}\mathbf{U}^T$ and $\mathbf{A}$ share the same distribution.

Our proof proceeds in two steps. First, we establish that learning $\mathbf{U} \circ \mathrm{DSD}$ with $m$ samples requires learning $k$ "nearly independent" subtasks, $\{\mathbf{U} \circ \mathrm{SSD}_t\}_k$, with $m/k$ samples each. The underlying rationale of this result is that learning $\mathbf{U} \circ \mathrm{DSD}$ entails recovering each mean vector $\{\mathbf{U}\boldsymbol{\mu}_t\}_k$. Note that these mean vectors are pair-wise orthogonal, $(\mathbf{U}\boldsymbol{\mu}_i)^T(\mathbf{U}\boldsymbol{\mu}_j) = \boldsymbol{\mu}_i^T \boldsymbol{\mu}_j = 0$. Therefore, even with the knowledge of $\{\mathbf{U}\boldsymbol{\mu}_t\}_{t \neq i}$, the only information we have about $\mathbf{U}\boldsymbol{\mu}_i$ is the $kd - k + 1 \simeq kd$ dimensional subspace in which it lies. Thus, to learn DSD, we have to recover all the means vectors, $\{\mathbf{U}\boldsymbol{\mu}_t\}_k$, "nearly independently" from each other.

In the second step, we reduce the problem of learning $\text{SSD}_t$, into a problem of Gaussian mean estimation. For this, we show that if there exists an algorithm that learns $\text{SSD}_t$, then we can extract a weakly aligned mean estimate of $\mathbf{U}\boldsymbol{\mu}_t$ from the FCN returned by the algorithm. We propose a scheme that reliably boosts this estimate, to generate a strongly aligned mean estimate. This is necessary because standard information theoretic tools do not work with weakly aligned mean estimates. We then bound the sample complexity for any algorithm that is able to return a strongly aligned Gaussian mean estimate using our Fano's Theorem for Randomized Estimators 5.1 as $m/k = \Omega(\sigma^2 kd)$. This implies that $m = \Omega(\sigma^2 k^2 d)$, proving the result.

The formal statement of the theorem and its proof can be found in appendix C $\hfill\square$

**Theorem 6.2.** *(Sketched) Consider the groups* $\mathcal{U}_1 := \{Block\left(\{\mathbf{U}_1,\ldots,\mathbf{U}_k\}\right) \mid \mathbf{U}_i \in \mathcal{O}(d)\}$, *and* $\mathcal{U}_2 := \{\mathbf{U} \in \mathcal{O}_p(kd) \mid idx_{kd}(\mathbf{U}e_{(i-1)d+1}) + j - 1 = idx_{kd}(\mathbf{U}e_{(i-1)d+j}), \, \forall i \in [k], j \in [d]\}$. *Let* $\mathcal{U} = \mathcal{U}_1 \star \mathcal{U}_2$. *Then there exists a* $\mathcal{U}$-*equivariant algorithm that trains LCNs with* $\tilde{O}(\sigma^2 k(k+d))$ *samples, to achieve a risk less than* $\delta$.

*Proof.* To justify our choice of $\mathcal{U}$ for LCNs, we establish equivariance under $\mathcal{U}_1$ and $\mathcal{U}_2$ separately. The equivariance under $\mathcal{U}$ simply follows from an induction on the number of finite combinations of elements of $\mathcal{U}_1 \cup \mathcal{U}_2$.

Equivariance under $\mathcal{U}_1$

Consider an input $\boldsymbol{x} \in \mathbb{R}^{kd}$, then any transformation $\mathbf{U} \in \mathcal{U}_1$ operates on $\boldsymbol{x}$ on a per-patch basis. On each patch, $\mathbf{U}$ induces, a possibly distinct, orthogonal transformation. We now show equivariance under the notation from section 5.2. The linear layer parameter $\mathbf{A} \in \mathbb{R}^{k \times kd}$ is given by $Block(\boldsymbol{w}_1,\ldots,\boldsymbol{w}_k)$. Observe that, $\mathbf{A}\mathbf{U}^T = Block(\mathbf{U}^{(1)}\boldsymbol{w}_1,\ldots,\mathbf{U}^{(k)}\boldsymbol{w}_k)$, which corresponds to a LCN with parameter vectors $\{\mathbf{U}^{(1)}\boldsymbol{w}_1,\ldots,\mathbf{U}^{(k)}\boldsymbol{w}_k\}$. And if each $\boldsymbol{w}_i$ is sampled as $\boldsymbol{w}_i \sim \mathcal{N}(\mathbf{0},\mathbf{I}_d)$, then $\mathbf{A}\mathbf{U}^T$ and $\mathbf{A}$ share the same distribution.

Equivariance under $\mathcal{U}_2$

A transformation $\mathbf{U} \in \mathcal{U}_2$ permutes the $k$ input patches amongst each other, while retaining each internal structure of each patch. Let $\pi \colon [k] \to [k]$, be the permutation function corresponding to $\mathbf{U}$. Then observe that $\mathbf{A}\mathbf{U}^T = Block(\boldsymbol{w}_{\pi(1)},\ldots,\boldsymbol{w}_{\pi(k)})$, which corresponds to a LCN with parameter vectors $\{\boldsymbol{w}_{\pi(1)},\ldots,\boldsymbol{w}_{\pi(k)}\}$. And, if $\boldsymbol{w}_i \sim \mathcal{N}(\mathbf{0},\mathbf{I}_d)$, then $\mathbf{A}\mathbf{U}^T$ and $\mathbf{A}$ share the same distribution.

Our training uses gradient descent, accompanied by a projection on the unit ball after every descent step. We included this projection to simplify the analysis, though we note that it can be removed without changing the core proof structure. The training proceeds in two steps. We show that after the first update, each parameter vector achieve an alignment of $(\boldsymbol{w}^\star)^T \boldsymbol{w}_i = \Omega(\sqrt{(k+d)/kd})$. In the second step, we use this alignment to reliably filter out the noise patches, while retaining the signal patches. This denoising enables us to prove a stronger $\Omega(1)$ alignment, which implies that the model has successfully recovered signal vector. Consequently, the model has a small risk $\leq \delta$.

Detailed theorem statements and proofs are available in appendix C.2. $\hfill\square$

## 7 LCNs vs CNNs Separation Results

We now present the separation results between LCNs and CNNs, along-with their sketched proofs. Specifically, we show that a LCN trained with any equivariant algorithm, requires $\Omega(\sigma^2 kd)$ samples to learn DSD upto a risk of $\delta$. On the other hand, there exists an equivariant algorithm that can train CNNs with $\tilde{O}(\sigma^2(k+d))$ samples to achieve a risk that is less than $\delta$.

**Theorem 7.1.** *(Sketched) Consider the groups* $\mathcal{U}_1 := \{Block\left(\{\mathbf{U}_1,\ldots,\mathbf{U}_k\}\right) \mid \mathbf{U}_i \in \mathcal{O}(d)\}$, *and* $\mathcal{U}_2 := \{\mathbf{U} \in \mathcal{O}_p(kd) \mid idx_{kd}(\mathbf{U}e_{(i-1)d+1}) + j - 1 = idx_{kd}(\mathbf{U}e_{(i-1)d+j}), \, \forall i \in [k], j \in [d]\}$. *Let* $\mathcal{U} = \mathcal{U}_1 \star \mathcal{U}_2$. *Then any* $\mathcal{U}$-*equivariant algorithm that is used to train LCNs requires* $\Omega(\sigma^2 k^2 d)$ *samples to achieve a risk of* $\delta$.

*Proof.* We have already justified the choice of $\mathcal{U}$ for LCNs in the sketched proof of theorem 6.2.

We follow in the footsteps of the proof of theorem 6.1. First, we establish that learning $\mathbf{U} \circ \mathrm{DSD}$ with $m$ samples requires learning $k$ independent subtasks, $\{\mathbf{U} \circ \mathrm{SSD}_t\}_k$, with $m/k$ samples each. The distinction from the proof of theorem 6.1, is that the subtasks are fully independent. This is because, the group $\mathcal{U}$ does not permit interaction amongst the $k$ patches. In other words, the vectors $\{\mathbf{U}^{(1)}\boldsymbol{\mu}_1, \ldots, \mathbf{U}^{(k)}\boldsymbol{\mu}_k\}$ are all $d$-sparse, and occupy non-overlapping subspaces. Therefore, even if we have the knowledge of $\{\mathbf{U}^{(t)}\boldsymbol{\mu}_t\}_{t\neq i}$, we would still have no information about $\mathbf{U}^{(i)}\boldsymbol{\mu}_i$. Thus, we have to recover all the $d$-sparse mean vectors independently of each other.

In the second step, we prove an information-theoretic lower bound to learn $\mathbf{U} \circ \mathrm{SSD}_t$ with $m/k$ samples. We find a function that lower bounds the risk incurred by a LCN on $\mathrm{SSD}_t$. This function satisfies the weakened conditions of theorem 5.1. Finally, we use theorem 5.1 together with the Gilbert-Varshamov lemma A.1.1, to show that $m/k = \Omega(\sigma^2 d)$. And implies that $m = \Omega(\sigma^2 kd)$.

The complete statement of the theorem with its proof can be found in appendix D.1 $\qquad \square$

**Theorem 7.2.** *(Sketched) Define* $\mathcal{U}_1 \coloneqq \{\mathrm{Block}\left(\{\mathbf{U}_1, \ldots, \mathbf{U}_k\}\right) \mid \mathbf{U}_i = \mathbf{U}_j, \mathbf{U}_i \in \mathcal{O}(d)\}$, *and* $\mathcal{U}_2 \coloneqq \{\mathbf{U} \in \mathcal{O}_p(kd) \mid idx_{kd}(\mathbf{U}\boldsymbol{e}_{(i-1)d+1}) + j - 1 = idx_{kd}(\mathbf{U}\boldsymbol{e}_{(i-1)d+j}), \forall i \in [k], j \in [d]\}$. *Let* $\mathcal{U} = \mathcal{U}_1 \star \mathcal{U}_2$. *Then there exists a* $\mathcal{U}$-*equivariant algorithm that trains CNNs, as defined in 6, with* $\tilde{O}(\sigma^2(k + d))$ *samples, to achieve a risk of less than* $\delta$.

*Proof.* To justify our choice of $\mathcal{U}$ for CNNs, we establish equivariance under $\mathcal{U}_1$ and $\mathcal{U}_2$ separately. The equivariance under $\mathcal{U}$ follows from induction on the number of finite combinations in $\mathcal{U}_1 \star \mathcal{U}_2$.

Equivariance under $\mathcal{U}_1$

Consider an input $\boldsymbol{x} \in \mathbb{R}^{kd}$, then any transformation $\mathbf{U} \in \mathcal{U}_1$ induces the same orthogonal transformation on every patch of $\boldsymbol{x}$. Moreover, it does not allow for any inter-patch interaction. To prove equivariance, observe that the parameter $\mathbf{A} \in \mathbb{R}^{k \times kd}$ is given by $\mathrm{Block}(\boldsymbol{w}, .. k \text{ times } .., \boldsymbol{w})$. Note that, $\mathbf{A}\mathbf{U}^T = \mathrm{Block}(\mathbf{U}^{(1)}\boldsymbol{w}, \ldots, \mathbf{U}^{(k)}\boldsymbol{w}) = \mathrm{Block}(\mathbf{U}^{(1)}\boldsymbol{w}, \ldots, \mathbf{U}^{(1)}\boldsymbol{w})$, which corresponds to a CNN with parameter vectors $\{\mathbf{U}^{(1)}\boldsymbol{w}, \ldots, \mathbf{U}^{(1)}\boldsymbol{w}\}$. And if the parameter vector, $\boldsymbol{w}$, is initialized as $\boldsymbol{w} \sim \mathcal{N}(\mathbf{0}, \mathbf{I}_d)$, then $\mathbf{A}\mathbf{U}^T$ and $\mathbf{A}$ share the same distribution.

Equivariance under $\mathcal{U}_2$

A transformation $\mathbf{U} \in \mathcal{U}_2$ permutes the $k$ input patches, while retaining the internal structure of each patch. Equivariance follows directly from the argument in the proof of theorem 6.2.

Our approach exactly follows the proof theorem 6.2. We train the CNN using gradient descent, followed by a projection on the unit ball. The training algorithm has two iterations. We show an alignment of $\Omega(\sqrt{(k + d)/kd})$ after the first update, and a stronger alignment of $\Omega(1)$ via denoising after the second update. This implies that the model has successfully recovered signal vector, and consequently it has a small risk $\leq \delta$. $\qquad \square$

Detailed theorem statements and proofs are available in appendix D.2.

## 8 Conclusion And Future Work

In this paper, we established a sample complexity separation between FCNs, LCNs, and CNNs that are trained using equivariant algorithms on the Dynamic Signal Distribution (DSD) task. Unlike previous works, this task encodes the concepts of signal, noise, locality, and translation invariance, thus incorporating the salient characteristics of vision-based tasks. We quantify the benefits of locality and weight sharing on the DSD task. Specifically, we show that FCNs incur an extra multiplicative cost of $k^2$ because they lacks both architectural biases, LCNs incur a $k$ cost because of the absence of weight sharing, whereas CNNs avoid these costs because it exhibits both locality and weight sharing.

In future work, we plan to incorporate second-order characteristics of images into the data model. For instance, allowing multiple signals to appear across different patches simultaneously would mirror real-world scenarios where multiple objects occur. Additionally, an interesting direction would be to analyze the role of depth in a CNN in capturing dependency between different patches.

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

## A  RESTATED GILBERT VARSHAMOV BOUND

**Theorem A.1** (Massart et al. (2007), Lemma 4.7). *Let $\{0,1\}^N$ be equipped with Hamming distance $\delta$ and given $1 \leq D < N$ define $\{0,1\}^N_D = \{x \in \{0,1\}^N : \delta(0,x) = D\}$. For every $\alpha \in (0,1)$ and $\beta \in (0,1)$ such that $D \leq \alpha\beta N$, there exists some subset $\Theta$ of $\{0,1\}^N$ with the following properties,*

$$\delta(\theta,\theta') > 2(1-\alpha)D \ \ \forall(\theta,\theta') \in \Theta^2, \ \theta \neq \theta', \tag{12}$$

$$\ln|\Theta| \geq \rho D \ln\left(\frac{N}{D}\right), \tag{13}$$

*where,*

$$\rho = \frac{\alpha}{-\ln(\alpha\beta)}(-\ln(\beta) + \beta - 1). \tag{14}$$

**Corollary A.1.1.** *Let $\mathcal{S}$ be the set of all unit vectors of $\mathbb{R}^N$, that is, $\mathcal{S} := \{u \mid u \in \mathbb{R}^N, \|u\| = 1\}$. Then for any constant $c \geq \frac{2}{N}$, there exists some subset $\tilde{\mathcal{S}} \subseteq \mathcal{S}$ of size $\ln(|\tilde{\mathcal{S}}|) \geq N$ such that, for all $u, v \in \tilde{\mathcal{S}}$, $u^T v < c$.*

*Proof.* We set the value of $D = \frac{N}{2}$. Consider the set $S_1 := \{\frac{u}{\sqrt{D}} \mid u \in \{0,1\}^N, \|u\|_0 = D\}$. It is easy to see that $S_1 \subseteq S$. Observe that for any $u, v \in S_1$, $\delta(u,v) > N(1 - \frac{c}{2})$ if and only if $u^T v < c$. Now, we set $\alpha = \frac{c}{2}$, $\beta = \frac{1}{c}$, and apply Gilbert-Varshamov Bound,

$$\ln(|\tilde{S}|) \geq (c\ln(c) - c + 1)N. \tag{15}$$

$\square$

# B PROOF OF THEOREM 5.1

The following helper lemma derives the KL divergence between two transformations of $\text{SSD}_t$, namely $\mathbf{U} \circ \text{SSD}_t$ and $\mathbf{V} \circ \text{SSD}_t$.

**Lemma B.1.** *For any* $\mathbf{U}, \mathbf{V} \in \mathcal{O}(kd)$, *then the KL Divergence between* $\mathbf{U} \circ SSD_t$ *and* $\mathbf{V} \circ SSD_t$ *is,*

$$KL(\mathbf{U} \circ SSD_t \parallel \mathbf{V} \circ SSD_t) = \frac{1-\cos(\alpha)}{\sigma^2} \tag{16}$$

*where* $\cos(\alpha) = (\mathbf{U}\boldsymbol{\mu}_t)^T \mathbf{V}\boldsymbol{\mu}_t$

*Proof.*

$$\text{KL}(\mathbf{U} \circ \text{SSD}_t \parallel \mathbf{V} \circ \text{SSD}_t) = \mathbb{E}_{(\boldsymbol{x},y)\sim\mathbf{U}\circ\text{SSD}_t} \ln\left(\frac{\exp\left(-\frac{\|\boldsymbol{x}-y\mathbf{U}\boldsymbol{\mu}_t\|^2}{2\sigma^2}\right)}{\exp\left(-\frac{\|\boldsymbol{x}-y\mathbf{V}\boldsymbol{\mu}_t\|^2}{2\sigma^2}\right)}\right) \tag{17}$$

$$= \mathbb{E}_y \mathbb{E}_{\boldsymbol{x}=y\mathbf{U}\boldsymbol{\mu}_t+\sigma\boldsymbol{\epsilon}} \ln\left(\frac{\exp\left(-\frac{\|\boldsymbol{x}-y\mathbf{U}\boldsymbol{\mu}_t\|^2}{2\sigma^2}\right)}{\exp\left(-\frac{\|\boldsymbol{x}-y\mathbf{V}\boldsymbol{\mu}_t\|^2}{2\sigma^2}\right)}\right) \tag{18}$$

$$= \mathbb{E}_y \mathbb{E}_{\boldsymbol{x}=\mathbf{U}\boldsymbol{\mu}_t+\sigma\boldsymbol{\epsilon}} \ln\left(\frac{\exp\left(-\frac{\|\boldsymbol{x}-\mathbf{U}\boldsymbol{\mu}_t\|^2}{2\sigma^2}\right)}{\exp\left(-\frac{\|\boldsymbol{x}-\mathbf{V}\boldsymbol{\mu}_t\|^2}{2\sigma^2}\right)}\right) \tag{19}$$

$$= \mathbb{E}_{\boldsymbol{x}=\mathbf{U}\boldsymbol{\mu}_t+\sigma\boldsymbol{\epsilon}} \ln\left(\frac{\exp(\boldsymbol{x}^T\mathbf{U}\boldsymbol{\mu}_t/\sigma^2)}{\exp(\boldsymbol{x}^T\mathbf{V}\boldsymbol{\mu}_t/\sigma^2)}\right) \tag{20}$$

$$= \mathbb{E}_{\boldsymbol{x}=\mathbf{U}\boldsymbol{\mu}_t+\sigma\boldsymbol{\epsilon}} \left[\boldsymbol{x}^T\mathbf{U}\boldsymbol{\mu}_t/\sigma^2 - \boldsymbol{x}^T\mathbf{V}\boldsymbol{\mu}_t/\sigma^2\right] \tag{21}$$

$$= \boldsymbol{\mu}_t^T\mathbf{U}^T\mathbf{U}\boldsymbol{\mu}_t/\sigma^2 - \boldsymbol{\mu}_t^T\mathbf{U}^T\mathbf{V}\boldsymbol{\mu}_t/\sigma^2 \tag{22}$$

$$= \frac{1-\cos(\alpha)}{\sigma^2}, \tag{23}$$

which proves the required result. $\qquad\square$

**Theorem 5.1** (Fano's Theorem for Randomized Algorithms). Under the notation established above, let $\mathcal{V}$ be an index set of finite cardinality of some chosen subset of $\mathcal{P}$. Then, we define $\mathcal{P}_\mathcal{V} := \{P_v \mid \forall v \in \mathcal{V}\}$, and $\mathcal{F}_\mathcal{V} := \{\theta^\star(P_v) \mid \forall v \in \mathcal{V}\}$. For some fixed parameter $\delta > 0$, let $\rho$ satisfy the condition that, for all $f_u \neq f_v \in \mathcal{F}_\mathcal{V}$ and $f \in \mathcal{F}$, if $\rho(f, f_u) < \delta$, then $\rho(f, f_v) > \delta$. And, for all $P_u, P_v \in \mathcal{P}_\mathcal{V}$, $u \neq v$, let the KL divergence satisfy $\text{KL}(P_u \parallel P_v) \leq D$ for some $D > 0$. Then,

$$\mathfrak{M}_n(\Theta) \geq \delta\left(1 - \frac{nD+\ln(2)}{\ln(|\mathcal{V}|)}\right).$$

*Proof.* From the definition of minimax risk,

$$\mathfrak{M}_n(\Theta) = \inf_{\theta\in\Theta} \sup_{P\in\mathcal{P}} \mathbb{E}_{S^n\sim P^n,\xi\sim P(\Xi)} \left[\rho(\theta(S^n,\xi),\theta^\star(P))\right],$$

$$\geq \inf_{\theta\in\Theta} \sup_{P\in\mathcal{P}_\mathcal{V}} \mathbb{E}_{S^n\sim P^n,\xi\sim P(\Xi)} \left[\rho(\theta(S^n,\xi),\theta^\star(P))\right],$$

$$= \inf_{\theta\in\Theta} \sup_{Q\in\mathcal{Q}_\mathcal{V}^n} \mathbb{E}_{(S^n,\xi)\sim Q} \left[\rho(\theta(S^n,\xi),\theta^\star(Q))\right],$$

where $\mathcal{Q}_\mathcal{V}^n := \{Q(S^n,\xi) := P^n(S^n) * P_\Xi(\xi) \mid P \in \mathcal{P}_\mathcal{V}\}$, and we overload the target mapping notation and set $\theta^\star(Q) = \theta^\star(P)$, where $P$ is the distribution corresponding to $Q$. First, observe that for all $Q_u, Q_v \in \mathcal{Q}_\mathcal{V}^n$, $u \neq v$, the KL divergence between the two distributions is given by,

$$\text{KL}(Q_u \parallel Q_v) = \mathbb{E}_{(S^n,\xi)\sim Q_u} \frac{Q_u(S^n,\xi)}{Q_v(S^n,\xi)} = \mathbb{E}_{S^n\sim P_u^n,\xi\sim P_\Xi} \frac{P_u^n(S^n)P_\Xi(\xi)}{P_v^n(S^n)P_\Xi(\xi)},$$

$$= \mathbb{E}_{S^n\sim P_u^n} \frac{P_u^n(S^n)}{P_v^n(S^n)} = \text{KL}(P_u^n \parallel P_v^n) = n\text{KL}(P_u \parallel P_v) = nD.$$

We follow in the footsteps of the proof of Fano's Theorem [Prop 7.3 Duchi (2021)]. For any $Q \in \mathcal{Q}_{\mathcal{V}}^n$,

$$\mathop{\mathbb{E}}_{Q}[\rho(\theta_n, \theta^\star(Q))] \geq \mathop{\mathbb{E}}_{Q}[\delta \, \mathbf{1}\{\rho(\theta_n, \theta^\star(Q)) \geq \delta\}] \geq \delta \, \mathbb{P}[\rho(\theta_n, \theta^\star(Q)) \geq \delta].$$

We define the testing function, $\Psi \colon \mathcal{F} \to \mathcal{V}$ as,

$$\Psi(f) := \operatorname*{arg\,min}_{v \in \mathcal{V}}\{\rho(f, \theta^\star(Q_v))\},$$

where ties can be broken arbitrarily and the analysis would still hold. Let $v$ be the uniform random variable over $\mathcal{V}$. Recall the assumption on $\rho$ that, if $\rho(f, Q_u) < \delta$, then $\rho(f, Q_v) > \delta$,

$$\sup_{Q \in \mathbb{Q}_{\mathcal{V}}^n} \mathbb{P}[\rho(\theta_n, \theta^\star(Q)) \geq \delta] \geq \frac{1}{|\mathcal{V}|} \sum_{v \in \mathcal{V}} \mathbb{P}[\rho(\theta_n, \theta^\star(Q_v)) \geq \delta \mid \boldsymbol{v} = v],$$

$$\geq \frac{1}{|\mathcal{V}|} \sum_{v \in \mathcal{V}} \mathbb{P}[\Psi(\theta_n) \neq v \mid \boldsymbol{v} = v],$$

$$\geq \inf_{\Psi} \mathbb{P}[\Psi(\theta_n) \neq \boldsymbol{v}].$$

From the above, 24, and Prop 7.10 and Eq 7.4.5 from Duchi (2021), we have the result,

$$\mathfrak{M}_n(\Theta) \geq \delta \left(1 - \frac{nD + \ln(2)}{\ln(|\mathcal{V}|)}\right).$$

$\square$

## C    FCNs vs LCNs Separation Results

### C.1    FCN SSD Lower Bound

**Lemma C.1.** *Let $S^n \sim (SSD_1)^n$ be $n$ i.i.d. data samples drawn from $SSD_1$. Define the equivariance group $\mathcal{U} \coloneqq \mathcal{O}(kd)$. Define the subset $\tilde{\mathcal{U}} \subseteq \mathcal{U}$ such that, for all $\mathbf{U} \in \tilde{\mathcal{U}}$, $t \in \{2, \ldots, k\}$, $\mathbf{U}\boldsymbol{\mu}_t = e_{kd-k+t}$. Let $\mathcal{P} \coloneqq \{\mathbf{U} \circ P | \mathbf{U} \in \tilde{\mathcal{U}}\}$ be the set of problem distributions. Let $\Xi$ be the sample space that encapsulates algorithmic randomness, and $P_\Xi$ be a distribution over $\Xi$. Let $\Theta \coloneqq \{\theta \colon (\mathcal{X}^m, \Xi) \to \mathbb{S}^{kd-1}\}$ be the set of $\mathcal{U}$-equivariant randomized algorithms that estimate the mean of the input distribution using $n$ i.i.d. samples. If,*

$$\inf_{\theta_n \in \Theta} \sup_{\mathbf{U} \in \tilde{\mathcal{U}}} \mathbb{E}_{S^n \sim (\mathbf{U} \circ P)^n, \xi \sim P_\Xi} \|\theta_n - \mathbf{U}\boldsymbol{\mu}_1\| \geq 0.25, \tag{24}$$

*then $n = \Omega(\sigma^2 kd)$, for large enough $k, d$.*

*Proof.* We will prove this statement using Fano's Theorem for Randomized Algorithms 5.1. Observe that since $\|\cdot\|$ is already a metric, the relaxed semi-metric property holds for all $\delta$.

We begin by constructing a $2\delta$, $\delta = 0.25$, packing of the set of means $\mathcal{S} = \{\mathbf{U}\boldsymbol{\mu}_1 \mid \mathbf{U} \in \tilde{\mathcal{U}}\}$. Observe from the construction of $\tilde{\mathcal{U}}$ that,

$$\begin{aligned}
\mathcal{S} \supset \mathcal{S}_1 &\coloneqq \{\boldsymbol{u} \mid \boldsymbol{u} \in \mathbb{R}^{kd}, \|\boldsymbol{u}\| = 1, \boldsymbol{u} \in \text{Span}(\{e_1, \cdots, e_{kd-k}\})\} \\
&\cong \mathcal{S}_2 \coloneqq \{\boldsymbol{u} \mid \mathbb{R}^{kd-k}, \|\boldsymbol{u}\| = 1\}, \\
&\supset \mathcal{S}_3 \coloneqq \{\boldsymbol{u} \mid \mathbb{R}^{kd-k}, \|\boldsymbol{u}\| = 1, \boldsymbol{u}[j] = \tfrac{1}{\sqrt{kd-k}}, j \in [\tfrac{kd-k}{2}]\}, \\
&\cong \mathcal{S}_4 \coloneqq \{\boldsymbol{u} \mid \mathbb{R}^{\frac{kd-k}{2}}, \|\boldsymbol{u}\| = \tfrac{1}{2}\},
\end{aligned}$$

where $\cong$ denotes the fact that $\mathcal{S}_1, \mathcal{S}_2$, and $\mathcal{S}_3$, $\mathcal{S}_4$ are isometric sets under the Euclidean norm. Therefore, it is enough to find a $2\delta$ packing of $\mathcal{S}_4$ to find a $2\delta$ packing of $\mathcal{S}_1$. Now, define the set $\mathcal{S}_5 \coloneqq \{\boldsymbol{u} \mid \mathbb{R}^{\frac{kd-k}{2}}, \|\boldsymbol{u}\| = 1\}$, and observe that $(\mathcal{S}_4, 2 * \|\cdot\|)$ and $(\mathcal{S}_5, \|\cdot\|)$ are isometric. Therefore, it is enough to find a $4\delta$ packing of $\mathcal{S}_5$.

Now observe that for any $\boldsymbol{u}, \boldsymbol{v} \in \mathcal{S}_5$, $\|\boldsymbol{u} - \boldsymbol{v}\| \geq 4 * 0.25 \iff \boldsymbol{u}^T \boldsymbol{v} \leq \tfrac{1}{2}$. Therefore, for large enough $k, d$, by Corollary A.1.1, we have that the size ($N$) of a $2\delta$ packing of $\mathcal{S}$ satisfies,

$$\ln(N) \geq 0.15kd. \tag{25}$$

Note from Lemma B.1, that the KL divergence between any two distinct distributions, $P, Q$, corresponding to the $2\delta$ packing satisfies $\text{KL}(P \parallel Q) \leq \tfrac{1}{2\sigma^2}$. Applying Fano's Theorem for Randomized Algorithms 5.1, we get,

$$\mathfrak{M}_n(\Theta) \geq 0.25 \left(1 - \tfrac{n/2\sigma^2 + \ln(2)}{0.15kd}\right), \tag{26}$$

which implies that $n = \Omega(\sigma^2 kd)$, completing the proof.    $\square$

**Theorem C.1.** *Let $\mathcal{F}$ denote the class of functions represented by the set of fully connected neural network models, $\mathcal{M}_\mathcal{F}[\mathcal{W}]$, as defined in 4. Let $S^n \sim (SSD_1)^n$ be the $n$ i.i.d. data samples drawn from $SSD_1$, with $\sigma = \tilde{O}(1/\sqrt{k})$, and $k = O(\exp(d))$. Define the equivariance group $\mathcal{U} \coloneqq \mathcal{O}(kd)$. Define the subset $\tilde{\mathcal{U}} \subseteq \mathcal{U}$ such that, for all $\mathbf{U} \in \tilde{\mathcal{U}}$, $t \in \{2, \ldots, k\}$, $\mathbf{U}\boldsymbol{\mu}_t = e_{kd-k+t}$. Let $\xi \in \Xi$ encapsulate the randomization, and let $\xi \sim P_\Xi$. Let $\Theta = \{\theta \mid \theta \colon ((\mathcal{X}, \mathcal{Y})^n \times \Xi) \to \mathcal{F}\}$ be the set of $\mathcal{U}$-equivariant algorithms, such that $b^T = b_{\min} \coloneqq 10^{-2}$, then for large enough $k, d$,*

$$\inf_{\theta \in \Theta} \sup_{\mathbf{U} \in \tilde{\mathcal{U}}} \mathbb{E}_{\xi \sim P_\Xi} \mathbb{E}_{S^m \sim (\mathbf{U} \circ SSD_1)^m} [R(\theta(S^m, \xi), \mathbf{U} \circ SSD_1)] \leq \delta, \tag{27}$$

*iff $n = \Omega(\sigma^2 kd)$, for $\delta = 0.5 \times 10^{-2}$.*

*Proof.* The proof proceeds by reducing the problem of finding a fully-connected neural network with a small expected risk to a problem of estimating the unknown mean of a Gaussian distribution. We then use Lemma C.1 to establish the required sample complexity bound.

For brevity, we refer to the distribution $\text{SSD}_1$ by $P$. If any algorithm $\bar{\theta}_n \in \Theta$ achieves the maximum expected risk of $\delta$, then we have,

$$\sup_{\mathbf{U} \in \tilde{\mathcal{U}}} \mathbb{E}\left[R\left(\bar{\theta}_n, \mathbf{U} \circ P\right)\right] \leq \delta \implies \forall\, \mathbf{U},\, \mathbb{E}\left[R\left(\bar{\theta}_n, \mathbf{U} \circ P\right)\right] \leq \delta \qquad (28)$$

$$\stackrel{\text{Markov}}{\implies} \forall\, \mathbf{U},\, \mathbb{P}\left[R\left(\bar{\theta}_n, \mathbf{U} \circ P\right) \geq 0.5\right] \leq 2\delta \qquad (29)$$

Let the parameter vector of the fully connected neural network (FCN) that is returned by the algorithm $\bar{\theta}_n$ be given by $\boldsymbol{v} = [\boldsymbol{w}_1, .., \boldsymbol{w}_k, b]$. We define $\cos(\alpha_i) = (\mathbf{U}\boldsymbol{\mu}_1)^T \boldsymbol{w}_i$ as the alignment between the mean of the $\mathbf{U} \circ P$ distribution and the parameter $\boldsymbol{w}_i$. Then, the following holds:

$$R\left(\bar{\theta}_n, \mathbf{U} \circ P\right) < 0.5 \implies \exists\, i \in [k],\, \cos(\alpha_i) \geq b/2. \qquad (30)$$

We prove this via contradiction. For an i.i.d. data sample $(\boldsymbol{x}, y) \sim \mathbf{U} \circ P$, consider the the push-forward of the sample $y\boldsymbol{x}$ through the FCN. If for all $i \in [k]$, $\cos(\alpha_i) < b/2$, then the probability that the for each FCN node, its push-forward is $< 0$, is given by $\geq 1 - k\Phi(-b_{\min}/2\sigma) := 1 - p$, where $p$ can be made arbitrarily small for large enough $k, d$. Therefore $(y - \sum_{i=1}^{k} \phi_b(\boldsymbol{w}_i^T \boldsymbol{x}))^2 \geq 1$ with probability $\geq 1 - p$, which results in a contradiction, because the expected risk is $\leq \delta$.

**Mean estimation minimax problem**
Consider the following problem: Let $\mathcal{P} := \{\mathbf{U} \circ P | \mathbf{U} \in \tilde{\mathcal{U}}\}$ be the set of distributions. Let $m = (n+1)100/b_{\min}^2$ be the number of samples. Let $\Xi_1$ be the sample space that encapsulates algorithmic randomness, and $P_{\Xi_1}$ be a distribution over $\Xi_1$. Let $\Theta_1 := \{\theta \colon (\mathcal{X}^m, \Xi_1) \to \mathbb{S}^{kd-1}\}$ be the set of $\mathcal{U}$-equivariant randomized algorithms that estimate the mean of the input distribution using $m$ i.i.d. samples. Then the minimax problem is,

$$\inf_{\theta_m \in \Theta_1} \sup_{\mathbf{U} \in \tilde{\mathcal{U}}} \mathbb{E}_{S^m \sim (\mathbf{U} \circ P)^m, \xi \sim P_{\Xi_1}} \|\theta_m - \mathbf{U}\boldsymbol{\mu}_1\| \qquad (31)$$

We will now propose an $\mathcal{U}$-equivariant algorithm $\hat{\theta}_m$ for the above problem, that uses the algorithm $\bar{\theta}_n$ as a subroutine such that it achieves a constant max error of $1/4$,

$$\sup_{\mathbf{U} \in \mathcal{U}} \mathbb{E}_{S^m \sim (\mathbf{U} \circ P)^n, \xi \sim P_{\Xi_1}} \|\hat{\theta}_m - \mathbf{U}\boldsymbol{\mu}_1\| \leq 1/4. \qquad (32)$$

**Identification procedure**
Before we define $\hat{\theta}_m$, we provide a method, which given a FCN, can *identify* (one of) the parameter $\boldsymbol{w}_i$ such that $\cos(\alpha_i) \geq b_{\min}/2$, with probability $\geq 1 - p$, if there exists such a parameter. Otherwise, the method returns nothing, indicating that such a parameter does not exist, and this indication is correct with probability $\geq 1 - p$. The method is to push-forward an i.i.d. data sample $(\boldsymbol{x}, y) \sim \mathbf{U} \circ P$, through the neural network, and return the parameter corresponding to the first node whose output was positive. In the first case, the probability that none of the nodes with $\cos(\alpha_i) \geq b_{\min}/2$ have a positive output, or any nodes with $\cos(\alpha_i) < b_{\min}/2$ has a positive output is $\leq k\Phi(-b_{\min}/2\sigma) = p$. In the second case, the probability that none of the nodes with $\cos(\alpha_i) < b_{\min}/2$ have a positive output is also $\geq 1 - k\Phi(-b_{\min}/2\sigma) = 1 - p$. This concludes the proof.

**$\hat{\theta}_m$ definition and analysis**
The algorithm $\hat{\theta}_m$, divides the data into $S = 1000/b_{\min}^2$ sections, each of size $n + 1$. In each section, $s$, it runs the algorithm $\bar{\theta}_n$ on $n$ training samples, and uses the remaining sample to identify the parameter $\boldsymbol{w}_s$, such that $\boldsymbol{w}_s^T \mathbf{U}\boldsymbol{\mu}_1 \geq b_{\min}/2$ using the procedure outlined above. If this procedure returns nothing, then we set $\boldsymbol{w}_s = \mathbf{0}$. Finally, it projects the sum of these $s$ vectors to the unit sphere, $\hat{\boldsymbol{\mu}} = \frac{\sum_s \boldsymbol{w}_s}{\|\sum_s \boldsymbol{w}_s\|}$. In case, the sum $\sum_s \boldsymbol{w}_s = \mathbf{0}$, the algorithm returns $\boldsymbol{e}_1$.

Analysis: We now show that for all $\mathbf{U}$, $\mathbb{E}\|\hat{\boldsymbol{\mu}} - \mathbf{U}\boldsymbol{\mu}_1\| \leq 1/4$. We begin with the observation that if the identification procedure did not fail, then the random variable $\boldsymbol{w}_s$ can be written as,

$$\boldsymbol{w}_s = \lambda \boldsymbol{\mu}_1 + \sqrt{1 - \lambda^2} \boldsymbol{\mu}_1^\perp, \qquad (33)$$

where $\lambda, \boldsymbol{\mu}_1^\perp$ are both random variables, such that $\lambda \geq b$, and $\boldsymbol{\mu}_1^T \boldsymbol{\mu}_1^\perp = 0$. We define the $kd - 2$ dimensional unit sphere $\mathcal{S} = \{\boldsymbol{u} | \boldsymbol{u}^T \boldsymbol{\mu}_1^\perp = 0, \|\boldsymbol{u}\| = 1\}$. Then, we claim that $\boldsymbol{\mu}_1^\perp \sim \text{Uniform}(\mathcal{S})$. To see this, consider two points $\boldsymbol{u}, \boldsymbol{v} \in \mathcal{S}$. Then, there exists a $\mathbf{U}_1 \in \mathcal{U}$, such that $\mathbf{U}_1 \boldsymbol{\mu}_1 = \boldsymbol{\mu}_1$, and $\mathbf{U}_1 \boldsymbol{u} = \boldsymbol{v}$. Now consider running $\hat{\theta}_m$ on the input data $\{\boldsymbol{x}_i, y_i\}_m \sim \mathbf{U} \circ P$ such that $\boldsymbol{\mu}_1^\perp = \boldsymbol{u}$.

Instead, if we were to run $\hat{\theta}_m$ on $\{\mathbf{U}_1 \boldsymbol{x}_i, y_i\}_m \sim \mathbf{U}_1 \mathbf{U} \circ P$, then observe that $\boldsymbol{\mu}_1^\perp = \boldsymbol{v}$. Therefore, $\mathbb{P}_{\mathbf{U}}[\boldsymbol{\mu}_1^\perp = \boldsymbol{u}] = \mathbb{P}_{\mathbf{U}_1 \mathbf{U}}[\boldsymbol{\mu}_1^\perp = \boldsymbol{v}]$. Also, note that $\mathbf{U} \mathbf{U}_1 \circ P = \mathbf{U} \circ P$, because Gaussian is an equivariant distribution. Therefore, $\mathbb{P}_{\mathbf{U}}[\boldsymbol{\mu}_1^\perp = \boldsymbol{u}] = \mathbb{P}_{\mathbf{U}}[\boldsymbol{\mu}_1^\perp = \boldsymbol{v}]$. Since $\mathbf{U}, \boldsymbol{u}, \boldsymbol{v}$ were arbitrary, this proves the claim that $\boldsymbol{\mu}_1^\perp \sim \mathrm{Uniform}(\mathcal{S})$.

Let $\tilde{m}$ be the number of sections for which $\boldsymbol{w}_s \neq \mathbf{0}$. By Chernoff bound on the Bernouilli random variable corresponding to the event $R\left(\bar{\theta}_n, \mathbf{U} \circ P\right) \geq 0.5$, and the Identification procedure analysis, $\tilde{m} \geq 100/b_{\min}^2$ with probability $\geq 1 - \exp(-450/b_{\min}^2) - 1000p/b_{\min}^2 \geq 1 - 10^{-10}$. Now observe,

$$\boldsymbol{\mu}_1^T \left( \frac{\sum_{\boldsymbol{w}_s \neq \mathbf{0}} \boldsymbol{w}_s}{\tilde{m}} \right) = \boldsymbol{\mu}_1^T \left( \frac{\sum_{\boldsymbol{w}_s \neq \mathbf{0}} \lambda_s \boldsymbol{\mu}_1 + \sqrt{1 - \lambda_s^2} \boldsymbol{\mu}_{1,s}^\perp}{\tilde{m}} \right) \tag{34}$$

$$\geq b_{\min} + \boldsymbol{\mu}_1^T \frac{\sum_{\boldsymbol{w}_s \neq \mathbf{0}} \sqrt{1 - \lambda_s^2} \boldsymbol{\mu}_{1,s}^\perp}{\tilde{m}} \tag{35}$$

$$\geq b_{\min} + \boldsymbol{\mu}_1^T \left( \frac{1}{\tilde{m}} \sum_{\boldsymbol{w}_s \neq \mathbf{0}} \sqrt{1 - \lambda_s^2} \frac{\boldsymbol{\epsilon}_s}{\|\boldsymbol{\epsilon}_s\|} \right), \tag{36}$$

where $\boldsymbol{\epsilon}_s \sim \mathcal{N}(\mathbf{0}, \mathbf{I}_{kd})$. By the concentration of Gaussian norm, $0.5\sqrt{kd} \leq \|\boldsymbol{\epsilon}_s\| \leq 2\sqrt{kd}$, for all $s$, with probability $\geq 1 - 10^{-10}$, for large enough $k, d$.

$$\boldsymbol{\mu}_1^T \left( \frac{\sum_{\boldsymbol{w}_s \neq \mathbf{0}} \boldsymbol{w}_s}{\tilde{m}} \right) \geq b_{\min} + \boldsymbol{\mu}_1^T \left( \frac{1}{\tilde{m}} \sum_{\boldsymbol{w}_s \neq \mathbf{0}} \sqrt{1 - \lambda_s^2} \frac{\boldsymbol{\mu}_1^T \boldsymbol{\epsilon}_s}{\|\boldsymbol{\epsilon}_s\|} \right), \tag{37}$$

$$\geq b_{\min} + \left( \frac{1}{\tilde{m}} \sum_{\boldsymbol{w}_s \neq \mathbf{0}} \sqrt{1 - \lambda_s^2} \frac{\boldsymbol{\epsilon}_s}{\|\boldsymbol{\epsilon}_s\|} \right), \tag{38}$$

$$\geq b_{\min} + \left( \frac{1}{\tilde{m}} \sum_{\boldsymbol{w}_s \neq \mathbf{0}} \sqrt{1 - \lambda_s^2} \frac{\boldsymbol{\epsilon}_s}{\|\boldsymbol{\epsilon}_s\|} \right) \geq 0.99 b_{\min}, \tag{39}$$

with probability $\geq 1 - 10^{-10}$, for large enough $k, d$, using the Gaussian CDF. Also, we analyze,

$$\frac{1}{\tilde{m}} \| \sum_{\boldsymbol{w}_s \neq \mathbf{0}} \boldsymbol{w}_s \| \leq \left\| \frac{\sum_{\boldsymbol{w}_s \neq \mathbf{0}} \lambda_s \boldsymbol{\mu}_1 + \sqrt{1 - \lambda_s^2} \boldsymbol{\mu}_{1,s}^\perp}{\tilde{m}} \right\| \tag{40}$$

$$\leq b_{\min} + \left\| \frac{\sum_{\boldsymbol{w}_s \neq \mathbf{0}} \sqrt{1 - \lambda_s^2} \boldsymbol{\epsilon}_{1,s}^\perp}{\|\boldsymbol{\epsilon}_{1,s}\| \tilde{m}} \right\| \tag{41}$$

$$\leq b_{\min} + \left\| \frac{\sum_{\boldsymbol{w}_s \neq \mathbf{0}} 2\sqrt{1 - \lambda_s^2} \boldsymbol{\epsilon}_{1,s}^\perp}{\sqrt{kd} \, \tilde{m}} \right\| \tag{42}$$

$$\leq b_{\min} + \left\| 2 \frac{\boldsymbol{\epsilon}}{\sqrt{kd} \sqrt{\tilde{m}}} \right\| \leq b_{\min} + 0.4 b_{\min} \leq 1.4 b_{\min}. \tag{43}$$

Combining the dot-porduct and norm analyses, $\hat{\boldsymbol{\mu}}^T \boldsymbol{\mu}_1 \geq 0.7$ with probability $\geq 1 - 10^{-9}$. Therefore the expected risk of $\hat{\theta}_m$ is,

$$\mathbb{E} \| \hat{\theta}_m - \mathbf{U} \boldsymbol{\mu}_1 \| \leq \sqrt{2(1 - 0.7)} - \sqrt{2} * 10^{-9} \leq 1/4. \tag{44}$$

Using Lemma C.1, we have that,

$$\frac{100}{b_{\min}^2} (n + 1) = \Omega(\sigma^2 kd) \implies n = \Omega(\sigma^2 kd). \tag{45}$$

$\square$

**Theorem 6.1** (Formal). Let $\mathcal{F}$ denote the class of functions represented by the set of fully connected neural network models, $\mathcal{M}_{\mathcal{F}}[\mathcal{W}]$, as defined in 4. Let $S^n \sim (\mathrm{DSD})^n$ be the $n$ i.i.d. data samples drawn from DSD, with $\sigma = \tilde{O}(\frac{1}{\sqrt{k}})$, and $k = O(\exp(d))$. Define the equivariance group $\mathcal{U} = \mathcal{O}(kd)$, let $\{F^t\}_T$ be a set of update functions, and let the model parameters be initialized as $\boldsymbol{w}^0 \sim W$, for some distribution $W$. If the algorithm, $\bar{\theta}_n(S^n, \boldsymbol{w}^0; \mathcal{M}_F[\mathcal{W}], \{F_t\}_T)$, is $\mathcal{U}$-equivariant, such that $b^T \geq b_{\min}$, then for large enough $k, d$,

$$n_\delta(\bar{\theta}_n) = \max(\Omega\left(\sigma^2 k^2 d\right), 40k), \tag{46}$$

where $\delta = 0.25 \times 10^{-2}$, $b_{\min} := 10^{-2}$.

*Proof.* For simplicity we refer to the distribution DSD by $P$, and the distribution $SSD_t$ by $Q_t$, for $t \in [k]$. Since the algorithm $\bar{\theta}_n$ is $\mathcal{U}$-equivariant, lemma 5.1 gives us that for all $\mathbf{U} \in \mathcal{U}$,

$$\bar{\theta}(\{\boldsymbol{x}_i, y_i\}_n)(\boldsymbol{x}) \stackrel{d}{=} \bar{\theta}(\{\mathbf{U}\boldsymbol{x}_i, y_i\}_n)(\mathbf{U}\boldsymbol{x}), \tag{47}$$

$$\mathrm{err}\left(\bar{\theta}(\{\boldsymbol{x}_i, y_i\}_n)(\boldsymbol{x}), y\right) \stackrel{d}{=} \mathrm{err}\left(\bar{\theta}(\{\mathbf{U}\boldsymbol{x}_i, y_i\}_n)(\mathbf{U}\boldsymbol{x}), y\right), \tag{48}$$

$$\mathbb{E}_{S^n \sim P^n} \mathbb{E}_{(\boldsymbol{x},y) \sim P} \mathrm{err}\left(\bar{\theta}(\{\boldsymbol{x}_i, y_i\}_n)(\boldsymbol{x}), y\right) = \mathbb{E}_{S^n \sim P^n} \mathbb{E}_{(\boldsymbol{x},y) \sim P} \mathrm{err}\left(\bar{\theta}(\{\mathbf{U}\boldsymbol{x}_i, y_i\}_n)(\mathbf{U}\boldsymbol{x}), y\right), \tag{49}$$

$$\mathbb{E}_{S^n \sim P^n}\left[R\left(\bar{\theta}_n, P\right)\right] = \mathbb{E}_{S^n \sim (\mathbf{U} \circ P)^n}\left[R\left(\bar{\theta}_n, \mathbf{U} \circ P\right)\right], \tag{50}$$

$$= \mathbb{E}_{S^n \sim (\mathbf{U} \circ P)^n} \frac{1}{k} \sum_{i=1}^{k}\left[R\left(\bar{\theta}_n, \mathbf{U} \circ Q_i\right)\right], \tag{51}$$

$$= \frac{1}{k} \sum_{i=1}^{k} \mathbb{E}_{S^n \sim (\mathbf{U} \circ P)^n}\left[R\left(\bar{\theta}_n, \mathbf{U} \circ Q_i\right)\right]. \tag{52}$$

To simplify 52, we begin by showing that the expected risk incurred by the algorithm is the same for every distribution $\mathbf{U} \circ Q_i$. Specifically, for all $i, j \in [k]$,

$$\mathbb{E}_{S^n \sim (\mathbf{U} \circ P)^n}\left[R\left(\bar{\theta}_n, \mathbf{U} \circ Q_i\right)\right] = \mathbb{E}_{S^n \sim (\mathbf{U} \circ P)^n}\left[R\left(\bar{\theta}_n, \mathbf{U} \circ Q_j\right)\right]. \tag{53}$$

For $i = j$, the 53 trivially holds. So we can assume that $i \neq j$. From the definition of DSD, note that for any $\alpha, \beta \in [k]$, we have that $\boldsymbol{\mu}_\alpha^T \boldsymbol{\mu}_\beta = \mathbf{1}[\alpha = \beta]$. Consequently, for any $\mathbf{U} \in \mathcal{U}$, we observe that,

$$(\mathbf{U}\boldsymbol{\mu}_\alpha)^T \mathbf{U}\boldsymbol{\mu}_\beta = \boldsymbol{\mu}_\alpha^T \mathbf{U}^T \mathbf{U}\boldsymbol{\mu}_\beta = \boldsymbol{\mu}_\alpha^T \boldsymbol{\mu}_\beta = \mathbf{1}[\alpha = \beta] \tag{54}$$

The above fact ensures that for all $\mathbf{U} \in \mathcal{U}$, there exists a $\mathbf{U}_1 \in \mathcal{U}$, such that for all $\alpha \in [k] \setminus \{i, j\}$, $\mathbf{U}_1 \mathbf{U}\boldsymbol{\mu}_\alpha = \mathbf{U}\boldsymbol{\mu}_\alpha$, and that $\mathbf{U}_1 \mathbf{U}\boldsymbol{\mu}_i = \mathbf{U}\boldsymbol{\mu}_j$ and $\mathbf{U}_1 \mathbf{U}\boldsymbol{\mu}_j = \mathbf{U}\boldsymbol{\mu}_i$. In other words, the map $\mathbf{U}_1$ swaps the vectors $\mathbf{U}\boldsymbol{\mu}_i$, and $\mathbf{U}\boldsymbol{\mu}_j$, and keeps the other vectors unchanged. We again use the $\mathcal{U}$-equivariance of $\bar{\theta}_n$ to infer from lemma 5.1 that,

$$\bar{\theta}(\{\boldsymbol{x}_i, y_i\}_n)(\boldsymbol{x}) \stackrel{d}{=} \bar{\theta}(\{\mathbf{U}_1\boldsymbol{x}_i, y_i\}_n)(\mathbf{U}_1\boldsymbol{x}), \tag{55}$$

$$\mathrm{err}\left(\bar{\theta}(\{\boldsymbol{x}_i, y_i\}_n)(\boldsymbol{x}), y\right) \stackrel{d}{=} \mathrm{err}\left(\bar{\theta}(\{\mathbf{U}_1\boldsymbol{x}_i, y_i\}_n)(\mathbf{U}_1\boldsymbol{x}), y\right), \tag{56}$$

$$\mathbb{E}_{S^n \sim (\mathbf{U} \circ P)^n} \mathbb{E}_{\mathbf{U} \circ Q_i}\left[\mathrm{err}\left(\bar{\theta}(\{\boldsymbol{x}_i, y_i\}_n)(\boldsymbol{x}), y\right)\right] = \mathbb{E}_{S^n \sim (\mathbf{U} \circ P)^n} \mathbb{E}_{\mathbf{U} \circ Q_i}\left[\mathrm{err}(\bar{\theta}(\{\mathbf{U}_1\boldsymbol{x}_i, y_i\}_n) \right.$$
$$\left. (\mathbf{U}_1\boldsymbol{x}), y)\right], \quad (57)$$

$$\mathbb{E}_{S^n \sim (\mathbf{U} \circ P)^n} \mathbb{E}_{\mathbf{U} \circ Q_i}\left[\mathrm{err}\left(\bar{\theta}(\{\boldsymbol{x}_i, y_i\}_n)(\boldsymbol{x}), y\right)\right] = \mathbb{E}_{S^n \sim (\mathbf{U}_1 \mathbf{U} \circ P)^n} \mathbb{E}_{\mathbf{U}_1 \mathbf{U} \circ Q_i}\left[\mathrm{err}(\bar{\theta}(\{\boldsymbol{x}_i, y_i\}_n) \right.$$
$$\left. (\boldsymbol{x}), y)\right]. \quad (58)$$

From construction of $\mathbf{U}_1$, we know that $\mathbf{U}_1 \mathbf{U} \circ P \stackrel{d}{=} \mathbf{U} \circ P$, and $\mathbf{U}_1 \mathbf{U} \circ Q_i \stackrel{d}{=} \mathbf{U} \circ Q_j$,

$$\mathbb{E}_{S^n \sim (\mathbf{U} \circ P)^n} \mathbb{E}_{\mathbf{U} \circ Q_i}\left[\mathrm{err}\left(\bar{\theta}(\{\boldsymbol{x}_i, y_i\}_i)(\boldsymbol{x}), y\right)\right] = \mathbb{E}_{S^n \sim (\mathbf{U} \circ P)^n} \mathbb{E}_{\mathbf{U} \circ Q_j}\left[\mathrm{err}\left(\bar{\theta}(\{\boldsymbol{x}_i, y_i\}_i)(\boldsymbol{x}), y\right)\right], \tag{59}$$

$$\mathbb{E}_{S^n \sim (\mathbf{U} \circ P)^n}\left[R\left(\bar{\theta}_n, \mathbf{U} \circ Q_i\right)\right] = \mathbb{E}_{S^n \sim (\mathbf{U} \circ P)^n}\left[R\left(\bar{\theta}_n, \mathbf{U} \circ Q_j\right)\right]. \tag{60}$$

This proves the claim 53. Substituting it back into 52,

$$\mathbb{E}_{S^n \sim P^n}\left[R\left(\bar{\theta}_n, P\right)\right] = \mathbb{E}_{S^n \sim (\mathbf{U} \circ P)^n}\left[R\left(\bar{\theta}_n, \mathbf{U} \circ Q_1\right)\right], \tag{61}$$

$$= \sup_{\mathbf{U} \in \tilde{\mathcal{U}}} \mathbb{E}_{S^n \sim (\mathbf{U} \circ P)^n}\left[R\left(\bar{\theta}_n, \mathbf{U} \circ Q_1\right)\right], \tag{62}$$

where, we define $\tilde{\mathcal{U}} \subseteq \mathcal{U}$ such that, for all $\mathbf{U} \in \tilde{\mathcal{U}}$, $t \in \{2, \ldots, k\}$, $\mathbf{U}\boldsymbol{\mu}_t = e_{kd-k+t}$. Let $\Xi = \mathcal{W}$, $P_\Xi = W$, and $\Theta = \{\theta \mid \theta \colon ((\mathcal{X}, \mathcal{Y})^n, \Xi) \to \mathcal{F}\}$ be the set of $\mathcal{O}(kd)$ equivariant algorithms, such that $b^T \geq b_{\min}$. It is easy to note that $\bar{\theta}_n \in \Theta$. Therefore,

$$\mathbb{E}_{S^n \sim P^n}\left[R\left(\bar{\theta}_n, P\right)\right] \geq \inf_{\theta_n \in \Theta} \sup_{\mathbf{U} \in \tilde{\mathcal{U}}} \mathbb{E}_{S^n \sim (\mathbf{U} \circ P)^n}\left[R\left(\theta_n, \mathbf{U} \circ Q_1\right)\right], \tag{63}$$

We will now perform a series of reductions to lower bound the above minimax problem, with the minimax problem of learning $SSD_1$. The central concept behind these reductions is to demonstrate that a given minimax problem can be 'simulated' by a more tractable one, and thus the tractable problem serves as a lower bound on the original problem.

Let $\Theta_1 = \{\theta \mid \theta \colon (((\mathcal{X}, \mathbb{Z}_+), \mathcal{Y})^n, \Xi) \to \mathcal{F}\}$ be the set of algorithms that are $\mathcal{U}_1 = \{\mathrm{Block}(\mathbf{U}, \mathbf{I}_1) \mid \mathbf{U} \in \mathcal{O}(kd)\}$ equivariant, and $b^T \geq b_{\min}$. Define the set $\tilde{\mathcal{U}}_1 \subseteq \mathcal{U}_1$, $\tilde{\mathcal{U}}_1 = \{\mathrm{Block}(\mathbf{U}, \mathbf{I}_1) \mid \mathbf{U} \in \tilde{\mathcal{U}}\}$. Define $\tilde{P}$ to be the indexed distribution with the generative story: Sample $j \sim \mathrm{Unif}[k]$, then sample $(\boldsymbol{x}, y) \sim Q_j$, and return $((\boldsymbol{x}, j), y)$. We can then lower bound the minimax expression 63 as,

$$\geq \inf_{\theta \in \Theta_1} \sup_{\mathbf{U}_1 \in \tilde{\mathcal{U}}_1} \mathbb{E}_{\boldsymbol{w} \sim W} \mathbb{E}_{((\boldsymbol{x},j),y)^n \sim (\mathbf{U}_1 \circ \tilde{P})^n} \left[ R\left(\theta(((\boldsymbol{x}, j), y)^n, \boldsymbol{w}), \mathbf{U} \circ Q_1\right) \right]. \tag{64}$$

The inequality follows from the fact that for every $\theta_n^a \in \Theta$, there exists $\theta_n^b \in \Theta_1$, that discards the index $j$ and returns the output of $\theta_n^a$.

Let $n_1$ be the random variable that denotes the number of samples drawn from $\mathbf{U}_1 \circ \tilde{P}$ when $j = 1$. Using Berstein's inequality, we get that, $\frac{n}{2k} \leq n_1 \leq m := \frac{3n}{2k}$, with probability $\geq c := 1 - 2\exp(\frac{-n}{10k})$. We will refer to the event, $\frac{n}{2k} \leq n_1 \leq m$, as $E$. Then, we can lower bound 64 as,

$$\geq c \inf_{\theta \in \Theta_1} \sup_{\mathbf{U}_1 \in \tilde{\mathcal{U}}_1} \mathbb{E}_{\boldsymbol{w} \sim W} \mathbb{E}_{((\boldsymbol{x},j),y)^n \sim (\mathbf{U}_1 \circ \tilde{P})^n} \left[ R\left(\theta(((\boldsymbol{x}, j), y)^n, \boldsymbol{w}), \mathbf{U} \circ Q_1\right) \mid E \right]. \tag{65}$$

For the next reduction, we generalize the definition of $n_1$, and define $n_i$ to be the random variable corresponding to the number of samples drawn from the distribution $\mathbf{U} \circ Q_i$, for all $i \in [k]$. Let $\boldsymbol{y} \sim (\mathrm{Unif}[\mathcal{Y}])^n$ be a uniform random vector over $\{+1, -1\}$ of size $n$, and $\boldsymbol{\epsilon} \sim \mathcal{N}(\mathbf{0}_{nkd}, \mathbf{I}_{nkd})$ be a vector of i.i.d. standard Gaussian random variables. Let $\Theta_2 = \{\theta \mid \theta \colon ((\mathcal{X}, \mathcal{Y})^m \times ((\mathbb{R}^{kd})^{k-1} \times (\mathbb{N} \cup \{0\})^k \times (\mathbb{R})^n \times \mathbb{R}^{nkd} \times \mathcal{W})) \to \mathcal{F}\}$ be a set of $\mathcal{O}(kd)$ equivariant algorithms that take as input the training data, $\{\mathbf{U}\boldsymbol{\mu}\}_{i=2}^k$ mean vectors, the number of samples to be drawn from each mean, pre-sampled values of $\boldsymbol{y}$ and $\boldsymbol{\epsilon}$, and the parameter initialization respectively. It subsequently returns a function within $\mathcal{F}$. Then we can bound 65 as,

$$\geq c \inf_{\theta \in \Theta_2} \sup_{\mathbf{U} \in \tilde{\mathcal{U}}} \mathbb{E}_{\boldsymbol{w} \sim W} \mathbb{E}_{\{n_i\}_1^k} \mathbb{E}_{\boldsymbol{y}, \boldsymbol{\epsilon}} \mathbb{E}_{S^m \sim (\mathbf{U} \circ Q_1)^m} \left[ R\left(\theta(S^m, (\{\mathbf{U}\boldsymbol{\mu}_i\}_2^k, \{n_i\}_1^k, \boldsymbol{y}, \boldsymbol{\epsilon}, \boldsymbol{w})), \mathbf{U} \circ Q_1\right) \mid E \right]. \tag{66}$$

The last inequality follows from the fact that for every $\theta_n^a \in \Theta_1$, there exists $\theta_n^b \in \Theta_2$, that first deterministically creates the indexed dataset using $S^m, \{\mathbf{U}\boldsymbol{\mu}_i\}_2^k, \{n_i\}_1^k, \boldsymbol{y}, \boldsymbol{\epsilon}$ and then runs $\theta_n^a$.

For notational brevity, we define $\Xi_1 := (\mathbb{R}^{kd})^{k-1} \times (\mathbb{N} \cup \{0\})^k \times (\mathbb{R})^n \times \mathbb{R}^{nkd} \times \mathcal{W}$, to encapsulate the randomness in $\{\mathbf{U}\boldsymbol{\mu}_i\}_2^k, \{n_i\}_1^k, \boldsymbol{y}, \boldsymbol{\epsilon}$, and $\boldsymbol{w}$. We denote its associated product distribution by $P_{\Xi_1}$. Recall that this distribution must be independent of the input data distribution. To see this, we recall that from the construction of $\tilde{\mathcal{U}}$, that for all $\mathbf{U} \in \tilde{\mathcal{U}}, t \in \{2, \ldots, k\}, \mathbf{U}\boldsymbol{\mu}_t = e_{kd-k+t}$, which are fixed deterministic quantities. Rewriting 66, we get,

$$= c \inf_{\theta \in \Theta_2} \sup_{\mathbf{U} \in \tilde{\mathcal{U}}} \mathbb{E}_{\xi \sim P_{\Xi_1}} \mathbb{E}_{S^m \sim (\mathbf{U} \circ Q_1)^m} \left[ R\left(\theta(S^m, \xi), \mathbf{U} \circ Q_1\right) \right], \tag{67}$$

We have effectively reduced solving the original problem $P$ into solving its constituent problem $Q_1$ with approximately $n/k$ samples. We have already proven a lower bound for the SSD problem in Theorem 4. Using that result, we have,

$$\mathbb{E}_{S^n \sim P^n} \left[ R\left(\bar{\theta}_n, P\right) \right] \geq c * 0.5 * 10^{-2}, \tag{68}$$

iff $m = \Omega(\sigma^2 kd)$, which implies that $n = \Omega(\sigma^2 k^2 d)$. Since, $c \geq \left(1 - 2\exp(\frac{-n}{10k})\right)$, using $n \geq 40k$, we can bound $c \geq (1 - 2\exp(-\ln(4))) = \frac{1}{2}$. Therefore,

$$\mathbb{E}_{S^n \sim P^n} \left[ R\left(\bar{\theta}_n, P\right) \right] \geq 0.25 * 10^{-2} \tag{69}$$

iff $n = \Omega(\sigma^2 k^2 d)$, proving the result. $\qquad \square$

## C.2 LCN Upper Bound

**Theorem 6.2** (Formal). Let $\mathcal{F}$ denote the class of functions represented by the set of locally connected neural network models $\mathcal{M}_{\mathcal{L}}$, defined in equation 5. Let the input data samples be drawn from the DSD distribution, $S^n \sim (\text{DSD})^n$, with $\sigma = \tilde{O}(\frac{1}{\sqrt{k}})$, and $k = O(\exp(d))$. Define the following groups: $\mathcal{U}_1 := \{\text{Block}\,(\mathbf{U}_1, \ldots, \mathbf{U}_k) \mid \mathbf{U}_i \in \mathcal{O}(d)\}$, $\mathcal{U}_2 := \{\mathbf{U} \in \mathcal{O}_p(kd) \mid \text{idx}_{kd}(\mathbf{U}e_{(i-1)d+1}) + j - 1 = \text{idx}_{kd}(\mathbf{U}e_{(i-1)d+j}), \forall i \in [k], \forall j \in [d]\}$, and $\mathcal{U} := \mathcal{U}_1 \star \mathcal{U}_2$. Then, there exist update functions $\{F_t\}_T$, and a model parameter initialization distribution $W$, such that $\bar{\theta}_n(\mathcal{M}_L, \{F_t\}_T, W, S^n)$ is a $\mathcal{U}$-equivariant algorithm, and for large enough $k, d$,

$$n_\delta(\bar{\theta}_n) = \max(O\left(\sigma^2 k(d+k)\ln(kd)\right), 80k\ln(kd)), \tag{70}$$

for $\delta = O(1)$.

*Proof.* We begin by defining the algorithm $\bar{\theta}_n$, we then establish that $\bar{\theta}_n$ is a $\mathcal{U}$-equivariant algorithm, and then we analyze each iteration of the algorithm to prove the required sample complexity bound.

1. Algorithm Definition

To define the algorithm $\bar{\theta}_n(\mathcal{M}_L, \{F_t\}_T, W, S^n)$, we need to specify the initialization distribution $W$, and the update functions $\{F_t\}_T$. At iteration $t = 0$, we initialize the model parameter $v^0 = [w_1^0, .., w_k^0, b^0]$ as follows: for each $i \in [k]$, the vector $w_i^0$ is independently sampled from the distribution $\mathcal{N}(\mathbf{0}, \gamma \mathbf{I}_d)$, where $\gamma^{-1} = 100k^2d^2$, and bias is set as $b^0 = 0$. The superscript denotes the iteration number. We define the empirical loss function for $N \in \mathbb{Z}_+$ data samples as,

$$l: (\mathcal{W}, (\mathcal{X}, \mathcal{Y})^N) \to \mathbb{R} := \frac{1}{N} \sum_{j=1}^N \left(y_j - \sum_{i=1}^k \phi_b(w_i^T x_j^{(i)})\right)^2. \tag{71}$$

The algorithm proceeds in $T = 2$ iterations. For simpler analysis, we split the input dataset $S^n$ into two equal sized datasets $S_1^m$, and $S_2^m$, with $m := \frac{n}{2}$ samples each. Then, for each $t \in \{1, 2\}$,

$$F_t(v, S^n) := \left[\frac{w_1 - \eta_t \nabla_{w_1} l(v; S_t^m)}{\|w_1 - \eta_t \nabla_{w_1} l(v; S_t^m)\|}, \ldots, \frac{w_k - \eta_t \nabla_{w_k} l(v; S_t^m)}{\|w_k - \eta_t \nabla_{w_k} l(v; S_t^m)\|}, \ b_t\right], \tag{72}$$

where $\eta_1 = 1, \eta_2 = k \times 10^3, b_1 = \frac{1}{32}\sqrt{\frac{(k+d)\ln(kd)}{kd}}, b_2 = 10^{-4}$.

2. Algorithm is Equivariant

To establish that $\bar{\theta}_n$ is $\mathcal{U}$-equivariant, we only need to show that it is both $\mathcal{U}_1$- and $\mathcal{U}_2$-equivariant, because, every element in $\mathcal{U}$ is a finite matrix product of elements from $\mathcal{U}_1$ and $\mathcal{U}_2$. We define groups $\mathcal{V}_1 := \{\text{Block}(\mathbf{V}_1, \ldots, \mathbf{V}_k, \mathbf{I}_1) \mid \mathbf{V}_i \in \mathcal{O}(d), i \in [k]\}$, where $\mathbf{I}_1$ is the identity matrix of size $1 \times 1$, $\mathcal{V}_2 := \{\text{Block}(\mathbf{U}, \mathbf{I}_1) \mid \mathbf{U} \in \mathcal{U}_2\}$, and $\mathcal{V} := \mathcal{V}_1 \star \mathcal{V}_2$.

To prove $\mathcal{U}_1$-equivariance, we need to verify the three conditions in Definition 6. For any data sample $x, y \in (\mathcal{X}, \mathcal{Y})$, $\mathbf{U} \in \mathcal{U}_1$, $w_i \in \mathbb{R}^d; i \in [k], b \in \mathbb{R}_+$, choose $\mathbf{V} = \text{Block}(\{\mathbf{U}^{(1)}, \ldots, \mathbf{U}^{(k)}, \mathbf{I}_1\}) \in \mathcal{V}$. Then, the first property 1 holds as,

$$\mathcal{M}_L[v](x) = \sum_{i=1}^k \phi_b(w_i^T x^{(i)}) = \sum_{i=1}^k \phi_b(w_i^T (\mathbf{U}^{(i)})^T (\mathbf{U}^{(i)}) x^{(i)}) = \mathcal{M}_L[\mathbf{V}v](\mathbf{U}x). \tag{73}$$

For each iteration $t \in [2]$, and $S^n \in (\mathcal{X}, \mathcal{Y})^n$, the second property 2 follows as,

$$F_t\left(\mathbf{V}v, \mathbf{U} \circ S^n\right) = \left[\frac{\mathbf{U}^{(1)} w_1 - \eta_t \nabla_{\mathbf{U}^{(1)} w_1} l(\mathbf{V}v; \mathbf{U} \circ S_t^m)}{\|\mathbf{U}^{(1)} w_1 - \eta_t \nabla_{\mathbf{U}^{(1)} w_1} l(\mathbf{V}v; \mathbf{U} \circ S_t^m)\|}, \ldots, \right. \tag{74}$$

$$\left. \frac{\mathbf{U}^{(k)} w_k - \eta_t \nabla_{\mathbf{U}^{(k)} w_k} l(\mathbf{V}v; \mathbf{U} \circ S_t^m)}{\|\mathbf{U}^{(k)} w_k - \eta_t \nabla_{\mathbf{U}^{(k)} w_k} l(\mathbf{V}v; \mathbf{U} \circ S_t^m)\|}, \ b_t\right],$$

$$= \left[\frac{\mathbf{U}^{(1)}\left(w_1 - \eta_t \nabla_{w_1} l(v; S_t^m)\right)}{\|\mathbf{U}^{(1)}\left(w_1 - \eta_t \nabla_{w_1} l(v; S_t^m)\right)\|}, \ldots, \frac{\mathbf{U}^{(k)}\left(w_k - \eta_t \nabla_{w_k} l(v; S_t^m)\right)}{\|\mathbf{U}^{(k)}\left(w_k - \eta_t \nabla_{w_k} l(v; S_t^m)\right)\|}, \ b_t\right], \tag{75}$$

$$= \left[\frac{\mathbf{U}^{(1)}\left(w_1 - \eta_t \nabla_{w_1} l(v; S_t^m)\right)}{\|w_1 - \eta_t \nabla_{w_1} l(v; S_t^m)\|}, \ldots, \frac{\mathbf{U}^{(k)}\left(w_k - \eta_t \nabla_{w_k} l(v; S_t^m)\right)}{\|w_k - \eta_t \nabla_{w_k} l(v; S_t^m)\|}, \ b_t\right], \tag{76}$$

$$= \mathbf{V}F_t\left(v, S^n\right). \tag{77}$$

And property 3 can be affirmed by observing that, for all $\mathbf{V} \in \mathcal{V}_1$,

$$\mathbf{V}\boldsymbol{v}^0 = [\mathbf{U}^{(1)}\boldsymbol{w}_1^0, \ldots, \mathbf{U}^{(k)}\boldsymbol{w}_k^0, b^0], \tag{78}$$

$$\overset{d}{=} [\boldsymbol{w}_1^0, \ldots, \boldsymbol{w}_k^0, b^0] = \boldsymbol{v}. \tag{79}$$

This proves that $\bar{\theta}_n$ is $\mathcal{U}_1$-equivariant. We now establish $\mathcal{U}_2$-equivariance. Observe that action of any matrix $\mathbf{U} \in \mathcal{U}_2$ is to permute the $k$ patches of the input. Let $\pi \colon [k] \to [k]$ be the permutation function corresponding to $\mathbf{U}$. For this given $\mathbf{U}$, we choose $\mathbf{V} = \text{Block}(\mathbf{U}, \mathbf{I}_1) \in \mathcal{V}$. For any $\boldsymbol{x}, y \in (\mathcal{X}, \mathcal{Y})$, $\boldsymbol{w}_i \in \mathbb{R}^d, i \in [k], b \in \mathbb{R}_+$, property 1 of equivariance holds as,

$$\mathcal{M}_L[\boldsymbol{v}](\boldsymbol{x}) = \sum_{i=1}^k \phi_b(\boldsymbol{w}_i^T \boldsymbol{x}^{(i)}) = \sum_{i=1}^k \phi_b(\boldsymbol{w}_{\pi(i)}^T \boldsymbol{x}^{\pi(i)}) = \mathcal{M}_L[\mathbf{V}\boldsymbol{v}](\mathbf{U}\boldsymbol{x}). \tag{80}$$

For each iteration $t \in [2]$, and $S^n \in (\mathcal{X}, \mathcal{Y})^n$, the second property 2 follows as,

$$F_t(\mathbf{V}\boldsymbol{v}, \mathbf{U} \circ S^n) = \left[ \frac{\boldsymbol{w}_{\pi(1)} - \eta_t \nabla_{\pi(1)} l(\mathbf{V}\boldsymbol{v}; \mathbf{U} \circ S_t^m)}{\|\boldsymbol{w}_{\pi(1)} - \eta_t \nabla_{\boldsymbol{w}_{\pi(1)}} l(\mathbf{V}\boldsymbol{v}; \mathbf{U} \circ S_t^m)\|}, \ldots, \right. \tag{81}$$

$$\left. \frac{\boldsymbol{w}_{\pi(k)} - \eta_t \nabla_{\boldsymbol{w}_{\pi(k)}} l(\mathbf{V}\boldsymbol{v}; \mathbf{U} \circ S_t^m)}{\|\boldsymbol{w}_{\pi(k)} - \eta_t \nabla_{\boldsymbol{w}_{\pi(k)}} l(\mathbf{V}\boldsymbol{v}; \mathbf{U} \circ S_t^m)\|}, b_t \right],$$

$$= \mathbf{V} \left[ \frac{\boldsymbol{w}_1 - \eta_t \nabla_{\boldsymbol{w}_1} l(\boldsymbol{v}; S_t^m)}{\|\boldsymbol{w}_1 - \eta_t \nabla_{\boldsymbol{w}_1} l(\boldsymbol{v}; S_t^m)\|}, \ldots, \frac{\boldsymbol{w}_k - \eta_t \nabla_{\boldsymbol{w}_k} l(\boldsymbol{v}; S_t^m)}{\|\boldsymbol{w}_k - \eta_t \nabla_{\boldsymbol{w}_k} l(\boldsymbol{v}; S_t^m)\|}, b_t \right], \tag{82}$$

$$= \mathbf{V} F_t(\boldsymbol{v}, S^n). \tag{83}$$

And finally property 3 can be shown by observing that, for all $\mathbf{V} \in \mathcal{V}_2$,

$$\mathbf{V}\boldsymbol{v}^0 = [\boldsymbol{w}_{\pi(1)}^0, \ldots, \boldsymbol{w}_{\pi(k)}^0, b^0], \tag{84}$$

$$\overset{d}{=} [\boldsymbol{w}_1^0, \ldots, \boldsymbol{w}_k^0, b^0] = \boldsymbol{v}. \tag{85}$$

Thus, the algorithm $\bar{\theta}_n$ is $\mathcal{U}_2$-equivariant, and therefore is $\mathcal{U}$-equivariant.

3. Algorithm Analysis

We analyze each iteration of $\bar{\theta}_n$, with $n = \max(2\sigma^2 k(k+d)\ln(kd), 80k\ln(kd))$ samples, and establish that $\bar{\theta}_n$ achieve an expected risk of at most $\delta = 2.5 \times 10^{-3}$. Since $k = O(\exp(d))$ and $\sigma = \tilde{O}(\frac{1}{\sqrt{k}})$, we assume that $\sqrt{d} \geq 20\sqrt{\ln(k)}$ and $100\sqrt{k\ln(kd)^3}\sigma \leq 1$.

The outline of the analysis is as follows: We show that after the first update step, we reliably recover the unknown signal vector, upto an alignment of $\Omega(\sqrt{\frac{k+d}{kd}})$. In the second step, we show that this alignment is enough to threshold out the "noise" patches while only letting the "signal" patch pass through the first hidden layer. This enables us to recover the signal upto an alignment of $\Omega(1)$, which results in the expected risk of the learned LCN being smaller than $\delta$.

3a. Update Step 1

For each $i \in [k]$, we denote $\tilde{\boldsymbol{w}}_i^1 = \boldsymbol{w}_i^0 - \nabla_{\boldsymbol{w}_i^0} l(\boldsymbol{w}_i^0, 0; S_1^m)$ to be the unnormalized version of $\boldsymbol{w}_i^1$. Thus, the alignment of $\boldsymbol{w}_i^1$ with the signal can be written as, $(\boldsymbol{w}_i^1)^T \boldsymbol{w}^\star = \frac{(\tilde{\boldsymbol{w}}_i^1)^T \boldsymbol{w}^\star}{\|\tilde{\boldsymbol{w}}_i^1\|}$. To compute this alignment, we begin by simplifying $\nabla_{\boldsymbol{w}_i^0} l([\boldsymbol{w}_i^0, 0]; S_1^m)$,

$$\nabla_{\boldsymbol{w}_i^0} l([\boldsymbol{w}_i^0, 0]; S_1^m) = \frac{1}{m} \sum_{j=1}^m \nabla_{\boldsymbol{w}_i^0} \left( y_i - \sum_{i=1}^k \phi_0((\boldsymbol{w}_i^0)^T \boldsymbol{x}_j^{(i)}) \right)^2, \tag{86}$$

$$= \frac{-2}{m} \sum_{j=1}^m \left( y_j - \sum_{i=1}^k \phi_0((\boldsymbol{w}_i^0)^T \boldsymbol{x}_j^{(i)}) \right) \left( \boldsymbol{x}_j^{(i)} \phi_0'((\boldsymbol{w}_i^0)^T \boldsymbol{x}_j^{(i)}) \right), \tag{87}$$

$$= \frac{-2}{m} \sum_{j=1}^m \left( 1 - \sum_{i=1}^k y_j \phi_0((\boldsymbol{w}_i^0)^T \boldsymbol{x}_j^{(i)}) \right) \left( y_j \boldsymbol{x}_j^{(i)} \phi_0'((\boldsymbol{w}^0)^T \boldsymbol{x}_j^{(i)}) \right), \tag{88}$$

$$= \frac{-2}{m} \sum_{j=1}^m \left( 1 - \sum_{i=1}^k y_j (\boldsymbol{w}_i^0)^T \boldsymbol{x}_j^{(i)} \right) \left( y_j \boldsymbol{x}_j^{(i)} \right), \tag{89}$$

where $\phi_0'(x) := \frac{d}{dx}\phi_0(x)$, and the last equality follows by observing that $\phi_0$ is the identity function, and $\phi_0'$ is the constant function 1. As a shorthand, we define $\alpha_j := 1 - \sum_{i=1}^k y_j(\boldsymbol{w}_i^0)^T \boldsymbol{x}_j^{(i)})$, and $\boldsymbol{\beta}_{ij} := y_j \boldsymbol{x}_j^{(i)}$. Substituting this back into 89,

$$\nabla_{\boldsymbol{w}_i^0} l([\boldsymbol{w}_i^0, 0]; S_1^m) = \frac{-2}{m} \sum_{j=1}^m \alpha_j \boldsymbol{\beta}_{ij}, \tag{90}$$

To analyze 90, we begin by proving high probability bounds on the range of $\alpha_j$, for each $j \in [m]$. From the initialization procedure described above, we know that $\boldsymbol{w}_i^0 \stackrel{d}{=} \gamma \boldsymbol{\epsilon}_i$, where $\boldsymbol{\epsilon}_i$ is the Gaussian random vector defined as $\boldsymbol{\epsilon}_i \sim \mathcal{N}(\boldsymbol{0}, \mathbf{I}_d)$. And with the input distribution being DSD, we know that $\boldsymbol{x}_j^{(i)} = y_j r_{ij} \boldsymbol{w}^\star + \sigma \boldsymbol{\epsilon}_j^{(i)}$, for all $i$ in $[k]$, where $\boldsymbol{\epsilon}_j^{(i)} \sim \mathcal{N}(\boldsymbol{0}, \mathbf{I}_d)$ is also a Gaussian random vector, and $r_{ij} = 1$, if in the $j$-th data sample the signal patch appears in the $i$-th patch, and 0 otherwise.

$$\alpha_j = 1 - \sum_{i=1}^k y_j(\boldsymbol{w}_i^0)^T \boldsymbol{x}_j^{(i)}, \tag{91}$$

$$= 1 - \sum_{i=1}^k \gamma r_{ij} y_j^2 \boldsymbol{\epsilon}_i^T \boldsymbol{w}^\star - \sum_{i=1}^k \gamma \sigma y_j \boldsymbol{\epsilon}_i^T \boldsymbol{\epsilon}_j^{(i)}. \tag{92}$$

We can now bound the range of $\alpha_j$ as,

$$1 + |\sum_{i=1}^k \gamma r_{ij} \boldsymbol{\epsilon}_i^T \boldsymbol{w}^\star| + |\sum_{i=1}^k \gamma \sigma \boldsymbol{\epsilon}_i^T \boldsymbol{\epsilon}_j^{(i)}| \geq \alpha_j \geq 1 - |\sum_{i=1}^k \gamma r_{ij} \boldsymbol{\epsilon}_i^T \boldsymbol{w}^\star| - |\sum_{i=1}^k \gamma \sigma \boldsymbol{\epsilon}_i^T \boldsymbol{\epsilon}_j^{(i)}|. \tag{93}$$

We first upper bound $\max_j |\sum_{i=1}^k \gamma r_{ij} \boldsymbol{\epsilon}_i^T \boldsymbol{w}^\star| \leq \max_i |\gamma \boldsymbol{\epsilon}_i^T \boldsymbol{w}^\star|$. Since the norm of the signal is 1, $\|\boldsymbol{w}^\star\| = 1$, we have that $\boldsymbol{\epsilon}_i^T \boldsymbol{w}^\star \sim \mathcal{N}(0, 1)$. We can now upper bound $\max_i |\gamma \boldsymbol{\epsilon}_i^T \boldsymbol{w}^\star|$ as,

$$\max_i |\gamma \boldsymbol{\epsilon}_i^T \boldsymbol{w}^\star| \leq \frac{1}{100k^2 d^2} \max_i |\boldsymbol{\epsilon}_i^T \boldsymbol{w}^\star| \leq \frac{1}{8}, \tag{94}$$

with probability $\geq 1 - 10^{-7}$. To derive inequality 94, we have used the concentration inequality, $\mathbb{P}[\max_{i \in [k]} |\boldsymbol{\epsilon}_i^T \boldsymbol{w}^\star| \geq \sqrt{32 \ln(k)}] \leq \frac{2}{k^9} \leq 10^{-7}$.

And now we seek to bound $\max_j |\sum_{i=1}^k \gamma \sigma \boldsymbol{\epsilon}_i^T \boldsymbol{\epsilon}_j^{(i)}|$. By the concentration of the norm of the Gaussian random vector, $\mathbb{P}[\max_i \|\boldsymbol{\epsilon}_i\| \geq \sqrt{d} + 10\sqrt{\ln(k)}] \leq 2k \exp(-\frac{100 \ln(k)}{16}) \leq 2kk^{-6} \leq 10^{-7}$. Now, define $\boldsymbol{u}_i = \frac{\boldsymbol{\epsilon}_i}{\|\boldsymbol{\epsilon}_i\|}$. Therefore,

$$\max_{i \in [k], j \in [m]} |\sum_{i=1}^k \gamma \sigma \boldsymbol{\epsilon}_i^T \boldsymbol{\epsilon}_j^{(i)}| \leq \gamma \sigma(\sqrt{d} + 10\sqrt{\ln(k)}) \max_{j \in [m]} |\sum_{i=1}^k \boldsymbol{u}_i^T \boldsymbol{\epsilon}_j^{(i)}| \tag{95}$$

$$\stackrel{d}{=} \gamma \sigma(\sqrt{d} + 10\sqrt{\ln(k)}) \max_{j \in [m]} |\sqrt{k}\epsilon_j|, \tag{96}$$

where $\epsilon_j \sim \mathcal{N}(0, 1)$. The last equality in distribution follows from the fact the sum of $k$ independent Gaussian random variables is a Gaussian random variable with variance $k$. Now, from the concentration inequality, $\mathbb{P}[\max_{j \in [m]} |\epsilon_j| \geq \sqrt{32 \ln(m)}] \leq \frac{2}{m^9} \leq 10^{-7}$. Substituting this above,

$$\max_{i \in [k], j \in [m]} |\sum_{i=1}^k \gamma \sigma \boldsymbol{\epsilon}_i^T \boldsymbol{\epsilon}_j^{(i)}| \leq \gamma \sigma(\sqrt{d} + 10\sqrt{\ln(k)}) \max_{j \in [m]} |\sqrt{k}\epsilon_j|, \tag{97}$$

$$\leq \frac{3}{2} \gamma \sigma \sqrt{kd} \sqrt{32 \ln(m)} \tag{98}$$

$$\leq \frac{3}{2} \frac{1}{100k^2 d^2} \frac{1}{100\sqrt{k \ln(kd)^3}} \sqrt{kd} \sqrt{32 \ln(m)} \leq \frac{1}{8}, \tag{99}$$

Substituting 94 and 99 into 93, we can now bound $\alpha_j$, for all $j \in [m]$,

$$1 + \frac{1}{8} + \frac{1}{8} \geq \alpha_j \geq 1 - \frac{1}{8} - \frac{1}{8}, \tag{100}$$

$$\frac{5}{4} \geq \alpha_j \geq \frac{3}{4}, \tag{101}$$

We are now in the position to analyze $\tilde{\boldsymbol{w}}_i^1$,

$$\tilde{\boldsymbol{w}}_i^1 = \boldsymbol{w}_i^0 - \eta_1 \nabla_{\boldsymbol{w}_i^0} l([\boldsymbol{w}_i^0, 0]; S_1^m), \tag{102}$$

$$\stackrel{90}{=} \boldsymbol{w}_i^0 + \tfrac{1}{m} \sum_{j=1}^m 2\alpha_j \boldsymbol{\beta}_{ij}, \tag{103}$$

$$= \boldsymbol{w}_i^0 + \tfrac{1}{m} \sum_{j=1}^m 2 y_j \alpha_j \left( y_j r_{ij} \boldsymbol{w}^\star + \sigma \boldsymbol{\epsilon}_j^{(i)} \right) \tag{104}$$

$$\stackrel{d}{=} \boldsymbol{w}_i^0 + \tfrac{2\sum_{j=1}^m r_{ij}\alpha_j}{m} \boldsymbol{w}^\star + \tfrac{2\sigma\sqrt{\sum_{j=1}^m \alpha_j^2}}{m} \bar{\boldsymbol{\epsilon}}_i, \tag{105}$$

where $\bar{\boldsymbol{\epsilon}}_i \sim \mathcal{N}(\boldsymbol{0}, \mathbf{I}_d)$. Similarly to $\boldsymbol{\epsilon}_i$, we can use the concentration of the norm of the Gaussian random vector to show, $\mathbb{P}[\max_i |\|\bar{\boldsymbol{\epsilon}}_i\| - \sqrt{d}| \geq 10\sqrt{\ln(k)}] \leq 2k \exp(-\tfrac{100\ln(k)}{16}) \leq 2kk^{-6} \leq 10^{-7}$. Also note that by using Chernoff and Union bounds, we have $\tfrac{m}{2k} \leq \sum_{j=1}^m r_{ij} \leq \tfrac{3m}{2k}$, with probability $\geq 1 - 2k \exp(\tfrac{-40k\ln(k)}{10k}) \geq 1 - 10^{-7}$, for large enough $k$. Recall that the aim is to bound $(\boldsymbol{w}_i^1)^T \boldsymbol{w}^\star = \tfrac{(\tilde{\boldsymbol{w}}_i^1)^T \boldsymbol{w}^\star}{\|\tilde{\boldsymbol{w}}_i^1\|}$, for all $i \in [k]$. For this, we first prove an upper bound for $\|k\tilde{\boldsymbol{w}}_i^1\|$,

$$\|k\tilde{\boldsymbol{w}}_i^1\| = \|k\boldsymbol{w}_i^0 + \tfrac{2k\sum_{j=1}^m r_{ij}\alpha_j}{m} \boldsymbol{w}^\star + \tfrac{2k\sigma\sqrt{\sum_{j=1}^m \alpha_j^2}}{m} \bar{\boldsymbol{\epsilon}}_i, \|, \tag{106}$$

$$\leq \|k\boldsymbol{w}_i^0\| + \|\tfrac{2k\sum_{j=1}^m r_{ij}\alpha_j}{m} \boldsymbol{w}^\star\| + \|\tfrac{2k\sigma\sqrt{\sum_{j=1}^m \alpha_j^2}}{m} \bar{\boldsymbol{\epsilon}}_i, \|, \tag{107}$$

$$\stackrel{101}{\leq} \|k\boldsymbol{w}_i^0\| + \tfrac{5k}{2} \|\tfrac{\sum_{j=1}^m r_{ij}}{m} \boldsymbol{w}^\star\| + \tfrac{5k\sigma}{2} \sqrt{\tfrac{1}{m}} \|\bar{\boldsymbol{\epsilon}}_i\|, \tag{108}$$

$$\leq \gamma k\|\boldsymbol{\epsilon}_i\| + \tfrac{5k}{2} \|\tfrac{\sum_{j=1}^m r_{ij}}{m} \boldsymbol{w}^\star\| + \tfrac{5k\sigma}{2} \sqrt{\tfrac{1}{m}} \|\bar{\boldsymbol{\epsilon}}_i\|, \tag{109}$$

Now substituting the facts $\|\boldsymbol{\epsilon}_i\|, \|\bar{\boldsymbol{\epsilon}}_i\| \leq \tfrac{3}{2}\sqrt{d}$, and that $\sum_{j=1}^m r_{ij} \leq \tfrac{3m}{2k}$,

$$\|k\tilde{\boldsymbol{w}}_i^1\| \leq \gamma \tfrac{3k\sqrt{d}}{2} + \tfrac{15}{4} + \tfrac{15k\sigma}{4} \sqrt{\tfrac{d}{m}}, \tag{110}$$

$$\leq \tfrac{3\sqrt{d}}{2kd^2} + \tfrac{15}{4} + \tfrac{15\sigma}{4} \sqrt{\tfrac{kd}{\sigma^2(d+k)\ln(kd)}}, \tag{111}$$

$$\leq \tfrac{3}{2kd} + \tfrac{15}{4} + \tfrac{15}{4} \sqrt{\tfrac{kd}{(k+d)\ln(kd)}}, \tag{112}$$

$$\leq 4 \sqrt{\tfrac{kd}{(k+d)\ln(kd)}}. \tag{113}$$

And similarly, we lower bound $\|\tilde{\boldsymbol{w}}^1\|$,

$$\|k\tilde{\boldsymbol{w}}_i^1\| \geq \|\tfrac{2k\sigma\sqrt{\sum_{j=1}^m \alpha_j^2}}{m} \bar{\boldsymbol{\epsilon}}_i\| - \|k\boldsymbol{w}_i^0\| - \|\tfrac{2k\sum_{j=1}^m r_{ij}\alpha_j}{m} \boldsymbol{w}^\star\|, \tag{114}$$

$$\stackrel{101}{\geq} \tfrac{3k\sigma}{2} \sqrt{\tfrac{1}{m}} \|\bar{\boldsymbol{\epsilon}}_i\| - \|k\boldsymbol{w}_i^0\| - \tfrac{5}{2} \|\tfrac{k\sum_{j=1}^m r_{ij}}{m} \boldsymbol{w}^\star\|, \tag{115}$$

$$\geq \tfrac{3}{4} \sqrt{\tfrac{kd}{(k+d)\ln(kd)}} - \tfrac{3}{2kd} - \tfrac{15}{4}, \tag{116}$$

$$\geq \tfrac{1}{2} \sqrt{\tfrac{kd}{(k+d)\ln(kd)}}. \tag{117}$$

Next, we lower bound $k(\tilde{\boldsymbol{w}}_i^1)^T \boldsymbol{w}^\star$,

$$k(\tilde{\boldsymbol{w}}_i^1)^T \boldsymbol{w}^\star = k(\boldsymbol{w}_i^0)^T \boldsymbol{w}^\star + \tfrac{2k\sum_{j=1}^m r_{ij}\alpha_j}{m} (\boldsymbol{w}^\star)^T \boldsymbol{w}^\star + \tfrac{2k\sigma\sqrt{\sum_{j=1}^m \alpha_j^2}}{m} (\bar{\boldsymbol{\epsilon}}_i)^T \boldsymbol{w}^\star, \tag{118}$$

$$= k\gamma \boldsymbol{\epsilon}_i^T \boldsymbol{w}^\star + \tfrac{2k\sum_{j=1}^m r_{ij}\alpha_j}{m} + \tfrac{2k\sigma\sqrt{\sum_{j=1}^m \alpha_j^2}}{m} (\bar{\boldsymbol{\epsilon}}_i)^T \boldsymbol{w}^\star, \tag{119}$$

$$\stackrel{101}{\geq} -|k\gamma \boldsymbol{\epsilon}_i^T \boldsymbol{w}^\star| + \tfrac{3k}{2} \tfrac{\sum_{j=1}^m r_{ij}}{m} - \tfrac{5\sigma k}{2\sqrt{m}} |(\bar{\boldsymbol{\epsilon}}_i)^T \boldsymbol{w}^\star|, \tag{120}$$

$$\geq -\tfrac{1}{100kd^2} |\boldsymbol{\epsilon}_i^T \boldsymbol{w}^\star| + \tfrac{3}{4} - \tfrac{5\sigma k}{2\sqrt{k\sigma^2(k+d)\ln(kd)}} |(\bar{\boldsymbol{\epsilon}}_i)^T \boldsymbol{w}^\star|, \tag{121}$$

$$\geq -\tfrac{1}{100kd^2} |\boldsymbol{\epsilon}_i^T \boldsymbol{w}^\star| + \tfrac{3}{4} - \tfrac{5}{2\sqrt{\ln(kd)}} |(\bar{\boldsymbol{\epsilon}}_i)^T \boldsymbol{w}^\star|, \tag{122}$$

$$\geq -\tfrac{1}{8} + \tfrac{3}{4} - \tfrac{1}{8} \geq \tfrac{1}{2}, \tag{123}$$

where the last inequality follows from the bounds, $\mathbb{P}[\max_{i\in[k]}\frac{|\epsilon_i^T w^\star|}{100kd^2}\geq\frac{\sqrt{32\ln(k)}}{100kd^2}]\leq\frac{2}{k^9}\leq 10^{-7}$, and $\mathbb{P}[\max_{i\in[k]}\frac{|5\tilde{\epsilon}_i^T w^\star|}{2\sqrt{\ln(kd)}}\geq\frac{5\sqrt{32\ln(k)}}{2\sqrt{\ln(kd)}}]\leq\frac{2}{k^9}\leq 10^{-7}$. And similarly we upper bound $(\tilde{w}_i^1)^T w^\star$,

$$k(\tilde{w}_i^1)^T w^\star \leq \tfrac{1}{8}+\tfrac{3}{4}+\tfrac{1}{8}\leq 1. \tag{124}$$

Therefore, $2\sqrt{\frac{(k+d)\ln(kd)}{kd}}\geq(w_i^1)^T w^\star\geq\frac{1}{8}\sqrt{\frac{(k+d)\ln(kd)}{kd}}$, with probability $\geq 1-10^{-6}$. We can now express $w_i^1$ as $\lambda_i w^\star+\sqrt{1-\lambda_i^2}w_\perp^\star$, where $2\sqrt{\frac{(k+d)\ln(kd)}{kd}}\geq\lambda_i\geq\frac{1}{8}\sqrt{\frac{(k+d)\ln(kd)}{kd}}$, $\|w_\perp^\star\|=1$, and $(w^\star)^T w_\perp^\star=0$.

3b. Update Step 2

We will now show that the $\frac{1}{8}\sqrt{\frac{(k+d)\ln(kd)}{kd}}$ alignment with the signal vector enables us to filter out the "noise" patches from the "signal" patch. This de-noising effect allows us to achieve a stronger constant alignment of each parameter vector with the signal vector.

We begin by analyzing the push forward of all the noise in the dataset $S_2^m$ through the LCN model,

$$\max_{i\in[k],j\in[m]}|\sigma(w_i^1)^T\epsilon_j^{(i)}|\leq\frac{1}{100\sqrt{k\ln(kd)^3}}\max_{i,j}|(w_i^1)^T\epsilon_j^{(i)}|, \tag{125}$$

$$\leq\frac{1}{100\sqrt{k\ln(kd)^3}}\sqrt{32\ln(\sigma^2 k^2(k+d)\ln(kd))}, \tag{126}$$

$$\leq\frac{1}{4\sqrt{k}}\leq\frac{1}{32}\sqrt{\frac{(k+d)\ln(kd)}{kd}} \tag{127}$$

For inequality 126, we have used the concentration of the maximum of the absolute value of $mk$ i.i.d. Gaussian random variables, $\mathbb{P}[\max_{i,j}|(w_i^1)^T\epsilon_j^{(i)}|\geq\sqrt{32\ln(mk)}]\leq\frac{2}{(mk)^9}\leq 10^{-7}$. Recall from the analysis of the first update step that,

$$2\sqrt{\frac{(k+d)\ln(kd)}{kd}}\geq(w_i^1)^T w^\star\geq\frac{1}{8}\sqrt{\frac{(k+d)\ln(kd)}{kd}} \tag{128}$$

From 127, 128, for all $j\in[m]$, and $i\in[k]$, we have

$$(2+\tfrac{1}{32})\sqrt{\frac{(k+d)\ln(kd)}{kd}}\geq y_j(w_i^1)^T x_j^{(i)}\geq(\tfrac{1}{8}-\tfrac{1}{32})\sqrt{\frac{(k+d)\ln(kd)}{kd}} \text{ where, } r_{ij}=1, \tag{129}$$

$$\tfrac{1}{32}\sqrt{\frac{(k+d)\ln(kd)}{kd}}\geq y_j(w^1)^T x_j^{(i)}\geq-\tfrac{1}{32}\sqrt{\frac{(k+d)\ln(kd)}{kd}}, \text{ where, } r_{ij}=0. \tag{130}$$

Therefore, with $b_1=\frac{1}{32}\sqrt{\frac{(k+d)\ln(kd)}{kd}}$, we filter out all the noise and let the signal pass through,

$$2\sqrt{\frac{(k+d)\ln(kd)}{kd}}\geq\phi_{b_1}(y_j(w_i^1)^T x_j^{(i)})\geq(\tfrac{1}{8}-\tfrac{1}{16})\sqrt{\frac{(k+d)\ln(kd)}{kd}}, \text{ where, } r_{ij}=1, \tag{131}$$

$$\phi_{b_1}(y_j(w_i^1)^T x_j^{(i)})=0, \text{ where, } r_{ij}=0. \tag{132}$$

We will now follow in the footsteps of our analysis of update step 1. We seek to prove high probability upper and lower bounds for $(w_i^2)^T w^\star$. We define $\tilde{w}_i^2=w_i^1-\eta_2\nabla_{w_i^1}l(w_i^1,b_1;S_2^m)$, and therefore $(w_i^2)^T w^\star=\frac{(\tilde{w}_i^2)^T w^\star}{\|\tilde{w}_i^2\|}$. We first evaluate the gradient of the empirical loss function with respect to $w$,

$$\nabla_{w_i^1}l([w_i^1,b_1];S_2^m)=\frac{-1}{m}\sum_{j=1}^m 2\left(y_j-\sum_{i=1}^k\phi_{b_1}((w_i^1)^T x_j^{(i)})\right)\left(x_j^{(i)}\phi_{b_1}'((w_i^1)^T x_j^{(i)})\right), \tag{133}$$

$$=\frac{-2}{m}\sum_{j=1}^m\left(1-\sum_{i=1}^k\phi_{b_1}(y_j(w_i^1)^T x_j^{(i)})\right)\left(y_j r_{ij}x_j^{(i)}\right), \tag{134}$$

where the last equality follows from 131, and 132. Now, for large enough $k,d$,

$$1\geq 1-\tfrac{1}{8}\sqrt{\frac{(k+d)\ln(kd)}{kd}}\geq\alpha_j:=1-\sum_{i=1}^k\phi_{b_1}(y_j(w_i^1)^T x_j^{(i)})\geq 1-2\sqrt{\frac{(k+d)\ln(kd)}{kd}}. \tag{135}$$

Substituting the definition of $\alpha_j$ and simplifying 134,

$$\nabla_{\boldsymbol{w}_i^1} l([\boldsymbol{w}_i^1, b_1]; S_2^m) = \frac{-1}{m} \sum_{j=1}^m 2\alpha_j y_j r_{ij} \boldsymbol{x}_j^{(i)} \tag{136}$$

$$\stackrel{d}{=} \frac{-1}{m} \sum_{j=1}^m 2r_{ij}\alpha_j \left( \boldsymbol{w}^\star + \sigma \boldsymbol{\epsilon}_j^{(i)} \right), \tag{137}$$

$$\stackrel{d}{=} -\sum_{j=1}^m \frac{2r_{ij}\alpha_j}{m} \boldsymbol{w}^\star - \frac{2\sigma \sqrt{\sum_{j=1}^m r_{ij}^2 \alpha_j^2}}{m} \hat{\boldsymbol{\epsilon}}_i, \tag{138}$$

where $\hat{\boldsymbol{\epsilon}}_i \sim \mathcal{N}(\boldsymbol{0}, \mathbf{I}_d)$. By Chernoff and Union bounds, $\frac{m}{2k} \leq r_i := \sum_{j=m+1}^n \frac{kr_{ij}}{m} \leq \frac{3m}{2k}$, with probability $\geq 1 - 10^{-7}$. Also, we note the concentration of the norm of the Gaussian random vector to show, $\mathbb{P}[\max_i |\|\hat{\boldsymbol{\epsilon}}_i\| - \sqrt{d}| \geq 10\sqrt{\ln(k)}] \leq 2k \exp(-\frac{100 \ln(k)}{16}) \leq 2kk^{-6} \leq 10^{-7}$.

We are now ready to bound $(\boldsymbol{w}_i^2)^T \boldsymbol{w}^\star = \frac{(\tilde{\boldsymbol{w}}_i^2)^T \boldsymbol{w}^\star}{\|\tilde{\boldsymbol{w}}_i^2\|}$. We denote $a_i = \frac{k}{m} \sum_{j=1}^m \alpha_j r_{ij} \geq \frac{1}{3}$.

$$\|\tilde{\boldsymbol{w}}_i^2\| = \|\boldsymbol{w}_i^1 + \eta_2 \sum_{j=1}^m \frac{2r_{ij}\alpha_j}{m} \boldsymbol{w}^\star + \eta_2 \frac{2\sigma \sqrt{\sum_{j=1}^m r_{ij}^2 \alpha_j^2}}{m} \hat{\boldsymbol{\epsilon}}_i\|,$$

$$\leq 1 + \eta_2 \sum_{j=1}^m \|\frac{2r_{ij}\alpha_j}{m} \boldsymbol{w}^\star\| + \eta_2 \|\frac{2\sigma \sqrt{\sum_{j=1}^m r_{ij}^2 \alpha_j^2}}{m} \hat{\boldsymbol{\epsilon}}_i\|,$$

$$\leq 1 + \frac{2a_i \eta_2}{k} \|\boldsymbol{w}^\star\| + \frac{\sqrt{6}\sigma \eta_2}{\sqrt{mk}} \|\hat{\boldsymbol{\epsilon}}_i\|,$$

$$\leq 1 + 10^3 \left( 2a_i + \frac{\sqrt{6k}\sigma}{\sqrt{m}} \|\hat{\boldsymbol{\epsilon}}_i\| \right)$$

$$\leq 1 + 10^3 \left( 2a_i + \frac{\sqrt{6k}\sigma}{\sqrt{\sigma^2 k(k+d) \ln(k+d)}} \frac{3\sqrt{d}}{2} \right)$$

$$\leq 1 + 10^3 \left( 2a_i + 10^{-3,} \right)$$

for a large enough $k, d$. And the lower bound on $(\tilde{\boldsymbol{w}}_i^2)^T \boldsymbol{w}^\star$ is given by,

$$(\tilde{\boldsymbol{w}}_i^2)^T \boldsymbol{w}^\star = (\boldsymbol{w}_i^1)^T \boldsymbol{w}^\star + \eta_2 \sum_{j=1}^m \frac{2r_{ij}\alpha_j}{m} (\boldsymbol{w}^\star)^T \boldsymbol{w}^\star + \eta_2 \frac{2\sigma \sqrt{\sum_{j=1}^m r_{ij}^2 \alpha_j^2}}{m} \hat{\boldsymbol{\epsilon}}_i^T \boldsymbol{w}^\star, \tag{139}$$

$$\stackrel{d}{=} (\boldsymbol{w}_i^1)^T \boldsymbol{w}^\star + \eta_2 \sum_{j=1}^m \frac{2r_{ij}\alpha_j}{m} + \eta_2 \frac{2\sigma \sqrt{\sum_{j=1}^m r_{ij}^2 \alpha_j^2}}{m} \epsilon; \ \epsilon \sim \mathcal{N}(0, 1), \tag{140}$$

$$\geq -1 + 2a_i \times 10^3 - \frac{\sqrt{6k}\sigma}{\sqrt{\sigma^2 k(k+d) \ln(k+d)}} \epsilon \times 10^3 \tag{141}$$

$$\geq -1 + 10^3 (2a_i - 10^{-3}), \tag{142}$$

where we have used $\mathbb{P}[|\frac{\sqrt{6k}\sigma \epsilon}{\sqrt{\sigma^2 k(k+d) \ln(k+d)}}| \geq 10^{-3}]$, with probability $\leq 10^{-7}$. Therefore,

$$\frac{(\tilde{\boldsymbol{w}}_i^2)^T \boldsymbol{w}^\star}{\|\tilde{\boldsymbol{w}}_i^2\|} \geq \frac{-1 + 10^3 (2a_i - 10^{-3})}{1 + 10^3 (2a_i + 10^3)} \geq 0.96, \tag{143}$$

and this occurs with a probability $\geq 1 - 2 \times 10^{-6}$.

### 3c. LCN has Low Risk

We now show that this large constant alignment guarantees a low risk. We bound the push forward of the noise through the LCN,

$$|\sigma \max_{j \in [m]} ((\boldsymbol{w}_i^2)^T \boldsymbol{\epsilon}_i)| \leq \frac{6\sqrt{\ln(k)}}{100\sqrt{k \ln(kd)^3}} \leq 10^{-4} \tag{144}$$

To derive the last inequality, we have used the concentration of the maximum of the absolute value of $k$ i.i.d. Gaussian random variables, $\mathbb{P}[\max_{j \in [k]} |(\boldsymbol{w}_i^2)^T \boldsymbol{\epsilon}_j| \geq \sqrt{32 \ln(k)}] \leq \frac{2}{m^9} \leq 10^{-6}$. For this

data sample, let $t \in [k]$ be the index of the signal patch, then,

$$\phi_{b_2}(y_j(\boldsymbol{w}_i^2)^T \boldsymbol{x}_j^{(t)}) \geq 0.959, \tag{145}$$

$$\phi_{b_2}(y_j(\boldsymbol{w}_i^2)^T \boldsymbol{x}_j^{(i)}) = 0, \ \ \forall\, i \neq t. \tag{146}$$

We note that with probability $1 - 3 \times 10^{-6}$, the risk of the classifier less than $(1 - 0.959)^2$. To bound the risk in the failure case, we note that for any $\boldsymbol{v} \in \mathcal{W}$,

$$\mathbb{E}[(y - \sum_{i=1}^{k} \phi_b(\boldsymbol{w}_i^T \boldsymbol{x}^{(i)}))^2] = \mathbb{E}[(1 - \sum_{i=1}^{k} \phi_b(y\boldsymbol{w}_i^T \boldsymbol{x}^{(i)}))^2], \tag{147}$$

$$= \frac{1}{k} \sum_{j=1}^{k} \mathbb{E}_{(\boldsymbol{x},y)\sim\mathrm{SSD}_j} [(1 - \sum_{i=1}^{k} \phi_b(y\boldsymbol{w}_i^T \boldsymbol{x}^{(i)}))^2], \tag{148}$$

$$= \frac{1}{k} \sum_{j=1}^{k} \mathbb{E}[(1 - \sum_{i\neq j} \phi_b(\sigma\epsilon_{ij}) - \phi_b(\cos(\alpha_j) + \sigma\epsilon_{jj}))^2], \tag{149}$$

To evaluate the above expression, we observe that the expectation can be written as,

$$\mathbb{E}[(1 - \sum_{i\neq j} \phi_b(\sigma\epsilon_{ij}) - \phi_b(\cos(\alpha_j) + \sigma\epsilon_{jj}))^2] = \mathrm{Var}[(1 - \sum_{i\neq j} \phi_b(\sigma\epsilon_{ij}) - \phi_b(\cos(\alpha_j) + \sigma\epsilon_{jj}))]$$
$$+ \mathbb{E}[(1 - \sum_{i\neq j} \phi_b(\sigma\epsilon_{ij}) - \phi_b(\cos(\alpha_j) + \sigma\epsilon_{jj}))]^2, \tag{150}$$

$$= \sum_{i\neq j} \mathrm{Var}[\phi_b(\sigma\epsilon_{ij})] + \mathrm{Var}[\phi_b(\cos(\alpha_j) + \sigma\epsilon_{jj})] + (\mathbb{E}[\phi_b(\cos(\alpha_j) + \sigma\epsilon_{jj})])^2, \tag{151}$$

$$\leq \sum_{i} \sigma^2 + \cos^2(\alpha_j) \leq k\sigma^2 + 1 \leq 2. \tag{152}$$

Substituting this back,

$$\mathbb{E}[(y - \sum_{i=1}^{k} \phi_b(\boldsymbol{w}_i^T \boldsymbol{x}^{(i)}))^2] \leq \frac{1}{k} \sum_{j=1}^{k} 2 = 2 \tag{153}$$

Therefore, the expected risk of the trained LCN is upper bounded as,

$$\mathbb{E}\left[R\left(\bar{\theta}_n, P\right)\right] \leq (1 - 0.959)^2 + 6 \times 10^{-6} \leq \delta \tag{154}$$

$$\square$$

# D  LCN VS CNN SEPARATION RESULTS

## D.1  LCN LOWER BOUND

The following lemma provides a lower bound on the risk of each function in the class of LCNs over the set of transformations of $SSD_1$, made using the group $\mathcal{U}$. The group allows for orthogonal transformations within the patches, and does not allow patches to permute.

**Lemma D.1.** *Let $\mathcal{F}$ denote the class of functions represented by the set of locally connected neural network models, $\mathcal{M}_{\mathcal{L}}[\mathcal{W}]$, as defined in 5. We define $\mathcal{U} := \{Block\,(\mathbf{U}_1, \ldots, \mathbf{U}_k) \mid \mathbf{U}_i \in \mathcal{O}(d)\}$. Let $\mathcal{P}$ be the set of distributions $\{\mathbf{U} \circ SSD_1 \mid \mathbf{U} \in \mathcal{U}\}$. We define the target function $\theta^\star \colon \mathcal{P} \to \mathcal{F}$ as, $\theta^\star(\mathbf{U} \circ SSD_1) = \mathcal{M}[[\mathbf{U}^{(1)}\boldsymbol{w}^\star, .., \mathbf{U}^{(k)}\boldsymbol{w}^\star, b^\star]]$, where $\boldsymbol{w}^\star$ is the signal vector, and $b^\star$ is some fixed value in $(0,1)$ [3]. Let $\mathcal{F}_{\mathcal{P}}$ be the codomain of $\theta^\star$. Consider $\rho \colon (\mathcal{F}, \mathcal{F}_{\mathcal{P}}) \to \mathcal{R}$,*

$$\rho(f, \theta^\star(\mathbf{U} \circ SSD_1)) = \left(1 - \max(0, \|\boldsymbol{w}_1\|\cos(\alpha_1))\right)^2, \tag{155}$$

*where $\cos(\alpha_1) = \frac{\boldsymbol{w}_1^T \mathbf{U}^{(1)}\boldsymbol{w}^\star}{\|\boldsymbol{w}_1\|}$. Then, the risk of $f \in F$, on $\mathbf{U} \circ SSD_1 \in \mathcal{P}$ satisfies,*

$$R\left(f, \mathbf{U} \circ SSD_1\right) \geq \rho(f, \theta^\star(\mathbf{U} \circ SSD_1)). \tag{156}$$

*Proof.* Observe that,

$$R\left(f, \mathbf{U} \circ P\right) = \mathop{\mathbb{E}}_{(\boldsymbol{x},y) \sim \mathbf{U} \circ P}\left[(y - f(\boldsymbol{x}))^2\right], \tag{157}$$

$$= \mathop{\mathbb{E}}_{(\boldsymbol{x},y) \sim \mathbf{U} \circ P}\left[\left(y - \frac{1}{k}\sum_{i=1}^{k}\phi_b(\boldsymbol{w}_i^T \boldsymbol{x})\right)^2\right], \tag{158}$$

$$= \mathop{\mathbb{E}}_{\boldsymbol{\epsilon}, \boldsymbol{x} = \mathbf{U}\boldsymbol{w}^\star + \sigma\boldsymbol{\epsilon}}\left[\left(1 - \sum_{i=1}^{k}\phi_b(\boldsymbol{w}_i^T \boldsymbol{x})\right)^2\right], \tag{159}$$

$$\overset{\text{Jensens'}}{\geq} \left(\mathop{\mathbb{E}}_{\boldsymbol{\epsilon}, \boldsymbol{x}}\left[1 - \sum_{i=1}^{k}\phi_b(\boldsymbol{w}_i^T \boldsymbol{x})\right]\right)^2, \tag{160}$$

$$= \left(1 - \sum_{i\neq 1}^{k}\mathop{\mathbb{E}}_{\epsilon}[\phi_b(\|\boldsymbol{w}_i\|\sigma\epsilon)] - \mathop{\mathbb{E}}_{\epsilon}[\phi_b(\|\boldsymbol{w}_1\|\cos(\alpha_t) + \|\boldsymbol{w}_1\|\sigma\epsilon)]\right)^2, \tag{161}$$

$$= \left(1 - \mathop{\mathbb{E}}_{\epsilon}[\phi_b(\|\boldsymbol{w}_1\|\cos(\alpha_t) + \|\boldsymbol{w}_1\|\sigma\epsilon)]\right)^2, \tag{162}$$

where in 162, we used the fact that since $\phi_b(-x) = -\phi_b(x)$, and therefore $\mathbb{E}[\phi_b(\|\boldsymbol{w}_i\|\sigma\epsilon)] = 0$, for all $i \neq 1$. For brevity, we define $\bar{\mu} = \|\boldsymbol{w}_1\|\cos(\alpha_1)$, and $\bar{\sigma} = \|\boldsymbol{w}_1\|\sigma$. Then observe that,

$$\mathop{\mathbb{E}}_{\epsilon}[\phi_b(\bar{\mu} + \bar{\sigma}\epsilon)] = \mathop{\mathbb{E}}_{\epsilon}[\max(0, \bar{\mu} - b + \bar{\sigma}\epsilon)] - \mathop{\mathbb{E}}_{\epsilon}[\max(0, -\bar{\mu} - b - \bar{\sigma}\epsilon)], \tag{163}$$

$$= \mathop{\mathbb{E}}_{\epsilon}[\max(0, \bar{\mu} - b + \bar{\sigma}\epsilon)] - \mathop{\mathbb{E}}_{\epsilon}[\max(0, -\bar{\mu} - b + \bar{\sigma}\epsilon)]. \tag{164}$$

We begin by evaluating $\mathbb{E}_{\epsilon}[\max(0, \bar{\mu} - b + \bar{\sigma}\epsilon)]$,

$$\mathop{\mathbb{E}}_{\epsilon}[\max(0, \bar{\mu} - b + \bar{\sigma}\epsilon)] = \tfrac{1}{2}\left(\mathop{\mathbb{E}}_{\epsilon}[\bar{\mu} - b + \bar{\sigma}\epsilon] + \mathop{\mathbb{E}}_{\epsilon}[|\bar{\mu} - b + \bar{\sigma}\epsilon|]\right), \tag{165}$$

$$= \tfrac{1}{2}(\bar{\mu} - b) + \tfrac{1}{2}\left(\bar{\sigma}\sqrt{\tfrac{2}{\pi}}\exp\left(-\tfrac{(\bar{\mu}-b)^2}{2\bar{\sigma}^2}\right) + \right.$$
$$\left. (\bar{\mu} - b)\left(1 - 2\Phi\left(-\tfrac{\bar{\mu}-b}{\bar{\sigma}}\right)\right)\right), \tag{166}$$

$$= (\bar{\mu} - b)\left(1 - \Phi\left(-\tfrac{\bar{\mu}-b}{\bar{\sigma}}\right)\right) + \eta_1, \tag{167}$$

---

[3]The claim and the proof do not depend on the chosen value of $b^\star$

where $\eta_1 = \bar{\sigma}\sqrt{\frac{1}{2\pi}}\exp\left(-\frac{(\bar{\mu}-b)^2}{2\bar{\sigma}^2}\right)$. Similarly, for the second term, we have,

$$\mathbb{E}_{\epsilon}[\max(0, -\bar{\mu} - b + \bar{\sigma}\epsilon)] = (-\bar{\mu} - b)\left(1 - \Phi\left(\frac{\bar{\mu}+b}{\bar{\sigma}}\right)\right) + \eta_2, \tag{168}$$

where $\eta_2 = \bar{\sigma}\sqrt{\frac{1}{2\pi}}\exp\left(-\frac{(\bar{\mu}+b)^2}{2\bar{\sigma}^2}\right)$. Substituting these results back to 164,

$$\mathbb{E}_{\epsilon}[\phi_b(\bar{\mu} + \bar{\sigma}\epsilon)] = (\bar{\mu} - b)\left(1 - \Phi\left(-\frac{\bar{\mu}-b}{\bar{\sigma}}\right)\right) + \eta_1 + (\bar{\mu} + b)\left(1 - \Phi\left(\frac{\bar{\mu}+b}{\bar{\sigma}}\right)\right) - \eta_2. \tag{169}$$

Now observe that,

$$\frac{\partial}{\partial b}\mathbb{E}_{\epsilon}[\phi_b(\bar{\mu} + \bar{\sigma}\epsilon)] = -\left(1 - \Phi\left(-\frac{\bar{\mu}-b}{\bar{\sigma}}\right)\right) - \frac{\bar{\mu}-b}{\bar{\sigma}}\left(\Phi'\left(-\frac{\bar{\mu}-b}{\bar{\sigma}}\right)\right) + \frac{\bar{\mu}-b}{\sqrt{2\pi}\bar{\sigma}}\exp\left(-\frac{(\bar{\mu}-b)^2}{2\bar{\sigma}^2}\right)$$
$$+ \left(1 - \Phi\left(\frac{\bar{\mu}+b}{\bar{\sigma}}\right)\right) - \frac{(\bar{\mu}+b)}{\bar{\sigma}}\left(\Phi'\left(\frac{\bar{\mu}+b}{\bar{\sigma}}\right)\right) + \frac{(\bar{\mu}+b)}{\sqrt{2\pi}\bar{\sigma}}\exp\left(-\frac{(\bar{\mu}+b)^2}{2\bar{\sigma}^2}\right). \tag{170}$$

Substituting the expression for $\Phi'$,

$$\frac{\partial}{\partial b}\mathbb{E}_{\epsilon}[\phi_b(\bar{\mu} + \bar{\sigma}\epsilon)] = \left(1 - \Phi\left(\frac{\bar{\mu}+b}{\bar{\sigma}}\right)\right) - \left(1 - \Phi\left(-\frac{\bar{\mu}-b}{\bar{\sigma}}\right)\right), \tag{171}$$

$$= \Phi\left(\frac{b-\bar{\mu}}{\bar{\sigma}}\right) - \Phi\left(\frac{\bar{\mu}+b}{\bar{\sigma}}\right). \tag{172}$$

If $\bar{\mu} > 0$, then the gradient with respect to $b$ is always negative when $b > 0$, therefore the maxima of $\mathbb{E}_{\epsilon}[\phi_b(\bar{\mu} + \bar{\sigma}\epsilon)]$ occurs at $b = 0$, with the maxima being $\bar{\mu} \leq 1$. If $\bar{\mu} < 0$, then the gradient is always positive when $b > 0$, therefore the maxima of $\mathbb{E}_{\epsilon}[\phi_b(\bar{\mu} + \bar{\sigma}\epsilon)]$ occurs at $b = +\infty$, with the maxima being $0 \leq 1$. And finally, for $\bar{\mu} = 0$, $\mathbb{E}_{\epsilon}[\phi_b(\bar{\mu} + \bar{\sigma}\epsilon)] = 0 < 1$, by symmetry. Using these observations with 162 proves the result,

$$R(f, \mathbf{U} \circ P) \geq (1 - \max(\|\boldsymbol{w}_1\|\cos(\alpha_1), 0))^2 \tag{173}$$

$\square$

The next lemma establishes that the lower bound on the risk, as defined in Lemma D.1, meets the relaxed conditions of our variant of Fano's Theorem 5.1.

**Lemma D.2.** *Under the notation established in the statement of Lemma D.1, we define the set $\tilde{\mathcal{U}} \subseteq \mathcal{U}$, such that for all $\mathbf{U} \neq \mathbf{V} \in \tilde{\mathcal{U}}$, $(\mathbf{U}^{(1)}\boldsymbol{w}^\star)^T(\mathbf{V}^{(1)}\boldsymbol{w}^\star) < 10^{-3}$, and for all $\mathbf{U} \in \tilde{\mathcal{U}}$, $t \in \{2, \ldots, k\}$, $\mathbf{U}^{(t)}\boldsymbol{w}^\star = e_{dt}$. Then, for all $\mathbf{U} \neq \mathbf{V} \in \tilde{\mathcal{U}}$,*

$$\rho(f, \theta^\star(\mathbf{U} \circ P)) < 10^{-2} \implies \rho(f, \theta^\star(\mathbf{V} \circ P)) > 10^{-2}, \tag{174}$$

*Proof.* Let $\cos(\alpha_1) = \frac{\boldsymbol{w}_1^T\mathbf{U}^{(1)}\boldsymbol{w}^\star}{\|\boldsymbol{w}_1\|}$, and $\cos(\beta_1) = \frac{\boldsymbol{w}_1^T\mathbf{V}^{(1)}\boldsymbol{w}^\star}{\|\boldsymbol{w}_1\|}$. Now,

$$\rho(f, \theta^\star(\mathbf{U} \circ P)) < 10^{-2} \iff (1 - \max(0, \|\boldsymbol{w}_1\|\cos(\alpha_1)))^2 < 10^{-2}, \tag{175}$$
$$\iff \max(0, \|\boldsymbol{w}_1\|\cos(\alpha_1)) > 0.9. \tag{176}$$

By the triangle inequality, we can get an upper bound on $\cos(\beta_1)$ as,

$$\sqrt{2(1 - \cos(\beta_1))} \geq \sqrt{2(1 - 0.001)} - \sqrt{2(1 - \cos(\alpha_1))}, \tag{177}$$
$$\cos(\beta_i) \leq 1 - (\sqrt{(1 - 0.001)} - \sqrt{(1 - 0.9)})^2 \leq 0.7. \tag{178}$$

Therefore, $\max(0, \|\boldsymbol{w}_t\|\cos(\beta_t)) \leq 0.7$, which implies that

$$(1 - \max(\|\boldsymbol{w}_t\|\cos(\beta_t), 0))^2 \geq (0.3)^2 > 10^{-2}. \tag{179}$$

$\square$

In the following lemma, we prove a sample complexity lower bound of $\Omega(\sigma^2 kd)$ for FCNs on the sub-problem $SSD_1$ of DSD.

**Lemma D.3.** *Let $\mathcal{F}$ denote the class of functions represented by the set of locally connected neural network models, $\mathcal{M}_{\mathcal{L}}[\mathcal{W}]$, as defined in 5. Let $S^n \sim (SSD_1)^n$ be the $n$ i.i.d. data samples drawn from $SSD_1$. Consider the group $\tilde{\mathcal{U}} \subseteq \mathcal{O}(kd)$, such that for all $\mathbf{U} \neq \mathbf{V} \in \tilde{\mathcal{U}}$, $(\mathbf{U}^{(1)}\boldsymbol{w}^{\star})^T(\mathbf{V}^{(1)}\boldsymbol{w}^{\star}) < 10^{-3}$, and for all $\mathbf{U} \in \tilde{\mathcal{U}}$, $t \in \{2, \ldots, k\}$, $\mathbf{U}^{(t)}\boldsymbol{w}^{\star} = e_{dt}$. Let $\xi \in \Xi$ encapsulate the randomization, and let $\xi \sim P_{\Xi}$. If $\bar{\theta}(S^n, \xi)$ is a $\tilde{\mathcal{U}}$-equivariant algorithm then, for large enough $k, d$,*

$$n_{\delta}(\bar{\theta}_n) = \Omega\left(\sigma^2 d\right), \tag{180}$$

*where $\delta = 0.5 \times 10^{-2}$.*

*Proof.* We refer to the distribution $SSD_1$ by $P$. Since the algorithm $\bar{\theta}_n$ is $\mathcal{U}$- equivariant, lemma 5.1 gives us that for all $\mathbf{U} \in \tilde{\mathcal{U}}$,

$$\bar{\theta}(\{\boldsymbol{x}_i, y_i\}_n)(\boldsymbol{x}) \overset{d}{=} \bar{\theta}(\{\mathbf{U}\boldsymbol{x}_i, y_i\}_n)(\mathbf{U}\boldsymbol{x}), \tag{181}$$

$$\text{err}\left(\bar{\theta}(\{\boldsymbol{x}_i, y_i\}_n)(\boldsymbol{x}), y\right) \overset{d}{=} \text{err}\left(\bar{\theta}(\{\mathbf{U}\boldsymbol{x}_i, y_i\}_n)(\mathbf{U}\boldsymbol{x}), y\right), \tag{182}$$

$$\underset{S^n \sim P^n}{\mathbb{E}} \underset{(\boldsymbol{x}, y) \sim P}{\mathbb{E}} \text{err}\left(\bar{\theta}(\{\boldsymbol{x}_i, y_i\}_n)(\boldsymbol{x}), y\right) = \underset{S^n \sim P^n}{\mathbb{E}} \underset{(\boldsymbol{x}, y) \sim P}{\mathbb{E}} \text{err}\left(\bar{\theta}(\{\mathbf{U}\boldsymbol{x}_i, y_i\}_n)(\mathbf{U}\boldsymbol{x}), y\right), \tag{183}$$

$$\underset{S^n \sim P^n}{\mathbb{E}} \underset{(\boldsymbol{x}, y) \sim P}{\mathbb{E}} \left[\text{err}\left(\bar{\theta}_n(\boldsymbol{x}), y\right)\right] = \underset{S^n \sim (\mathbf{U} \circ P)^n}{\mathbb{E}} \underset{(\boldsymbol{x}, y) \sim \mathbf{U} \circ P}{\mathbb{E}} \left[\text{err}\left(\bar{\theta}_n(\boldsymbol{x}), y\right)\right] \tag{184}$$

$$\underset{S^n \sim P^n}{\mathbb{E}} \left[R\left(\bar{\theta}_n, P\right)\right] = \underset{S^n \sim (\mathbf{U} \circ P)^n}{\mathbb{E}} \left[R\left(\bar{\theta}_n, \mathbf{U} \circ P\right)\right]. \tag{185}$$

Taking $\sup$ on the right-hand side,

$$\underset{S^n \sim P^n}{\mathbb{E}} \left[R\left(\bar{\theta}_n, P\right)\right] = \sup_{\mathbf{U} \circ P \in \tilde{\mathcal{U}} \circ P} \mathbb{E}\left[R\left(\bar{\theta}_n, \mathbf{U} \circ P\right)\right], \tag{186}$$

An application of corollary A.1.1 gives the bound $\ln(|\bar{\mathcal{U}}|) \geq 0.99d$.

In order to apply our variant of Fano's Theorem 5.1, we set the following variables: $\mathcal{P} = \tilde{\mathcal{U}} \circ P$; $\mathcal{P}_{\mathcal{V}} = \tilde{\mathcal{U}} \circ P$; $\mathcal{F}$, $\Xi$, and $P_{\Xi}$ are already defined in the lemma; $\Theta = \{\theta \mid \theta : ((\mathcal{X}, \mathcal{Y})^n, \Xi) \to \mathcal{F}\}$; $\theta^{\star}(\mathbf{U} \circ P) = \mathcal{M}[[\mathbf{U}^{(1)}\boldsymbol{w}^{\star}, .., \mathbf{U}^{(k)}\boldsymbol{w}^{\star}, b^{\star}]]$, where $\mathbf{U} \in \tilde{\mathcal{U}}$, $b^{\star}$ is some fixed value in $(0, 1)^4$; and $\rho(f, \theta^{\star}(\mathbf{U} \circ P)) = (1 - \max(0, \|\boldsymbol{w}_1\| \cos(\alpha_1)))^2$, where $\mathbf{U} \in \tilde{\mathcal{U}}$, and $\cos(\alpha_1) = \frac{\boldsymbol{w}_1^T \mathbf{U}^{(1)} \boldsymbol{w}^{\star}}{\|\boldsymbol{w}_1\|}$. Recall from Lemma B.1 that $\text{KL}(\mathbf{U} \circ P \parallel \mathbf{V} \circ P) \leq \frac{0.999}{\sigma^2} < \frac{1}{\sigma^2}$.

We are now ready to apply Fano's Theorem 5.1, using the results from Lemmas D.1,D.2, and 186,

$$\underset{S^n \sim P^n}{\mathbb{E}} \left[R\left(\bar{\theta}_n, P\right)\right] \geq \sup_{\mathbf{U} \circ P \in \tilde{\mathcal{U}} \circ P} \mathbb{E}\left[R\left(\bar{\theta}_n, \mathbf{U} \circ P\right)\right], \tag{187}$$

$$\geq \inf_{\theta \in \Theta} \sup_{\mathbf{U} \circ P \in \tilde{\mathcal{U}} \circ P} \mathbb{E}\left[R\left(\theta_n, \mathbf{U} \circ P\right)\right], \tag{188}$$

$$\geq 10^{-2}\left(1 - \frac{n/\sigma^2 + \ln(2)}{0.99d}\right). \tag{189}$$

From the above, it is easy to see that with $n = \frac{1}{4}\sigma^2 d$ samples, the algorithm incurs an expected risk greater than $\frac{1}{2}10^{-2}$, proving the result. $\qquad \square$

---

[4]The claim and the proof do not depend on the chosen value of $b^{\star}$.

We now present the formal statement and the proof of Theorem 7.1, which establishes the $\Omega(\sigma^2 kd)$ sample complexity lower bound for LCNs when trained on DSD.

**Theorem 7.1** (Formal). Let $\mathcal{F}$ denote the class of functions represented by the set of locally connected neural network models, $\mathcal{M}_\mathcal{L}[\mathcal{W}]$, as defined in 4. Let $S^n \sim (\text{DSD})^n$ be the $n$ i.i.d. data samples drawn from DSD. We define the following groups, $\mathcal{U}_1 := \{\text{Block}(\mathbf{U}_1, \ldots, \mathbf{U}_k) \mid \mathbf{U}_i \in \mathcal{O}(d)\}$, $\mathcal{U}_2 := \{\mathbf{U} \in \mathcal{O}_p(kd) \mid \text{idx}_{kd}(\mathbf{U}e_{(i-1)d+1}) + j - 1 = \text{idx}_{kd}(\mathbf{U}e_{(i-1)d+j}), \forall i \in [k], j \in [d]\}$, and $\mathcal{U} = \mathcal{U}_1 \star \mathcal{U}_2$. Let $\{F_t\}_T$ be the set of update functions, and let the model parameters be initialized as $\boldsymbol{w}^0 \sim W$. If $\bar{\theta}_n(S^n, \boldsymbol{w}^0; \mathcal{M}_F[\mathcal{W}], \{F_t\}_T)$ is a $\mathcal{U}$-equivariant algorithm, then, for large enough $k, d$, the sample complexity is given by,

$$n_\delta(\bar{\theta}_n) = \max(\Omega\left(\sigma^2 kd\right), 40k), \tag{190}$$

where $\delta = 0.25 \times 10^{-2}$.

*Proof.* For simplicity will refer to the distribution DSD by $P$, and the distribution $\text{SSD}_t$ by $Q_t$, for $t \in [k]$. Since the algorithm $\bar{\theta}_n$ is $\mathcal{U}$-equivariant, lemma 5.1 gives us that for all $\mathbf{U} \in \mathcal{U}$,

$$\bar{\theta}(\{\boldsymbol{x}_i, y_i\}_n)(\boldsymbol{x}) \overset{d}{=} \bar{\theta}(\{\mathbf{U}\boldsymbol{x}_i, y_i\}_n)(\mathbf{U}\boldsymbol{x}), \tag{191}$$

$$\text{err}\left(\bar{\theta}(\{\boldsymbol{x}_i, y_i\}_n)(\boldsymbol{x}), y\right) \overset{d}{=} \text{err}\left(\bar{\theta}(\{\mathbf{U}\boldsymbol{x}_i, y_i\}_n)(\mathbf{U}\boldsymbol{x}), y\right), \tag{192}$$

$$\underset{S^n \sim P^n}{\mathbb{E}} \underset{(\boldsymbol{x}, y) \sim P}{\mathbb{E}} \text{err}\left(\bar{\theta}(\{\boldsymbol{x}_i, y_i\}_n)(\boldsymbol{x}), y\right) = \underset{S^n \sim P^n}{\mathbb{E}} \underset{(\boldsymbol{x}, y) \sim P}{\mathbb{E}} \text{err}\left(\bar{\theta}(\{\mathbf{U}\boldsymbol{x}_i, y_i\}_n)(\mathbf{U}\boldsymbol{x}), y\right), \tag{193}$$

$$\underset{S^n \sim P^n}{\mathbb{E}} \left[R\left(\bar{\theta}_n, P\right)\right] = \underset{S^n \sim (\mathbf{U} \circ P)^n}{\mathbb{E}} \left[R\left(\bar{\theta}_n, \mathbf{U} \circ P\right)\right], \tag{194}$$

$$= \underset{S^n \sim (\mathbf{U} \circ P)^n}{\mathbb{E}} \frac{1}{k} \sum_{i=1}^{k} \left[R\left(\bar{\theta}_n, \mathbf{U} \circ Q_i\right)\right], \tag{195}$$

$$= \frac{1}{k} \sum_{i=1}^{k} \underset{S^n \sim (\mathbf{U} \circ P)^n}{\mathbb{E}} \left[R\left(\bar{\theta}_n, \mathbf{U} \circ Q_i\right)\right]. \tag{196}$$

To simplify 196, we begin by showing that the expected risk incurred by the algorithm is the same for every distribution $\mathbf{U} \circ Q_i$. Specifically, for all $i, j \in [k]$,

$$\underset{S^n \sim (\mathbf{U} \circ P)^n}{\mathbb{E}} \left[R\left(\bar{\theta}_n, \mathbf{U} \circ Q_i\right)\right] = \underset{S^n \sim (\mathbf{U} \circ P)^n}{\mathbb{E}} \left[R\left(\bar{\theta}_n, \mathbf{U} \circ Q_j\right)\right]. \tag{197}$$

For $i = j$, the result trivially holds. So we can assume that $i \neq j$. Observe that because of the block structure of $\mathcal{U}_1$, $\mathbf{U}\boldsymbol{\mu}_i \in \text{Span}(\{e_{(i-1)d+j}\}_{j \in [d]}) \forall i \in [k]$. Therefore $\exists \mathbf{U}_1 \in \mathcal{U}_1, \mathbf{U}_2 \in \mathcal{U}_2$, such that, $\mathbf{U}_1\mathbf{U}_2\mathbf{U}\boldsymbol{\mu}_l = \mathbf{U}\boldsymbol{\mu}_l$ for all $l \notin \{i, j\}$, and $\mathbf{U}_1\mathbf{U}_2\mathbf{U}\boldsymbol{\mu}_i = \mathbf{U}\boldsymbol{\mu}_j$, $\mathbf{U}_1\mathbf{U}_2\mathbf{U}\boldsymbol{\mu}_j = \mathbf{U}\boldsymbol{\mu}_i$. Since $\tilde{\mathbf{U}} := \mathbf{U}_1\mathbf{U}_2 \in \mathcal{U}$ and $\bar{\theta}_n$ is a $\mathcal{U}$-orthogonally equivariant algorithm, from lemma 5.1,

$$\bar{\theta}(\{\boldsymbol{x}_i, y_i\}_n)(\boldsymbol{x}) \overset{d}{=} \bar{\theta}(\{\mathbf{U}_1\boldsymbol{x}_i, y_i\}_n)(\mathbf{U}_1\boldsymbol{x}), \tag{198}$$

$$\text{err}\left(\bar{\theta}(\{\boldsymbol{x}_i, y_i\}_n)(\boldsymbol{x}), y\right) \overset{d}{=} \text{err}\left(\bar{\theta}(\{\mathbf{U}_1\boldsymbol{x}_i, y_i\}_n)(\mathbf{U}_1\boldsymbol{x}), y\right), \tag{199}$$

$$\underset{S^n \sim (\mathbf{U} \circ P)^n}{\mathbb{E}} \underset{\mathbf{U} \circ Q_i}{\mathbb{E}} \left[\text{err}\left(\bar{\theta}(\{\boldsymbol{x}_i, y_i\}_n)(\boldsymbol{x}), y\right)\right] = \underset{S^n \sim (\mathbf{U} \circ P)^n}{\mathbb{E}} \underset{\mathbf{U} \circ Q_i}{\mathbb{E}} \left[\text{err}(\bar{\theta}(\{\mathbf{U}_1\boldsymbol{x}_i, y_i\}_n)\right.$$
$$\left.(\mathbf{U}_1\boldsymbol{x}), y)\right], \tag{200}$$

$$\underset{S^n \sim (\mathbf{U} \circ P)^n}{\mathbb{E}} \underset{\mathbf{U} \circ Q_i}{\mathbb{E}} \left[\text{err}\left(\bar{\theta}(\{\boldsymbol{x}_i, y_i\}_n)(\boldsymbol{x}), y\right)\right] = \underset{S^n \sim (\mathbf{U}_1\mathbf{U} \circ P)^n}{\mathbb{E}} \underset{\mathbf{U}_1\mathbf{U} \circ Q_i}{\mathbb{E}} \left[\text{err}(\bar{\theta}(\{\boldsymbol{x}_i, y_i\}_n)\right.$$
$$\left.(\boldsymbol{x}), y)\right]. \tag{201}$$

From the construction of $\mathbf{U}_1$, we know that $\mathbf{U}_1\mathbf{U} \circ P \overset{d}{=} \mathbf{U} \circ P$, and $\mathbf{U}_1\mathbf{U} \circ Q_i \overset{d}{=} \mathbf{U} \circ Q_j$,

$$\underset{S^n \sim (\mathbf{U} \circ P)^n}{\mathbb{E}} \underset{\mathbf{U} \circ Q_i}{\mathbb{E}} \left[\text{err}\left(\bar{\theta}(\{\boldsymbol{x}_i, y_i\}_i)(\boldsymbol{x}), y\right)\right] = \underset{S^n \sim (\mathbf{U} \circ P)^n}{\mathbb{E}} \underset{\mathbf{U} \circ Q_j}{\mathbb{E}} \left[\text{err}\left(\bar{\theta}(\{\boldsymbol{x}_i, y_i\}_i)(\boldsymbol{x}), y\right)\right], \tag{202}$$

$$\underset{S^n \sim (\mathbf{U} \circ P)^n}{\mathbb{E}} \left[R\left(\bar{\theta}_n, \mathbf{U} \circ Q_i\right)\right] = \underset{S^n \sim (\mathbf{U} \circ P)^n}{\mathbb{E}} \left[R\left(\bar{\theta}_n, \mathbf{U} \circ Q_j\right)\right]. \tag{203}$$

This proves the claim 197. Substituting it back into 196,

$$\mathbb{E}_{S^n \sim P^n} \left[ R\left(\bar{\theta}_n, P\right) \right] = \mathbb{E}_{S^n \sim (\mathbf{U} \circ P)^n} \left[ R\left(\bar{\theta}_n, \mathbf{U} \circ Q_1\right) \right], \tag{204}$$

$$= \sup_{\mathbf{U} \in \tilde{\mathcal{U}}} \mathbb{E}_{S^n \sim (\mathbf{U} \circ P)^n} \left[ R\left(\bar{\theta}_n, \mathbf{U} \circ Q_1\right) \right], \tag{205}$$

where $\tilde{\mathcal{U}} \subseteq \mathcal{U}_1$ is the set of "hard instances" such that, for all $\mathbf{U} \neq \mathbf{V} \in \tilde{\mathcal{U}}$, $(\mathbf{U}\boldsymbol{\mu}_1)^T(\mathbf{V}\boldsymbol{\mu}_1) < 10^{-3}$, and for all $\mathbf{U} \in \tilde{\mathcal{U}}$, and $i \in \{2, \ldots, k\}$, $\mathbf{U}\boldsymbol{\mu}_i = \boldsymbol{e}_{dt}$, Let $\Xi = \mathcal{W}$, $P_\Xi = W$, and $\Theta = \{\theta \mid \theta \colon ((\mathcal{X}, \mathcal{Y})^n, \Xi) \to \mathcal{F}\}$. It is easy to note that $\bar{\theta}_n \in \Theta$. Therefore,

$$\mathbb{E}_{S^n \sim P^n} \left[ R\left(\bar{\theta}_n, P\right) \right] \geq \inf_{\theta_n \in \Theta} \sup_{\mathbf{U} \in \tilde{\mathcal{U}}} \mathbb{E}_{S^n \sim (\mathbf{U} \circ P)^n} \left[ R\left(\theta_n, \mathbf{U} \circ Q_1\right) \right], \tag{206}$$

We will now perform a series of reductions to lower bound the above minimax problem, with the minimax problem of learning $\mathrm{SSD}_1$. The main idea behind these reductions is to demonstrate that a given minimax problem can be 'simulated' by a more tractable one, and thus the tractable problem serves as a lower bound on the original problem.

Define the set of algorithms, $\Theta_1 = \{\theta \mid \theta \colon (([k], \mathcal{X}, \mathcal{Y})^n, \Xi) \to \mathcal{F}\}$, and let $\mathbf{U} \circ \tilde{P}$ be the indexed distribution with the generative story: Sample $j \sim \mathrm{Unif}[k]$, then sample $(\boldsymbol{x}, y) \sim \mathbf{U} \circ Q_j$, and then return $(j, \boldsymbol{x}, y)$. We can then lower bound 206 as,

$$\geq \inf_{\theta \in \Theta_1} \sup_{\mathbf{U} \in \tilde{\mathcal{U}}} \mathbb{E}_{\boldsymbol{w} \sim W} \mathbb{E}_{(j, \boldsymbol{x}, y)^n \sim (\mathbf{U} \circ \tilde{P})^n} \left[ R\left(\theta((j, \boldsymbol{x}, y)^n, \boldsymbol{w}), \mathbf{U} \circ Q_1\right) \right]. \tag{207}$$

The inequality follows from the fact that for every $\theta_n^a \in \Theta$, there exists $\theta_n^b \in \Theta_1$, that discards the index $j$ and returns the output of $\theta_n^a$.

We define $n_1$ to be the random variable that corresponds to the number of samples drawn from $\mathbf{U} \circ Q_1$. Using Berstein's inequality, we get that $\frac{n}{2k} \leq n_1 \leq m := \frac{3n}{2k}$, holds with probability $\geq c := 1 - 2\exp(\frac{-n}{10k})$. We will refer to this event as $E$. Then we can lower bound 207,

$$\geq c \inf_{\theta \in \Theta_1} \sup_{\mathbf{U} \in \tilde{\mathcal{U}}} \mathbb{E}_{\boldsymbol{w} \sim W} \mathbb{E}_{(j, \boldsymbol{x}, y)^n \sim (\mathbf{U} \circ \tilde{P})^n} \left[ R\left(\theta((j, \boldsymbol{x}, y)^n, \boldsymbol{w}), \mathbf{U} \circ Q_1\right) \mid E \right]. \tag{208}$$

For the next reduction, we define $n_i$ to be the random variable corresponding to the number of samples drawn from the distribution $\mathbf{U} \circ Q_i$, for all $i \in [k]$. Let $\boldsymbol{y} \sim (\mathrm{Unif}[\mathcal{Y}])^n$ be a uniform random vector over $\{+1, -1\}$ of size $n$, and $\boldsymbol{\epsilon} \sim \mathcal{N}(\boldsymbol{0}_{nkd}, \mathbf{I}_{nkd})$ be a vector of i.i.d. standard Gaussian random variables. Let $\Theta_2 = \{\theta \mid \theta \colon ((\mathcal{X}, \mathcal{Y})^m \times (\mathbb{R}^{kd})^{k-1} \times (\mathbb{N} \cup \{0\})^k \times (\mathbb{R})^n \times \mathbb{R}^{nkd} \times \mathcal{W}) \to \mathcal{F}\}$ be a set of algorithms that take as input the training data, $\{\mathbf{U}\boldsymbol{\mu}\}_{i=2}^k$ mean vectors, the number of samples to be drawn from each mean, pre-sampled values of $\boldsymbol{y}$ and $\boldsymbol{\epsilon}$, and the parameter initialization respectively. It subsequently returns a function within $\mathcal{F}$. Then we can lower bound 208 as,

$$\geq c \inf_{\theta \in \Theta_2} \sup_{\mathbf{U} \in \tilde{\mathcal{U}}} \mathbb{E}_{\boldsymbol{w} \sim W} \mathbb{E}_{\{n_i\}_1^k} \mathbb{E}_{\boldsymbol{y}, \boldsymbol{\epsilon}} \mathbb{E}_{S^m \sim (\mathbf{U} \circ Q_1)^m} \left[ R\left(\theta(S^m, \{\mathbf{U}\boldsymbol{\mu}_i\}_2^k, \{n_i\}_1^k, \boldsymbol{y}, \boldsymbol{\epsilon}, \boldsymbol{w}), \mathbf{U} \circ Q_1\right) \mid E \right]. \tag{209}$$

The last inequality follows from the fact that for every $\theta_n^a \in \Theta_1$, there exists $\theta_n^b \in \Theta_2$, that first deterministically creates the indexed dataset using $S^m, \{\mathbf{U}\boldsymbol{\mu}_i\}_2^k, \{n_i\}_1^k, \boldsymbol{y}, \boldsymbol{\epsilon}$ and then runs $\theta_n^a$. For notational brevity, we define $\Xi_1 := (\mathbb{N} \cup \{0\})^k \times (\mathbb{R})^n \times \mathbb{R}^{nkd} \times \mathcal{W}$, to encapsulate the randomness in $\{n_i\}_1^k, \boldsymbol{y}, \boldsymbol{\epsilon}$, and $\boldsymbol{w}$. We denote its associated product distribution by $P_{\Xi_1}$. Rewriting 209,

$$= c \inf_{\theta \in \Theta_2} \sup_{\mathbf{U} \in \tilde{\mathcal{U}}} \mathbb{E}_{\xi \sim P_{\Xi_1}} \mathbb{E}_{S^m \sim (\mathbf{U} \circ Q_1)^m} \left[ R\left(\theta(S^m, \{\mathbf{U}\boldsymbol{\mu}_i\}_2^k, \xi), \mathbf{U} \circ Q_1\right) \right]. \tag{210}$$

From the construction of "hard instances", we know that for all $\mathbf{U} \in \tilde{\mathcal{U}}$, $t \in \{2, \ldots, k\}$, $\mathbf{U}\boldsymbol{\mu}_t = \boldsymbol{e}_{dt}$. Substituting this back in 210,

$$= c \inf_{\theta \in \Theta_2} \sup_{\mathbf{U} \in \tilde{\mathcal{U}}} \mathbb{E}_{\xi \sim P_{\Xi_1}} \mathbb{E}_{S^m \sim (\mathbf{U} \circ Q_1)^m} \left[ R\left(\theta(S^m, \{\boldsymbol{e}_{di}\}_2^k, \xi), \mathbf{U} \circ Q_1\right) \right]. \tag{211}$$

Note that the set $\{\boldsymbol{e}_{di}\}_{i=2}^k$ is fixed and known. Consider, $\Theta_3 := \{\theta \mid \theta \colon ((\mathcal{X}, \mathcal{Y})^m \times \Xi_1) \to \mathcal{F}\}$, as the set of algorithms. For every $\theta_n^a \in \Theta_2$, there exists $\theta_n^b \in \Theta_3$ which runs $\theta_n^a$ using the input data, randomization $\xi$, and the known set $\{\boldsymbol{e}_{di}\}_{i=2}^k$. Therefore, we can bound 211,

$$\geq c \inf_{\theta \in \Theta_3} \sup_{\mathbf{U} \in \tilde{\mathcal{U}}} \mathbb{E}_{\xi \sim P_{\Xi_1}} \mathbb{E}_{S^m \sim (\mathbf{U} \circ Q_1)^m} \left[ R\left(\theta(S^m, \xi), \mathbf{U} \circ Q_1\right) \right], \tag{212}$$

We have already proven a lower bound for the above problem in Lemma D.3, specifically refer to equation 186. Substituting that result,

$$\mathbb{E}_{S^n \sim P^n} \left[ R\left(\bar{\theta}_n, P\right) \right] \geq c10^{-2} \left( 1 - \frac{m/\sigma^2 + \ln(2)}{0.99d} \right), \tag{213}$$

$$\geq c10^{-2} \left( 1 - \frac{\frac{3n}{2k\sigma^2} + \ln(2)}{0.99d} \right), \tag{214}$$

$$\geq \left( 1 - 2\exp(\tfrac{-n}{10k}) \right) 10^{-2} \left( 1 - \frac{\frac{3n}{2k\sigma^2} + \ln(2)}{0.99d} \right). \tag{215}$$

Using $n \geq 40k$, we can bound $c \geq \left( 1 - 2\exp(-\ln(4)) \right) = \frac{1}{2}$. And, choosing $n = \frac{1}{6}\sigma^2 kd$, we can bound $\left( 1 - \frac{\frac{3n}{2k\sigma^2} + \ln(2)}{0.99kd} \right) = \left( 1 - \frac{\frac{kd}{4} + \ln(2)}{0.99kd} \right) \geq \frac{1}{2}$. Therefore, we have the result,

$$\mathbb{E}_{S^n \sim P^n} \left[ R\left(\bar{\theta}_n, P\right) \right] \geq \tfrac{1}{4}10^{-2}. \tag{216}$$

$\square$

## D.2 CNN UPPER BOUND

**Theorem 7.2** (Formal). Let $\mathcal{F}$ denote the class of functions represented by the set of locally connected neural network models, $\mathcal{M}_C[\mathcal{W}]$, as defined in 6. Let the input data be drawn from the DSD distribution, $S^n \sim (\text{DSD})^n$, with $\sigma = \tilde{O}(\frac{1}{\sqrt{k}})$. We define the group $\mathcal{U} := \{\text{Block}(\mathbf{U}_1, \dots, \mathbf{U}_k) \mid \mathbf{U}_i \in \mathcal{O}(d), \mathbf{U}_i = \mathbf{U}_j\}$. Then there exists a weight initialization distribution $W$ and update functions $\{F_t\}_T$ such that $\bar{\theta}_n(\mathcal{M}_C[\mathcal{W}], \{F_t\}_T, W, S^n)$ is an $\mathcal{U}$-equivariant algorithm and, if $k, d$ are large enough, then

$$n_\delta(\bar{\theta}_n) = \max(O\left(\sigma^2(d+k)\ln(kd)\right), 10), \tag{217}$$

for some constant $\delta = O(1)$.

*Proof.* The outline of the proof will run parallel to the approach taken in the proof of Theorem 6.2. We will first present the algorithm $\bar{\theta}_n$, then show it is a $\mathcal{U}$-equivariant algorithm and then derive the required sample complexity bound upper bound.

1. Algorithm Definition

To define the algorithm $\bar{\theta}_n$, we need to specify its components: the model $\mathcal{M}_C[\mathcal{W}]$, the initialization distribution $W$, and the update functions $\{F_t\}_T$. At iteration $t = 0$, we initialize the model parameter $\boldsymbol{v}^0 = [\boldsymbol{w}^0, b^0]$ as $\boldsymbol{w}^0 \sim \mathcal{N}(\mathbf{0}, \gamma \mathbf{I}_d)$, where $\gamma^{-1} = 100k^2d^2$, and bias is set as $b^0 = 0$. The superscript denotes the iteration number. To specify the update functions, we define the empirical loss function,

$$l \colon (\mathcal{W}, (\mathcal{X}, \mathcal{Y})^n) \to \mathbb{R} := \frac{1}{n} \sum_{j=1}^{n} \left(y_i - \sum_{i=1}^{k} \phi_b(\boldsymbol{w}^T \boldsymbol{x}_j^{(i)})\right)^2. \tag{218}$$

The algorithm has $T = 2$ iterations. For simpler analysis, we divide the dataset, $S^n$, into two equal sized datasets $S_1^m$, and $S_1^m$, with $m := \frac{n}{2}$ samples each. The update function for each $t \in \{1, 2\}$ is,

$$F_t(\boldsymbol{v}, S_t^m) := \left[\frac{\boldsymbol{w} - \eta_t \nabla_{\boldsymbol{w}} l(\boldsymbol{w}, b; S_t^m)}{\|\boldsymbol{w} - \eta_t \nabla_{\boldsymbol{w}} l(\boldsymbol{w}, b; S_t^m)\|} \; ; \; b_t\right], \tag{219}$$

where $\eta_1 = 1, \eta_2 = 10^3, b_1 = \frac{1}{100}\sqrt{\frac{kd}{(k+d)\ln(kd)}}, b_2 = 10^{-4}$.

2. Algorithm is Equivariant

To establish that $\bar{\theta}_n$ is $\mathcal{U}$-equivariant, we verify the three conditions specified in Definition 6. We define the group, $\mathcal{V} := \{\text{Block}(\mathbf{V}, \mathbf{I}_1) \mid \mathbf{V} \in \mathcal{O}(d)\}$, where $\mathbf{I}_1$ is the identity matrix of size 1.

For $\boldsymbol{x}, y \in (\mathcal{X}, \mathcal{Y})$, $\mathbf{U} \in \mathcal{U}$, $\boldsymbol{w} \in \mathbb{R}^d$, $b \in \mathbb{R}_+$, choose $\mathbf{V} = \text{Block}(\{\mathbf{U}^{(1)}, \mathbf{I}_1\}) \in \mathcal{V}$, without loss of generality, as for all $i, j$, $\mathbf{U}^{(i)} = \mathbf{U}^{(j)}$. Then, the property 1 of equivariance holds as,

$$\mathcal{M}_C[\boldsymbol{v}](\boldsymbol{x}) = \sum_{i=1}^{k} \phi_b(\boldsymbol{w}^T \boldsymbol{x}^{(i)}) = \sum_{i=1}^{k} \phi_b(\boldsymbol{w}^T (\mathbf{U}^{(1)})^T \mathbf{U}^{(i)} \boldsymbol{x}^{(i)}) = \mathcal{M}_C[\mathbf{V}\boldsymbol{v}](\mathbf{U}\boldsymbol{x}). \tag{220}$$

For all $t \in [2]$ and $S_t^m \in (\mathcal{X}, \mathcal{Y})^m$ the second property 2 follows as,

$$F_t(\mathbf{V}\boldsymbol{v}, \mathbf{U} \circ S_t^m) = \left[\frac{\mathbf{U}^{(1)}\boldsymbol{w} - \eta_t \nabla_{\mathbf{U}^{(1)}\boldsymbol{w}} l(\mathbf{V}\boldsymbol{v}; \mathbf{U} \circ S_t^m)}{\|\mathbf{U}^{(1)}\boldsymbol{w} - \eta_t \nabla_{\mathbf{U}^{(1)}\boldsymbol{w}} l(\mathbf{V}\boldsymbol{v}; \mathbf{U} \circ S_t^m)\|} \; ; \; b_t\right], \tag{221}$$

$$= \left[\frac{\mathbf{U}^{(1)}(\boldsymbol{w} - \eta_t \nabla_{\boldsymbol{w}} l(\boldsymbol{v}; S_t^m))}{\|\mathbf{U}^{(1)}(\boldsymbol{w} - \eta_t \nabla_{\boldsymbol{w}} l(\boldsymbol{v}; S_t^m))\|} \; ; \; b_t\right], \tag{222}$$

$$= \left[\frac{\mathbf{U}^{(1)}(\boldsymbol{w} - \eta_t \nabla_{\boldsymbol{w}} l(\boldsymbol{v}; S_t^m))}{\|\boldsymbol{w} - \eta_t \nabla_{\boldsymbol{w}} l(\boldsymbol{v}; S_t^m)\|} \; ; \; b_t\right], \tag{223}$$

$$= \mathbf{V} F_t(\boldsymbol{v}, S_t^m). \tag{224}$$

And as for property 3, observe that,

$$\mathbf{V}\boldsymbol{v}^0 = [\mathbf{U}^{(1)}\boldsymbol{w}^0 b^0], \stackrel{d}{=} [\boldsymbol{w}^0, b^0] = \boldsymbol{v}, \tag{225}$$

holds for all $\mathbf{V} \in \mathcal{V}$.

### 3. Algorithm Analysis

We analyze the algorithm, with $n = \max(2\sigma^2(k+d)\ln(kd), 10)$ samples, to establish that $\bar{\theta}_n$ achieve an expected risk of at most $\delta = 2.5 \times 10^{-3}$. We set $\sigma \leq \frac{1}{100\sqrt{k\ln(kd)^3}}$. The outline of the proof is as follows: we first prove that after the first update step, the alignment of $\boldsymbol{w}^1$ with unknown signal vector $\boldsymbol{w}^\star$ is $\Omega(\sqrt{\frac{1}{k}})$. In the second step, we use this alignment is reliably threshold out the "noise" patches, while letting the "signal" patch pass through the first hidden layer. We then show that this denoising effect, enables us to recover the signal with an alignment of $\Omega(1)$, which would imply that the risk of the CNN on the task $\leq \delta$.

### 3a. Update Step 1

We define $\hat{\boldsymbol{w}}^1 = \boldsymbol{w}^0 - \nabla_{\boldsymbol{w}^0} l(\boldsymbol{w}^0, 0; S_1^m)$ to be the unnormalized parameter vector $\boldsymbol{w}^1$, and therefore the alignment with the signal is given by $(\boldsymbol{w}^1)^T \boldsymbol{w}^\star = \frac{(\hat{\boldsymbol{w}}^1)^T \boldsymbol{w}^\star}{\|\hat{\boldsymbol{w}}^1\|}$. To analyze $\hat{\boldsymbol{w}}^1$, we first evaluate the gradient with respect to $\boldsymbol{w}^0$, $\nabla_{\boldsymbol{w}^0} l(\boldsymbol{w}^0, 0; S_1^m)$,

$$\nabla_{\boldsymbol{w}^0} l(\boldsymbol{w}^0, 0; S_1^m) = \frac{1}{m} \sum_{j=1}^{m} \nabla_{\boldsymbol{w}^0} \left( y_i - \sum_{i=1}^{k} \phi_0((\boldsymbol{w}^0)^T \boldsymbol{x}_j^{(i)}) \right)^2, \tag{226}$$

$$= \frac{-2}{m} \sum_{j=1}^{m} \left( y_j - \sum_{i=1}^{k} \phi_0((\boldsymbol{w}^0)^T \boldsymbol{x}_j^{(i)}) \right) \left( \sum_{i=1}^{k} \boldsymbol{x}_j^{(i)} \phi_0'((\boldsymbol{w}^0)^T \boldsymbol{x}_j^{(i)}) \right), \tag{227}$$

$$= \frac{-2}{m} \sum_{j=1}^{m} \left( 1 - \sum_{i=1}^{k} y_j (\boldsymbol{w}^0)^T \boldsymbol{x}_j^{(i)} \right) \left( \sum_{i=1}^{k} y_j \boldsymbol{x}_j^{(i)} \right), \tag{228}$$

$$:= \frac{-2}{m} \sum_{j=1}^{m} \alpha_j \beta_j, \tag{229}$$

where $\phi_0'(x) := \frac{d}{dx}\phi_0(x)$. We have used the facts that $\phi_0$ is the identity function, and $\phi_0'$ is the constant function 1 . And, $\alpha_j := 1 - \sum_{i=1}^{k} y_j (\boldsymbol{w}^0)^T \boldsymbol{x}_j^{(i)}$, $\beta_j := \sum_{i=1}^{k} y_j \boldsymbol{x}_j^{(i)}$.

To further analyze 229, we first prove high probability bounds for $\alpha_j$, $j \in [m]$. From the initialization distribution $W$, we know that $\boldsymbol{w}^0 \stackrel{d}{=} \gamma \boldsymbol{\epsilon}$, where $\boldsymbol{\epsilon}$ is the Gaussian random vector defined as $\boldsymbol{\epsilon} \sim \mathcal{N}(\mathbf{0}, \mathbf{I}_d)$. And from the input distribution DSD, we know that $\boldsymbol{x}_j^{(i)} = y_j r_{ij} \boldsymbol{w}^\star + \sigma \boldsymbol{\epsilon}_j^{(i)}$, for all $i$ in $[k]$. Here, $\boldsymbol{\epsilon}_j^{(i)} \sim \mathcal{N}(\mathbf{0}, \mathbf{I}_d)$ is also a Gaussian random vector, and $r_{ij} = 1$, if the signal patch appears in the $j$-th data sample appears in the $i$-th patch, and 0 otherwise.

$$\alpha_j = 1 - \sum_{i=1}^{k} y_j (\boldsymbol{w}^0)^T \boldsymbol{x}_j^{(i)}, \tag{230}$$

$$= 1 - \sum_{i=1}^{k} r_{ij} y_j^2 \gamma \boldsymbol{\epsilon}^T \boldsymbol{w}^\star - \sum_{i=1}^{k} y_j \gamma \sigma \boldsymbol{\epsilon}^T \boldsymbol{\epsilon}_j^{(i)}, \tag{231}$$

$$= 1 - \gamma \boldsymbol{\epsilon}^T \boldsymbol{w}^\star - \sum_{i=1}^{k} y_j \gamma \sigma \boldsymbol{\epsilon}^T \boldsymbol{\epsilon}_j^{(i)}, \tag{232}$$

We can now bound the range of $\alpha_j$ as,

$$1 + |\gamma \boldsymbol{\epsilon}^T \boldsymbol{w}^\star| + |\sum_{i=1}^{k} \gamma \sigma \boldsymbol{\epsilon}^T \boldsymbol{\epsilon}_j^{(i)}| \geq \alpha_j \geq 1 - |\gamma \boldsymbol{\epsilon}^T \boldsymbol{w}^\star| - |\sum_{i=1}^{k} \gamma \sigma \boldsymbol{\epsilon}^T \boldsymbol{\epsilon}_j^{(i)}|. \tag{233}$$

We first upper bound $|\gamma \boldsymbol{\epsilon}^T \boldsymbol{w}^\star|$. Since the norm of the signal is 1, $\|\boldsymbol{w}^\star\| = 1$, $\boldsymbol{\epsilon}^T \boldsymbol{w}^\star \sim \mathcal{N}(0, 1)$, and

$$|\gamma \boldsymbol{\epsilon}^T \boldsymbol{w}^\star| \leq \frac{1}{100k^2d^2} |\boldsymbol{\epsilon}^T \boldsymbol{w}^\star| \leq \frac{1}{8}, \tag{234}$$

with probability $\geq 1 - 2\Phi(-10k^2 d^2) \geq 1 - 10^{-6}$, for large enough $k, d$. Next, we provide an upper bound for $|\sum_{i=1}^{k} \gamma\sigma\boldsymbol{\epsilon}^T \boldsymbol{\epsilon}_j^{(i)}|$, for all $j$. For this we analyze,

$$\max_{j\in[m]} |\sum_{i=1}^{k} \gamma\sigma\boldsymbol{\epsilon}^T \boldsymbol{\epsilon}_j^{(i)}| = \gamma\sigma \max_{j\in[m]} |\boldsymbol{\epsilon}^T \sum_{i=1}^{k} \boldsymbol{\epsilon}_j^{(i)}| \overset{d}{=} \gamma\sigma \max_{j\in[m]} |\sqrt{k}\boldsymbol{\epsilon}^T \bar{\boldsymbol{\epsilon}}_j|, \tag{235}$$

$$= \gamma\sigma\sqrt{k} \max_{j\in[m]} |\frac{\|\boldsymbol{\epsilon}\|}{\|\boldsymbol{\epsilon}\|}\boldsymbol{\epsilon}^T \bar{\boldsymbol{\epsilon}}_j| \leq 6\gamma\sigma\sqrt{kd} \max_{j\in[m]} |\frac{\boldsymbol{\epsilon}^T \bar{\boldsymbol{\epsilon}}_j}{\|\boldsymbol{\epsilon}\|}|, \tag{236}$$

with probability $\geq 1 - 2 \times 10^{-6}$. The last inequality 236 follows from the concentration of the norm of a Gaussian random variable, $\mathbb{P}[\|\boldsymbol{\epsilon}\| \geq 6\sqrt{d}] \leq 2\exp(-\frac{36d}{2d}) \leq 10^{-6}$. We define $\boldsymbol{u} = \frac{\boldsymbol{\epsilon}}{\|\boldsymbol{\epsilon}\|}$, and $\epsilon_j = \boldsymbol{u}^T \bar{\boldsymbol{\epsilon}}_j$, which is a standard Gaussian random variable. Then, from the concentration inequality, $\mathbb{P}[\max_{j\in[m]} |\epsilon_j| \geq \sqrt{32\ln(m)}] \leq \frac{2}{m^9} \leq 10^{-6}$. Substituting this in 236,

$$\max_{j\in[m]} |\gamma\sigma\boldsymbol{\epsilon}^T \sum_{i=1}^{k} \boldsymbol{\epsilon}_j^{(i)}| \leq 6\gamma\sigma\sqrt{kd} \max_{j\in[m]} |\epsilon_j|, \tag{237}$$

$$\leq 6\frac{1}{100k^2 d^2} \frac{1}{100\sqrt{k\ln(kd)^3}}\sqrt{kd} \max_{j\in[m]} |\epsilon_j|, \tag{238}$$

$$\leq 6\frac{1}{100k^2 d^2} \frac{1}{100\sqrt{k\ln(kd)^3}}\sqrt{kd}\sqrt{32\ln(m)} \leq \frac{1}{8}, \tag{239}$$

for large enough $k, d$. Using 234, 239 in 233, we bound $\alpha_j$, for all $j \in [m]$, as,

$$1 + |\gamma\boldsymbol{\epsilon}^T \boldsymbol{w}^\star| + |\sum_{i=1}^{k} \gamma\sigma\boldsymbol{\epsilon}^T \boldsymbol{\epsilon}_j^{(i)}| \geq \alpha_j \geq 1 - |\gamma\boldsymbol{\epsilon}^T \boldsymbol{w}^\star| - |\sum_{i=1}^{k} \gamma\sigma\boldsymbol{\epsilon}^T \boldsymbol{\epsilon}_j^{(i)}|. \tag{240}$$

$$\frac{5}{4} \geq \alpha_j \geq \frac{3}{4}. \tag{241}$$

Also, note that $\beta_j = \sum_{i=1}^{k} y_j \boldsymbol{x}_j^{(i)} \overset{d}{=} \boldsymbol{w}^\star + \sigma\sqrt{k}\bar{\boldsymbol{\epsilon}}_j$. We are now in the position to analyze $\hat{\boldsymbol{w}}^1$,

$$\hat{\boldsymbol{w}}^1 = \boldsymbol{w}^0 - \nabla_{\boldsymbol{w}} l(\boldsymbol{w}^0, 0; S_1^m), \tag{242}$$

$$= \boldsymbol{w}^0 + \frac{1}{m}\sum_{j=1}^{m} 2\alpha_j \beta_j, \tag{243}$$

$$\overset{d}{=} \boldsymbol{w}^0 + \frac{1}{m}\sum_{j=1}^{m} 2\alpha_j \left(\boldsymbol{w}^\star + \sigma\sqrt{k}\bar{\boldsymbol{\epsilon}}_j\right), \tag{244}$$

$$\overset{d}{=} \boldsymbol{w}^0 + \frac{2\sum_{j=1}^{m}\alpha_j}{m}\boldsymbol{w}^\star + \frac{2\sigma\sqrt{k\sum_{j=1}^{m}\alpha_j^2}}{m}\bar{\boldsymbol{\epsilon}}, \tag{245}$$

where $\bar{\boldsymbol{\epsilon}}$ is the Gaussian random vector $\sim \mathcal{N}(\boldsymbol{0}, \mathbf{I}_d)$. Note that from the concentration of the norm of a Gaussian random variable $\mathbb{P}[\|\bar{\boldsymbol{\epsilon}}\| \geq 6\sqrt{d}] \leq 2\exp(-\frac{36d}{2d}) \leq 10^{-6}$, and $\mathbb{P}[\|\bar{\boldsymbol{\epsilon}}\| \leq \sqrt{d}/6] \leq 10^{-6}$,

Recall that our aim is to bound $(\boldsymbol{w}^1)^T \boldsymbol{w}^\star = \frac{(\hat{\boldsymbol{w}}^1)^T \boldsymbol{w}^\star}{\|\hat{\boldsymbol{w}}^1\|}$. For this, we first upper bound $\|\hat{\boldsymbol{w}}^1\|$,

$$\|\hat{\boldsymbol{w}}^1\| = \|\boldsymbol{w}^0 + \frac{2\sum_{j=1}^{m}\alpha_j}{m}\boldsymbol{w}^\star + \frac{2\sigma\sqrt{k\sum_{j=1}^{m}\alpha_j^2}}{m}\bar{\boldsymbol{\epsilon}}\|, \tag{246}$$

$$\leq \|\boldsymbol{w}^0\| + \|\frac{2\sum_{j=1}^{m}\alpha_j}{m}\boldsymbol{w}^\star\| + \|\frac{2\sigma\sqrt{k\sum_{j=1}^{m}\alpha_j^2}}{m}\bar{\boldsymbol{\epsilon}}\|, \tag{247}$$

$$\overset{241}{\leq} \|\boldsymbol{w}^0\| + \frac{5}{2}\|\boldsymbol{w}^\star\| + \frac{5\sigma}{2}\sqrt{\frac{k}{m}}\|\bar{\boldsymbol{\epsilon}}\|, \tag{248}$$

$$= \gamma\|\boldsymbol{\epsilon}\| + \frac{5}{2} + \frac{5\sigma}{2}\sqrt{\frac{k}{\sigma^2(d+k)\ln(kd)}}\|\bar{\boldsymbol{\epsilon}}\|, \tag{249}$$

$$\leq \frac{6\sqrt{d}}{100k^2 d^2} + \frac{5}{2} + \frac{5\sigma}{2}\sqrt{\frac{6kd}{\sigma^2(d+k)\ln(kd)}} \leq 10\sqrt{\frac{kd}{(k+d)\ln(kd)}}, \tag{250}$$

for large enough $k, d$. Similarly, we lower bound $\|\hat{\boldsymbol{w}}^1\|$,

$$\|\hat{\boldsymbol{w}}^1\| \geq \|\frac{2\sigma\sqrt{k\sum_{j=1}^m \alpha_j^2}}{m}\bar{\boldsymbol{\epsilon}}\| - \|\boldsymbol{w}^0\| - \|\frac{2\sum_{j=1}^m \alpha_j}{m}\boldsymbol{w}^\star\|, \tag{251}$$

$$\stackrel{241}{\geq} \frac{3\sigma}{2}\sqrt{\frac{k}{m}}\|\bar{\boldsymbol{\epsilon}}\| - \|\boldsymbol{w}^0\| - \frac{5}{2}\|\boldsymbol{w}^\star\|, \tag{252}$$

$$\geq \frac{3\sigma}{2}\sqrt{\frac{kd}{6\sigma^2(k+d)\ln(kd)}} - \frac{6\sqrt{d}}{100k^2d^2} - \frac{5}{2} \geq \frac{1}{4}\sqrt{\frac{kd}{(k+d)\ln(kd)}}, \tag{253}$$

where the last inequality holds for a large enough $k, d$. Next, we lower bound $(\hat{\boldsymbol{w}}^1)^T\boldsymbol{w}^\star$,

$$(\hat{\boldsymbol{w}}^1)^T\boldsymbol{w}^\star = (\boldsymbol{w}^0)^T\boldsymbol{w}^\star + \frac{2\sum_{j=1}^m \alpha_j}{m}(\boldsymbol{w}^\star)^T\boldsymbol{w}^\star + \frac{2\sigma\sqrt{k\sum_{j=1}^m \alpha_j^2}}{m}(\bar{\boldsymbol{\epsilon}})^T\boldsymbol{w}^\star, \tag{254}$$

$$= \gamma\boldsymbol{\epsilon}^T\boldsymbol{w}^\star + \frac{2\sum_{j=1}^m \alpha_j}{m} + \frac{2\sigma\sqrt{k\sum_{j=1}^m \alpha_j^2}}{m}(\bar{\boldsymbol{\epsilon}})^T\boldsymbol{w}^\star, \tag{255}$$

$$\geq -|\gamma\boldsymbol{\epsilon}^T\boldsymbol{w}^\star| + \frac{3}{2} - \frac{5\sigma\sqrt{k}}{\sqrt{m}}|(\bar{\boldsymbol{\epsilon}})^T\boldsymbol{w}^\star|, \tag{256}$$

$$\stackrel{234}{\geq} -\frac{1}{8} + \frac{3}{2} - \frac{5\sigma\sqrt{k}}{\sqrt{\sigma^2(k+d)\ln(kd)}}|(\bar{\boldsymbol{\epsilon}})^T\boldsymbol{w}^\star|, \tag{257}$$

$$\geq \frac{11}{8} - \frac{1}{12}|(\boldsymbol{\epsilon}^1)^T\boldsymbol{w}^\star| \geq \frac{11}{8} - \frac{7}{10} \geq \frac{6}{10}, \tag{258}$$

with probability $\geq 1 - 2\Phi(-\frac{84}{10}) \geq 1 - 10^{-6}$. And similarly we upper bound $(\hat{\boldsymbol{w}}^1)^T\boldsymbol{w}^\star$,

$$(\hat{\boldsymbol{w}}^1)^T\boldsymbol{w}^\star \leq \frac{5}{2} + |\gamma\boldsymbol{\epsilon}^T\boldsymbol{w}^\star| + \frac{5\sigma\sqrt{k}}{\sqrt{m}}|(\boldsymbol{\epsilon}^1)^T\boldsymbol{w}^\star|, \tag{259}$$

$$\leq \frac{5}{2} + \frac{1}{8} + \frac{7}{10} \leq 4. \tag{260}$$

Therefore, $40\sqrt{\frac{kd}{(k+d)\ln(kd)}} \geq (\boldsymbol{w}^1)^T\boldsymbol{w}^\star \geq \frac{6}{100}\sqrt{\frac{kd}{(k+d)\ln(kd)}}$, with probability $\geq 1 - 10^{-5}$. We can now express $\boldsymbol{w}^1$ as $\lambda\boldsymbol{w}^\star + \sqrt{1-\lambda^2}\boldsymbol{w}_\perp^\star$, such that $40\sqrt{\frac{kd}{(k+d)\ln(kd)}} \geq \lambda \geq \frac{6}{100}\sqrt{\frac{kd}{(k+d)\ln(kd)}}$, $\|\boldsymbol{w}_\perp^\star\| = 1$, and $(\boldsymbol{w}^\star)^T\boldsymbol{w}_\perp^\star = 0$.

3b. Update Step 2

In this step, we will now show that the $\frac{6}{100}\sqrt{\frac{kd}{(k+d)\ln(kd)}}$ alignment achieved in the first step, enables the network to filter out the noise patches, while letting from the signal patch pass through. This denoising will enables us to achieve a stronger a $1 - 10^{-3}$ alignment with the signal vector.

We begin by analyzing the push forward of all noise patches in $S_2^m$, through the CNN model,

$$\max_{i\in[k], j\in[n]\setminus[m]} |\sigma(\boldsymbol{w}^1)^T\boldsymbol{\epsilon}_j^{(i)}| \leq \frac{1}{100\sqrt{k\ln(kd)^3}}\max_{i,j}|(\boldsymbol{w}_i^1)^T\boldsymbol{\epsilon}_j^{(i)}|, \tag{261}$$

$$\leq \frac{1}{100\sqrt{k\ln(kd)^3}}\sqrt{32\ln(\sigma^2 k(k+d)\ln(kd))}, \tag{262}$$

$$\leq \frac{1}{4\sqrt{k}} \leq \frac{1}{100}\sqrt{\frac{kd}{(k+d)\ln(kd)}} \tag{263}$$

To derive inequality 262, we have used the concentration of the maximum of the absolute value of $mk$ i.i.d. Gaussian random variables, $\mathbb{P}[\max_{i,j}|(\boldsymbol{w}^1)^T\boldsymbol{\epsilon}_j^{(i)}| \geq \sqrt{32\ln(mk)}] \leq \frac{2}{(mk)^9} \leq 10^{-6}$. Recall from the analysis of the first update step that,

$$40\sqrt{\frac{kd}{(k+d)\ln(kd)}} \geq (\boldsymbol{w}^1)^T\boldsymbol{w}^\star \geq \frac{6}{100}\sqrt{\frac{kd}{(k+d)\ln(kd)}}. \tag{264}$$

From 263, 264, and $b_1 = \frac{1}{100}\sqrt{\frac{kd}{(k+d)\ln(kd)}}$, we filter out the noise and let the signal pass for all $j$,

$$40\sqrt{\frac{kd}{(k+d)\ln(kd)}} \geq \phi_{b_1}(y_j(\boldsymbol{w}^1)^T\boldsymbol{x}_j^{(i)}) \geq \frac{4}{100}\sqrt{\frac{kd}{(k+d)\ln(kd)}}, \quad \text{where } r_{ij} = 1, \tag{265}$$

$$\phi_{b_1}(y_j(\boldsymbol{w}^1)^T\boldsymbol{x}_j^{(i)}) = 0, \quad \text{where } r_{ij} = 0. \tag{266}$$

We will follow in the footsteps of update step 1 and seek to bound $(\boldsymbol{w}^2)^T\boldsymbol{w}^\star$. First, we define $\hat{\boldsymbol{w}}^2 = \boldsymbol{w}^1 - \eta_2\nabla_{\boldsymbol{w}}l(\boldsymbol{w}^1, b_1; S_2^m)$, and therefore $(\boldsymbol{w}^2)^T\boldsymbol{w}^\star = \frac{(\hat{\boldsymbol{w}}^2)^T\boldsymbol{w}^\star}{\|\hat{\boldsymbol{w}}^2\|}$. Now, we begin by evaluate the gradient of the empirical loss function with respect to $\boldsymbol{w}^1$,

$$\nabla_{\boldsymbol{w}^1} l(\boldsymbol{w}^1, b_1; S_2^m) = \frac{-1}{m} \sum_{j=1}^{m} 2 \left( y_j - \sum_{i=1}^{k} \phi_{b_1}((\boldsymbol{w}^1)^T \boldsymbol{x}_j^{(i)}) \right) \left( \sum_{i=1}^{k} \boldsymbol{x}_j^{(i)} \phi_{b_1}'((\boldsymbol{w}^1)^T \boldsymbol{x}_j^{(i)}) \right),$$
(267)

$$= \frac{-1}{m} \sum_{j=1}^{m} 2 \left( 1 - \sum_{i=1}^{k} \phi_{b_1}(y_j(\boldsymbol{w}^1)^T \boldsymbol{x}_j^{(i)}) \right) \left( \sum_{i=1}^{k} r_{ij} y_j \boldsymbol{x}_j^{(i)} \right)$$
(268)

where 268 follows from 265, 266. Define $\alpha_j := 1 - \sum_{i=1}^{k} \phi_{b_1}(y_j(\boldsymbol{w}^1)^T \boldsymbol{x}_j^{(i)})$, for $j \in [n] \setminus [m]$. Then, $1 \geq 1 - \frac{4}{100} \sqrt{\frac{kd}{(k+d)\ln(kd)}} \geq \alpha_j \geq 1 - 40 \sqrt{\frac{kd}{(k+d)\ln(kd)}}$. We also define $\boldsymbol{x}_j^{(t)}$ to be the patch of the $j$-th data sample that corresponds to the occurrence of the signal, that is $r_{tj} = 1$. From 268,

$$\nabla_{\boldsymbol{w}^1} l(\boldsymbol{w}^1, b_1; S_2^m) = \frac{-1}{m} \sum_{j=1}^{m} 2\alpha_j y_j \boldsymbol{x}_j^{(t)} \stackrel{d}{=} \frac{-1}{m} \sum_{j=1}^{m} 2\alpha_j \left( \boldsymbol{w}^\star + \sigma \boldsymbol{\epsilon}_j^{(t)} \right),$$
(269)

$$= -\sum_{j=1}^{m} \frac{2\alpha_j}{m} \boldsymbol{w}^\star - \frac{1}{m} \sum_{j=1}^{m} 2\sigma \alpha_j \boldsymbol{\epsilon}_j^{(t)},$$
(270)

$$\stackrel{d}{=} -\sum_{j=1}^{m} \frac{2\alpha_j}{m} \boldsymbol{w}^\star - \frac{2\sigma \sqrt{\sum_{j=1}^{m} \alpha_j^2}}{m} \hat{\boldsymbol{\epsilon}},$$
(271)

where $\hat{\boldsymbol{\epsilon}} \sim \mathcal{N}(\boldsymbol{0}, \mathbf{I}_d)$. We define $a := \sum_{j=1}^{m} \frac{2\alpha_j}{m}$. And, observe that from the concentration of the norm of a Gaussian random variable $\mathbb{P}[\|\hat{\boldsymbol{\epsilon}}\| \geq 6\sqrt{d}] \leq 2\exp(-\frac{36d}{2d}) \leq 10^{-6}$. With these results, we are now ready to bound $(\boldsymbol{w}^2)^T \boldsymbol{w}^\star = \frac{(\hat{\boldsymbol{w}}^2)^T \boldsymbol{w}^\star}{\|\hat{\boldsymbol{w}}^2\|}$,

$$\|\hat{\boldsymbol{w}}^2\| = \|\boldsymbol{w}^1 + \eta_2 \sum_{j=1}^{m} \frac{2\alpha_j}{m} \boldsymbol{w}^\star + \eta_2 \frac{2\sigma \sqrt{\sum_{j=1}^{m} \alpha_j^2}}{m} \hat{\boldsymbol{\epsilon}}\|,$$

$$\leq 1 + a\eta_2 \|\boldsymbol{w}^\star\| + \frac{2\sigma \eta_2}{\sqrt{m}} \|\hat{\boldsymbol{\epsilon}}\|,$$

$$\leq 1 + a\eta_2 + \frac{12\sigma \eta_2 \sqrt{d}}{\sqrt{\sigma^2(k+d)\ln(kd)}} \leq 1 + \eta_2(a + 10^{-3}),$$

for a large enough $k, d$. And now we lower bound $(\hat{\boldsymbol{w}}^2)^T \boldsymbol{w}^\star$,

$$(\hat{\boldsymbol{w}}^2)^T \boldsymbol{w}^\star = (\boldsymbol{w}^1)^T \boldsymbol{w}^\star + \eta_2 \sum_{j=1}^{m} \frac{2\alpha_j}{m} (\boldsymbol{w}^\star)^T \boldsymbol{w}^\star + \eta_2 \frac{2\sigma \sqrt{\sum_{j=1}^{m} (\alpha_j^2)}}{m} \hat{\boldsymbol{\epsilon}}^T \boldsymbol{w}^\star$$
(272)

$$\stackrel{d}{=} -(\boldsymbol{w}^1)^T \boldsymbol{w}^\star + \eta_2 \frac{\sum_{j=1}^{m} 2\alpha_j}{m} + \eta_2 \frac{2\sigma \sqrt{\sum_{j=1}^{m} (\alpha_j^2)}}{m} \epsilon; \ \epsilon \in \mathcal{N}(0,1),$$
(273)

$$\geq -1 + \eta_2 a - \frac{2\eta_2 \sigma \epsilon}{\sqrt{m}}$$
(274)

$$\geq -1 + \eta_2 a - \eta_2 \frac{2\sigma \epsilon}{\sqrt{\sigma^2(k+d)\ln(kd)}} \geq -1 + \eta_2(a - 10^{-3}),$$
(275)

where we have used the fact that $\mathbb{P}[|\frac{2\epsilon}{\sqrt{(k+d)\ln(kd)}}| \geq 10^{-3}]$ holds with probability $\leq 10^{-6}$, for a large enough $k, d$. Therefore the alignment can be lower bounded as,

$$\frac{(\hat{\boldsymbol{w}}^2)^T \boldsymbol{w}^\star}{\|\hat{\boldsymbol{w}}^2\|} \geq \frac{1 + 10^3(a + 10^{-3})}{1 + 10^3(a + 10^{-3})} \geq 0.96,$$
(276)

as $1 \geq a \geq \frac{2}{3}$ for large $k, d$, and this occurs with a probability $\geq 1 - 2 \times 10^{-5}$.

3c. CNN has Low Risk

We now show that this large constant alignment guarantees a low risk. We bound the push forward of the noise through the CNN,

$$|\sigma \max_{j \in [m]} ((\boldsymbol{w}^2)^T \boldsymbol{\epsilon}_i)| \leq \frac{6\sqrt{\ln(k)}}{100\sqrt{k \ln(kd)^3}} \leq 10^{-4}$$
(277)

To derive the last inequality, we have used the concentration of the maximum of the absolute value of $k$ i.i.d. Gaussian random variables, $\mathbb{P}[\max_{j \in [k]} |(\boldsymbol{w}^2)^T \boldsymbol{\epsilon}_j| \geq \sqrt{32 \ln(k)}] \leq \frac{2}{m^9} \leq 10^{-6}$. For this data sample, let $t \in [k]$ be the index of the signal patch, then,

$$\phi_{b_2}(y_j (\boldsymbol{w}_i^2)^T \boldsymbol{x}_j^{(t)}) \geq 0.959, \tag{278}$$

$$\phi_{b_2}(y_j (\boldsymbol{w}_i^2)^T \boldsymbol{x}_j^{(i)}) = 0, \ \forall \, i \neq t. \tag{279}$$

To bound the risk in the failure case, we note that for any $\boldsymbol{v} \in \mathcal{W}$,

$$\mathbb{E}[(y - \sum_{i=1}^{k} \phi_b(\boldsymbol{w}^T \boldsymbol{x}^{(i)}))^2] = \mathbb{E}[(1 - \sum_{i=1}^{k} \phi_b(y \boldsymbol{w}^T \boldsymbol{x}^{(i)}))^2], \tag{280}$$

$$= \frac{1}{k} \sum_{j=1}^{k} \mathop{\mathbb{E}}_{(\boldsymbol{x},y) \sim \mathrm{SSD}_j} [(1 - \sum_{i=1}^{k} \phi_b(y \boldsymbol{w}^T \boldsymbol{x}^{(i)}))^2], \tag{281}$$

$$= \frac{1}{k} \sum_{j=1}^{k} \mathbb{E}[(1 - \sum_{i \neq j} \phi_b(\sigma \epsilon_{ij}) - \phi_b(\cos(\alpha_j) + \sigma \epsilon_{jj}))^2], \tag{282}$$

To evaluate the above expression, we observe that the expectation can be written as,

$$\mathbb{E}[(1 - \sum_{i \neq j} \phi_b(\sigma \epsilon_{ij}) - \phi_b(\cos(\alpha_j) + \sigma \epsilon_{jj}))^2] = \mathrm{Var}[(1 - \sum_{i \neq j} \phi_b(\sigma \epsilon_{ij}) - \phi_b(\cos(\alpha_j) + \sigma \epsilon_{jj}))]$$
$$+ \mathbb{E}[(1 - \sum_{i \neq j} \phi_b(\sigma \epsilon_{ij}) - \phi_b(\cos(\alpha_j) + \sigma \epsilon_{jj}))]^2, \tag{283}$$

$$= \sum_{i \neq j} \mathrm{Var}[\phi_b(\sigma \epsilon_{ij})] + \mathrm{Var}[\phi_b(\cos(\alpha_j) + \sigma \epsilon_{jj})] + (\mathbb{E}[\phi_b(\cos(\alpha_j) + \sigma \epsilon_{jj})])^2, \tag{284}$$

$$\leq \sum_i \sigma^2 + \cos^2(\alpha_j) \leq k \sigma^2 + 1 \leq 2. \tag{285}$$

Substituting this back,

$$\mathbb{E}[(y - \sum_{i=1}^{k} \phi_b(\boldsymbol{w}_i^T \boldsymbol{x}^{(i)}))^2] \leq \frac{1}{k} \sum_{j=1}^{k} 2 = 2 \tag{286}$$

Finally, the risk of the classifier is,

$$\mathbb{E}\left[ R\left( \bar{\theta}_n, P \right) \right] \leq (1 - 0.959)^2 + 4 \times 10^{-5} \leq \delta. \tag{287}$$

$\square$

# E  EXPERIMENTS

In this section, we validate our theoretical bounds with empirical results. We begin by presenting the test-error experiments, where we evaluate the test error of the three models across various training sample sizes. The results for these experiments show an order-of-magnitude decrease in the sample efficiency when comparing CNNs to LCNs, and comparing LCNs to FCNs.

We then present our sample complexity experiments, wherein we explicitly calculate the sample complexity of CNNs and LCNs for various $(k, d)$ pairs. However, these experiments are significantly more compute-intensive than the test error experiments. While the computational demands are manageable for CNNs, they increase significantly for LCNs and become prohibitively large for FCNs. This is primarily because FCNs require at least 10-20 times more samples than LCNs. Nonetheless, for both CNNs and LCNs, we successfully verify that the empirical sample complexity satisfies the respective theoretical bounds. Specifically, for CNNs, we show a $O(k)$ sample complexity growth with a fixed $d$ and a $O(d)$ growth with a fixed $k$. For LCNs, we establish that the sample complexity grows as $O(k^2), \Omega(k)$ with a fixed $d$ and as $\Theta(d)$ with a fixed $k$.

## E.1  TEST ERROR EXPERIMENTS

In this experiment, we evaluate the test error of each of the three models when trained with a sample size of $\{10, 50, 100, 250, 500\}$ for every $(k, d)$ pair with $k, d \in \{10, 20\}$. For each training session, we conduct a grid search over the learning rates for patch parameters being $\{10^{-1}, 10^{-2}, 10^{-3}\}$, and for the biases being $\{10^{-2}, 10^{-3}, 10^{-4}\}$. We choose the model with the lowest test error. The experiment is replicated 5 times, and we report the mean and standard deviation of the test errors.

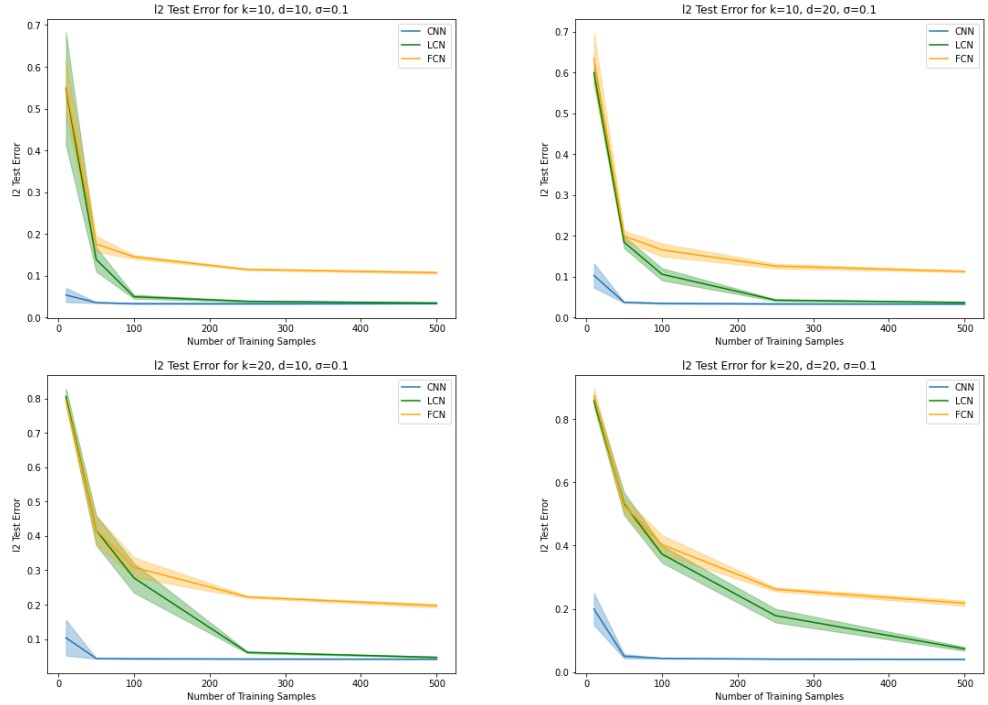

Figure 1: Test error incurred by CNNs, LCNs and FCNs for various values of $(k, d)$

Across all $(k, d)$ pairs we observe that LCNs require an order-of-magnitude (10-20 times) more samples than CNNs to achieve comparable test errors. This demonstrates the larger sample efficiency of CNNs over LCNs. Extrapolating the trend line for FCNs, it is evident that they would need even orders-of-magnitude more samples than LCNs for comparable error levels. These observations are consistent with our theoretical predictions of sample complexities: $\Omega(k^2 d)$ for FCNs, $O(k(k + d))$ and $\Omega(kd)$ for LCNs, and $O(k + d)$ for CNNs.

### E.2 SAMPLE COMPLEXITY EXPERIMENTS

In our first experiment, we fix the patch dimension $d$ at 20 and vary the number of patches $k$ across the range $\{10, 15, 20, 25, 30\}$. For each $(k, d)$ pair, we plot the sample complexity for both CNNs and LCNs. We evaluate the sample complexity via the following steps:

1. Target Loss Evaluation: We compute the optimal loss based on the ground truth and add a fixed tolerance of $0.03$ to establish the target loss.

2. Determining Sample Range: Through trial and error, we determine that a maximum of 1000 samples is sufficient for any model across all $k$ values.

3. Binary Search Method: To find the minimum number of samples required to reach the target loss, we perform a binary search. In each step, we conduct a grid search over the learning rates for weights being $[10^{-1}, 10^{-2}, 10^{-3}]$ and biases being $[10^{-2}, 10^{-3}, 10^{-4}]$, and select the model with the lowest test error.

4. Repetitions for Reliability: We repeat the steps (1-3) five times, plotting the mean and standard deviation of the sample complexities.

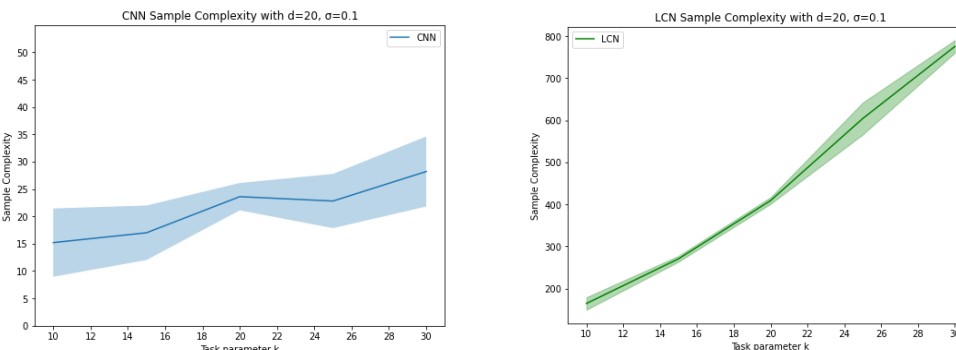

Figure 2: Sample complexity for CNNs (left) and LCNs (right) across various values of $k$

For a fixed $d$, the sample complexity for CNNs exhibits an $O(k)$ growth as (Figure 3, left), which is consistent with our CNN upper bound. Similarly, for LCNs, the complexity growth is consistent with our theoretical results of $O(k^2)$ and $\Omega(k)$ (Figure 3, right). Additionally, note that LCNs require about 10 to 20 times more samples than CNNs, which corresponds to the multiplicative $d$ factor in LCNs' sample complexity bound.

In our second experiment, we set the number of patches $k$ at 20 and vary the patch dimension $d$ across the range $[10, 15, 20, 25, 30]$. The same steps (1-4) are repeated for this setup.

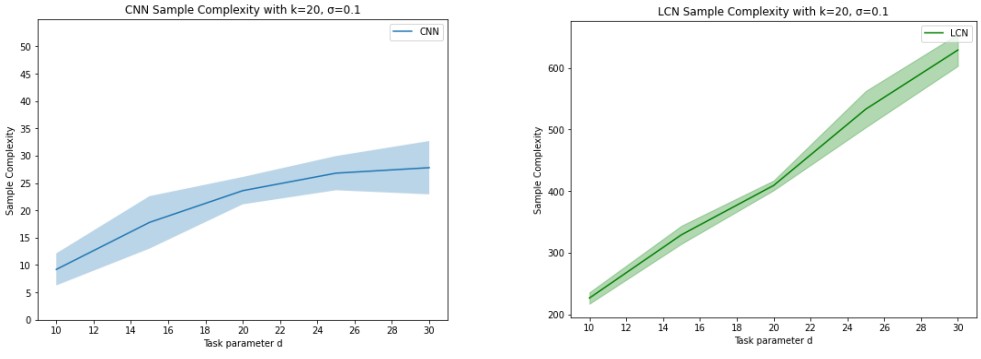

Figure 3: Sample complexity for CNNs (left) and LCNs (right) across various values of $d$

For a fixed $k$, we observe that the CNN sample complexity grows as $O(d)$ (Figure 4, left), and the LCN sample complexity grows as $\Theta(d)$ (Figure 4, right), both in line with our theoretical guarantees. Furthermore, akin to our findings in the first experiment, LCNs require approximately 20 times more samples than CNNs, owing to the multiplicative $k$ factor in their sample complexity.

