# OpenReview forum: "Role of Locality and Weight Sharing in Image-Based Tasks: A Sample Complexity Separation between CNNs, LCNs, and FCNs"
_ICLR.cc/2024/Conference — ICLR 2024 spotlight_

### Official Review · Reviewer_o7GU · 2023-10-18

**Soundness:** 3 good
**Presentation:** 3 good
**Contribution:** 2 fair
**Rating:** 8
**Confidence:** 5

**Summary:**

In this research, they investigate the superior performance of Convolutional Neural Networks (CNNs) in vision-centric tasks in comparison to Locally Connected Networks (LCNs) and Fully Connected Networks (FCNs). They introduce the concept of Dynamic Signal Distribution (DSD), an intricate data model devised to encapsulate inherent properties observed in real-world imagery, notably locality and translation invariance. Within the framework of this study, they establish a theoretical sample complexity separation between CNNs, LCNs, and FCNs, taking into account optimization considerations.

**Strengths:**

This paper is well-written and provides very supportive technical details. The authors introduce an interesting data model called DSD to represent real-world vision tasks. This task is straightforward and simple, making it suitable for theoretical analysis. Additionally, this task exhibits characteristics of locality and translation invariance.

**Weaknesses:**

Primary Concerns:

Novelty. My main reservation pertains to the novelty made over the work by Wang & Wu (2023). While I concede that the task depicted here is more realistic than that in Wang & Wu (2023), it appears the core insights remain consistent: FCNs are equivalent to global rotation groups, whereas LCNs relate to local rotation/permutation groups. Although the authors consider optimization in the upper bound analysis for LCNs and CNNs, the chosen algorithm, which has just two training iterations, seems quite unconventional and perhaps too simplistic. I'm uncertain whether this presents an improvement relative to the ERM analysis. Regarding the lower bound section, if I interpret it correctly, does the novelty lie mostly in the technical novelty? It appears that the primary objective is to determine the size of the enlarged function class, when the original task is enlarged by the global/local rotation group. I question the significance of the specific method used for this calculation. My primary concern remains to be the novelty over Wang & Wu (2023). If clarity on this point is provided, I'm inclined to upgrade my rating to at least 6. Moreover, it would be beneficial if the paper could elaborate more on the contributions of Wang & Wu (2023) and comparisons with Wang & Wu (2023). For instance, when stating on page 2, "Wang & Wu (2023) extended this line of work to show a separation between FCNs, LCNs and CNNs. However, that work suffer from the same drawbacks as Li et al. (2021).", a more explicit discussion on the advancements and remaining problems in Wang & Wu (2023) would be useful.

Secondary Concerns:

1. Regarding the model definitions, what is the reason for limiting the norm of $w_i$ to less than one? This seems implausible.
2. What necessitates the width of your FCN models to be defined as $k$? Shouldn't the lower bound be applicable to any width, at least intuitively?
3. Concerning the proof outline for theorem 6.1, in its first step, could there be an inclusion of a more comprehensive technical elaboration? While some readers might be able to grasp its logic intuitively, offering a detailed technical justification, especially given its novel approach, would be beneficial.

**Questions:**

No.

---

> ### Author Response · Authors · 2023-11-16
>
> We thank the reviewer for their thorough and insightful comments, and we address the issues raised below.
>
> **Comment 1: Novelty in our work over Wang et al.**
>
> Answer:
> We agree with the reviewer's observation on our use of global equivariance for FCNs, as in Li et al. and Wang et al., and local equivariance for LCNs, following Wang et al. Our work, however, extends these techniques to a more realistic task, which yields justifications for colloquial intuitions and practical insights. Moreover, given the non-rotationally invariant nature of our task, our novel analysis diverges from standard information-theoretic approaches used in Li et al. and Wang et al. Additionally, our use of gradient descent offers important theoretical advantages over ERM analysis: it establishes a separation between computationally-efficient equivariant algorithms and it circumvents the potential non-equivariance issues in ERM. We elaborate on these aspects below.
>
>
> *1. Realistic task and practical insights*
>
> We reemphasize the realistic nature of the task, mirroring real-world image classification, where outputs are determined by local signals embedded within uninformative noise. Specifically, we represent the label-determining signals by $\pm \mathbf{w}^\star \in \mathbb{R}^d$ against a backdrop of i.i.d Gaussian noise with variance $\sigma$, and the $k$ patches denote the potential areas for the signal to translate.
>
> In this context, our analysis rigorously brings forth the insight that FCNs incur a threefold cost: a multiplicative factor of $k$ due to the need of independently learning $k$ patches, a cost of $k$ for isolating each patch, and a cost of $d$ for learning the sparse signal. LCNs avoid the isolation cost, while CNNs only incur a cost of $d$ to learn the sparse signal.
>
> Our analysis offers another insight in the setting where the signal may only appear in a smaller subset of $\tilde{k} < k$ patches. This implies that the label-determining signal is "less mobile". Our method can be readily extended to show that the sample complexity for FCNs, LCNs, and CNNs scale approximately as $\tilde{k}kd$, $\tilde{k}d$, and $d$ respectively. As $\tilde{k}$ nears 1, the sample complexity of LCNs and CNNs converges, and the sample complexity of FCNs decreases, even though the number of trainable parameters remain unchanged.
>
> Furthermore, we can also infer from our bounds that when $d$ is relatively large as compared to $k$, that is if the signal is not "very sparse", the advantages of CNNs and LCNs over FNNs become less pronounced. Additionally, our analysis proves the intuitve fact that lower noise levels uniformly enhance the sample efficiency across all models.
>
> In contrast, the tasks defined in Li et al., $f(\mathbf{x}) = \sum_{i=1}^d x_i^2 - \sum_{i=d+1}^{2d} x_i^2$, where the input is sampled as $\mathbf{x} \sim \mathcal{N}(\mathbf{0}, \mathbf{I_{2d}})$ and in Wang et al. $g(\mathbf{x}) = (\sum_{i=1}^d x_{2i}^2 - x_{2i+1}^2) (\sum_{i=d+1}^{2d} x_{2i}^2 - x_{2i+1}^2)$, where the input is sampled as $\mathbf{x} \sim \mathcal{N}(\mathbf{0}, \mathbf{I_{4d}})$, do not possess the necessary structure to yield the insights mentioned above. Specifically, they cannot show how sample complexity changes with varying the degree of locality and translation invariance in the input, nor can they lay out conditions on the input under which conditions differences between CNNs, LCNs, and FCNs are observed or absent.
>
> Furthermore, as highlighted in our paper, the workhorse for the FCN lower bound in both Li et al. (Lemma D.2) and Wang et al. (Appendix H.2) is the quadratic interaction between the first $d$ coordinates and the subsequent $d$ coordinates. This signifies an interaction phenomenon in their tasks, which is distinct from what we capture in our task. While this interaction is an interesting phenomenon, it is not the primary characteristic of locality and translation invariance found in images. Nonetheless, we view incorporating signal-signal interaction into our framework represents a promising avenue for future research.
>
> *2. Technical challenges and significance of our analysis*
>
> Our approach diverges from the approaches used in Li et al. and Wang et al. because the marginal over $\mathcal{X}$ is not rotationally invariant. Specifically, this deviation precludes the use of Benedek Itai style bounds from Li et al., nor do we enjoy the semi-metricness of $l_2$ or $0 \text{-} 1$ loss under an invariant distribution. Recognizing this, Li et al. have previously [noted](https://openreview.net/forum?id=uCY5MuAxcxU&noteId=eEtVx0qv-36) the need for new proof techniques for such distributions as an area for future exploration and our work takes the first step in this direction.

---

> ### Author Response · Authors · 2023-11-16
>
> To address the above challenge, we developed a novel reduction technique that leveraged input-independent randomization and equivariance to show that both FCNs and LCNs must learn each patch independently from every other patch. This technique not only aids in computation, but also provides a theoretical justification for the intuition for the '$k$ cost' associated with learning each patch independently. Despite the fact that our reduced problem is relatively simpler, it still does not enjoy the semi-metricness on the $l_2$ loss function. To overcome this issue, we introduce a novel variant of Fano's Lemma for input-independent randomized algorithms. This allows to derive the lower bounds without requiring semi-metricness to hold across the entire space.
>
> We respectfully disagree with the reviewer's perspective on the significance of our methods. First, our reduction technique through randomization alongwith the tailored variant of Fano's Lemma have broad applicability, and we believe it should be of independent mathematical interest. Additionally, given the widespread use of Gaussian Mixture Models (GMMs), our techniques could offer valuable tools for researchers in this area. Lastly, our techniques could serve as a foundation for future extensions of our work, such as incorporating the aforementioned interaction effects.
>
> *3. Benefits of gradient descent analysis over ERM*
>
> We acknowledge the reviewer's observation regarding the simplicity of our gradient descent analysis on a one-hidden layer neural network with two iterations. However, our choice of gradient descent over an ERM-style argument was deliberate because of two key theoretical reasons:
>
> (1) The gradient descent demonstrates a sample complexity separation between FCNs, LCNs, and CNNs for computationally-efficient (polynomial time) equivariant algorithms. This distinction is crucial because while a separation may exist for computationally inefficient algorithms, the separation might disappear under constraints of computational efficiency. Given that models in practice are trained using efficient algorithms, we believe establishing a separation for efficient algorithms is theoretically important.
>
> (2) It is important to ensure that both the upper and lower bounds are derived for equivariant algorithms. This is because having lower bounds for equivariant algorithms and upper bounds for non-equivariant ones does not lead to a valid sample complexity separation as non-equivariant algorithms could potentially be more sample-efficient than their equivariant counterparts. Since, the equivariance of ERM is not clearly established and Wang et al. do not discuss this aspect in their work, we believe a gradient descent analysis is more appropriate.
>
> **Comment 2: Shouldn't the current FCN lower bound hold for any width? Why is the width of FCN set to $k$?**
>
> Answer:
> Our current lower bound may not trivially hold for any width. This is because, unlike the tasks in Li et al. and Wang et al., the lack of rotational invariance of the marginal over $\mathcal{X}$ means that we cannot assume that the $l_2$ loss function is a semimetric on the space of all functions. This limitation prevents us from applying standard Fano's Lemma as in Wang et al. Consequently, we developed a variant of Fano's Lemma, that only requires the semimetric property to hold when two out of three functions are from the set of "hard instances". We had to explicitly establish this semi-metric property for the underlying FCN function class with width equal to $k$. Therefore, we cannot immediately say that the lower bound holds for all widths.
>
> We set the width of the FCNs to $k$ so that like LCNs and CNNs, FCNs could also
> benefit from the denoising capability of the $k$ non-linear filters.
>
> **Comment 3: Why is the norm of $\mathbf{w}_i \le 1$?**
>
> Answer:
> We imposed the constraint $\mathbf{w}_i \le 1$ primarily to simplify our analysis. We use it to prove the technical requirement of our Fano's Lemma which is that the semimetric property should hold when two out of three functions belong to the set of 'hard instances'. We particularly selected the constant $1$ to enable the filters to capture the unit norm signal $\mathbf{w}^\star$. While this choice might appear somewhat artificial, it does not impact the main takeways of our theory.

---

> ### Author Response · Authors · 2023-11-16
>
> **Comment 4: Can the proof outline for step 1 of theorem 6.1 be elaborated more technically?**
>
> Answer:
> We appreciate the reviewer's request and we will definitely include a more technical explanation in our revised draft. We now outline step 1 of the proof, focusing on the key aspects while avoiding overly intricate details for better understanding.
>
> We denote the DSD distrbution by $P$ and its marginal for the $i^{\text{th}}$ patch as $Q_i$. We denote the mean of the positive marginal of each $Q_i$ as $\mathbf{\mu}_i$. The proof begins by noting that the risk of the algorithm, $\bar{\theta}_n$, on $\mathbf{U} \circ P$ is given by the average of the risk of $\bar{\theta}_n$ over each $\mathbf{U} \circ Q_i$. We then use equivariance to show that the risk of $\bar{\theta}_n$ over each $\mathbf{U} \circ Q_i$ is exactly the same,
>
> $$\mathbb{E}\_{S^n \sim (\mathbf{U} \circ P)^n} \left[
>         R\left(
>             \bar{\theta}\_n,
>             \mathbf{U} \circ Q\_i
>         \right)
>     \right]
>     =
>     \mathbb{E}\_{S^n \sim (\mathbf{U} \circ P)^n}
>     \left[
>         R\left(
>             \bar{\theta}\_n,
>             \mathbf{U} \circ Q\_j
>         \right)
>     \right].
> $$
>
> This implies that even though we are "training" the model using $\mathbf{U} \circ P$, it is enough to "test" the model on $\mathbf{U} \circ Q_1$. Since this holds for all $\mathbf{U}$ and $\bar{\theta}_n$,
>
> $$\mathbb{E}\_{S^n \sim P^n} \left[
>         R\left(
>             \bar{\theta}\_n,
>             P
>         \right)
>     \right]
>     \ge
>     \inf\_{\bar{\theta}\_n}
>     \sup\_{\mathbf{U}}
>     \mathbb{E}\_{S^n}
>     \left[
>         R\left(
>             \bar{\theta}\_n,
>             \mathbf{U} \circ Q\_1
>         \right)
>     \right].
> $$
>
> We denote $R(
>             \bar{\theta}_n,
>             \mathbf{U} \circ Q_1
>         ) = R_1(
>             \bar{\theta}_n
>         )$ for brevity. We now begin a series of reductions, first observe that the minimax risk can only decrease if the algorithm was instead supplied with patch separated input data $\{S^{n_i} \sim \mathbf{U} \circ Q_i\}$. This is because the algorithm can simply ignore the patch-id, which is strictly extra information. Please note that with high probability $n_i \approx n/k$,
>
> $$
>     \ge
>     \inf\_{\bar{\theta}\_n}
>     \sup\_{\mathbf{U}}
>     \mathbb{E}\_{\{S^{n\_i}\}}
>     [
>         R_1(
>             \bar{\theta}\_n
>         )
>     ].
> $$
>
> Next, we provide the algorithm with $S^{n_1}$ and all the required quantities so that it can internally construct $S^{n_2},..S^{n_k}$. It is easy to see that this can only reduce the minimax risk. These quantities include the means $\mathbf{U}\mathbf{\mu}_2, .., \mathbf{U}\mathbf{\mu}_k$ and presampled gaussian noise vector, label vector, and $n_2,..,n_k$. We refer to the presampled quantities by $\xi$.
>
> $$
>     \ge
>     \inf\_{\bar{\theta}\_n}
>     \sup\_{\mathbf{U}}
>     \mathbb{E}\_{\{S^{n\_1}, \xi\}}
>     [
>         R\_1(
>             \bar{\theta}\_n(\{\mathbf{U}\mathbf{\mu}_2, .., \mathbf{U}\mathbf{\mu}_k\})
>         )
>     ].
> $$
>
> The final step is to note that since $\mathbf{U}\mathbf{\mu}_1, .., \mathbf{U}\mathbf{\mu}_k$ are mutually orthogonal, the only information about $\mathbf{U}\mathbf{\mu}_1$ we get from $\mathbf{U}\mathbf{\mu}_1, .., \mathbf{U}\mathbf{\mu}_k$ is that the former lies in a subspace of size $kd-k+1 \approx kd$. This information is too little and can effectively be ignored,
>
> $$
>     \ge
>     \inf\_{\bar{\theta}\_n}
>     \sup\_{\mathbf{U}}
>     \mathbb{E}\_{S^{n_1}}
>     [
>         R\_1(
>             \bar{\theta}\_n
>         )
>     ].
> $$
>
> We have now reduced learning $\mathbf{U} \circ P$ with $n$ samples to learning $\mathbf{U} \circ Q_1$ with $n/k$ samples. This marks the end of Step 1. In step 2, we show that $n/k \approx \sigma^2kd$, which proves the result.
>
> ---
>
> We sincerely hope that our responses provided clarity on the points raised by the reviewer, particularly regarding the novelty and significance of our work.

---

> ### Comment · Reviewer_o7GU · 2023-11-16
> **Response**
>
> This addresses my concerns! I will accordingly raise my score to 8. I suggest that the authors can add more detailed and more explicit comparisons over Wang & Wu and Li et al. in this paper.

---

> > ### Author Response · Authors · 2023-11-16
> >
> > We sincerely thank the reviewer for their quick response and we would definitely include a more detailed comparison over Wang & Wu and Li et al.

---

### Official Review · Reviewer_jtyo · 2023-10-30

**Soundness:** 3 good
**Presentation:** 3 good
**Contribution:** 2 fair
**Rating:** 8
**Confidence:** 3

**Summary:**

The authors study the problem of sample complexity for different classes of models based on analyzing an equivariant gradient descent type algorithm.

**Strengths:**

- The relation between sample complexity and the type of network is interesting.
- There seems to be real improvements over previous works.

**Weaknesses:**

The only downside I see to this work is the fact that there is no empirical result which supports the theoretic result. I myself cannot find any issue or criticism with the theory that was presented, but the complexity and abtractness of the setting and the numerous variables involved makes it difficult to find any error in the derivation if one exists.

Therefore, it seems that there needs to be some empirical result of the predicted complexity and an actual observation rather than just a standalone theory with no phyisical evidence to support it. It seems that the problem setup is precise enough to run small experiments on the linear models outlined in section 4.2.

For example, for a fixed amount of $sigma$ and $d$, each linear model can be trained on an increasing number of classes to show the empirical observation of the $k$ term. Similarly, this can also be shown with a fixed $sigma$ and $k$. This is just the first example of what came to mind, but the authors may indeed be able to come up with an even better experiment given their study of the problem.

If this can be added, I would be happy to raise my score.

## Minor:

After the conclusion, there is a QED symbol which is likely a typo and should have appeared after the previous section.

**Questions:**

- I am curious if transformers can be described by FCN or LCN, or if they need a totally different treatment. I think this would be an interesting discussion point to add if there is any insight because the sample complexity for transformers is a relevant and interesting topic.

---

> ### Author Response · Authors · 2023-11-21
>
> We thank the reviewer for their valuable comments and we appreciate their insightful feedback. We now take this opportunity to address their queries below.
>
>
> **Comment 1: Can experiments be added that validate the analysis?**
>
> Answer:
> We completely agree with the reviewer's suggestion that empirical validation would strengthen the credibility of our theoretical analysis. We request the reviewer to refer to the newly added supplementary material, which includes the detailed experimental protocol and results. We also plan to incorporate them in the forthcoming revision.
>
> **Comment 2: Can our framework also provide bounds for transformers, given their prevalance in machine learning?**
>
> Answer:
> Regarding the potential application of our framework to transformers, we can consider a transformer architecture where each input patch corresponds to an input token. If we assume that the keys $W_K$, queries $W_Q$, values $W_V$ matrices are initialized using standard isotropic gaussian distribution, then this transformer, trained with SGD, is equivariant to the application of the same $U \in \mathcal{O}(d)$ to each patch of the input. Ignoring the issues of semi-metricness, this leads to a $\Omega(\sigma^2d)$ lower bound. The upper bound, however, may be a little challenging as it requires gradient descent analysis for the softmax-based attention mechanism. Nonetheless, we fully agree with the reviewer that this is an important direction for future work.
>
> **Comment 3: Misplaced QED symbol after the conclusion**
>
> Answer:
> We thank the reviewer for pointing out the typo, and we will fix in the next revision.
>
>
> We sincerely hope that our responses clarified the questions raised by the reviewer.

---

> > ### Author Response · Authors · 2023-11-22
> >
> > Respected Reviewer jtyo,
> >
> > We would like to express our gratitude once more for your insightful comments on our work. Following your valuable suggestion, we have added experimental results to validate the theoretical bounds in our paper. We sincerely hope you might consider a reevaluation of the score if we have successfully addressed your concerns. We are very much looking forward to addressing any further questions that you may have throughout the remainder of the discussion period.
> >
> > Best Regards,
> > Authors

---

> > > ### Comment · Reviewer_jtyo · 2023-11-23
> > > **Thank you**
> > >
> > > Thank you for adding the requested experiments. I have updated my score.

---

### Official Review · Reviewer_yeJ4 · 2023-10-31

**Soundness:** 3 good
**Presentation:** 3 good
**Contribution:** 3 good
**Rating:** 6
**Confidence:** 3

**Summary:**

The authors proof that CNNs has a better sample complexity than LCNs, and LCNs has a better sample complexity than FCNs under the framework of Dynamic Signal Distribution.

**Strengths:**

Theoretical proof is provided regarding the sample complexity of FCNs, LCNs, and CNNs. The theoretical proof is built upon several tools like Dynamic Signal Distribution (DSD), network architectures given DSD, equivariant algorithms, minimax risk, etc. Overall speaking, the paper is well-organized and easy to follow even for general audience.

**Weaknesses:**

1. The Local Signal Adaptivity (LSA) activation function is quite different compared to popular used activation function, like ReLU or more recent activation function. A better justification of using LSA could be provided.
2. From the proof sketch in section 6 and section 7, it seems that the sample complexity is dependent on what is the learning setting. For example, in section 6, 'we establish that learning U ◦ DSD with m samples requires learning k "nearly independent" subtasks,' and in section 7, 'we establish that learning U ◦ DSD with m samples requires learning k independent subtasks.' Dose this result in the difference between the sample complexity of LCNs in section 6 and 7? If this is the case, I think authors could add more analysis regarding how the results will change given the learning setting of 'U ◦ DSD', In addition, is there a setting where the sample complexity of FCNs, LCNs and CNNs is similar? I hope authors could provide more insights regarding this.

**Questions:**

Can the current framework support ReLU like activation functions?

---

> ### Author Response · Authors · 2023-11-16
>
> We are grateful for the reviewer's positive feedback and we address their questions below.
>
> **Comment 1: Can the framework be extended to ReLU activation function, given its popularity over LSA activation function?**
>
> Answer:
> We note that the LSA activation function can be written as the sum of two ReLU activation functions as $\phi_b(x) = \text{ReLU}(x-b) - \text{ReLU}(-x-b)$. This decomposition allows us to adapt our framework for ReLU through the following architectural modifications:
> 1. We augment an additional hidden node for each patch in LCNs and CNNs and additional $k$ hidden nodes for FNNs.
> 2. We include a bias term for each patch.
> 3. We fix the second layer to $\{-1,1,..,-1,1\} \in \mathbb{R}^{2d}$
>
> The idea is that the two hidden nodes corresponding to each patch learn $\mathbf{w}^\star$ and $-\mathbf{w}^\star$ respectively. Together with the bias and the fixed $\pm 1$ values from the second layer, we can simulate the thresholding effect of the LSA activation function.
>
> We also note that the LSA activation function, also known as the "soft-thresholding function", is extensively used in high-dimensional sparse recovery problems (Section 18.2 [1]). Since our task involves recovering the sparse mean vector $\mathbf{w}^\star$, it justifies our use of the LSA activation function.
>
> **Comment 2: Why is the sample complexity of LCNs different in section 6 and 7? Is this due to learning $k$ "nearly-independent" and $k$ "independent" subtasks respectively?**
>
> Answer:
> Our conjecture is that the difference in the sample complexity for LCN upper bound of $O(k(k+d))$ and the LCN lower bound of $\Omega(kd)$, stems primarily from the computational efficiency of the algorithms employed. The upper bound is derived for a computationally efficient (polynomial time) equivariant gradient descent style algorithm. In contrast, the lower bound is derived from minimax principles and is applicable to both computationally efficient and inefficient equivariant algorithms. It is generally the case that computationally inefficient algorithms have lower sample complexity as compared to their efficient counterparts, which possibly explains the difference in sample complexity.
>
> **Comment 3: Are there settings where the sample complexity of FCNs, LCNs and CNNs are similar?**
>
> Answer:
> There are indeed scenarios where the sample complexities of FCNs, LCNs, and CNNs converge. These include:
> 1. Consider the setting where the signal can only be present in a smaller subset of $\tilde{k} < k$ patches, implying that the label-determining signal is "less mobile". In this case, our analysis can be readily extended to show that the sample complexity for FCNs, LCNs, and CNNs scale approximately as $\tilde{k}kd$, $\tilde{k}d$, and $d$ respectively. Therefore, as $\tilde{k}$ approaches 1, LCNs and CNNs will perform similarly, and FCNs' sample complexity will decrease, despite unchanged number of trainable parameters.
> 2. In the setting when $d$ is relatively large as compared to $k$, that is if the signal is not too-sparse, our bounds show that the advantages of CNNs and LCNs over FNNs become less pronounced. For instance, if we choose $k$ to be a constant $c$, then the sample complexity of FCNs, LCNs and CNNs scales as $c^2 d, cd, d$ respectively and therefore the three models have similar sample efficiency.
>
> ---
>
> We hope that our explanations addressed the reviewer's inquiries comprehensively.
>
>
> Bibliography
>
> [1] Hastie, Trevor, et al. The elements of statistical learning: data mining, inference, and prediction. Vol. 2. New York: springer, 2009.

---

> > ### Author Response · Authors · 2023-11-22
> >
> > Dear Reviewer yeJ4,
> >
> > We would like to sincerely thank you for your valuable comments to help improve our work. We genuinely hope that our responses have effectively addressed your queries. We are also happy and willing to answer any further questions you may have, or provide any additional details that you may need.
> >
> > Warm regards,
> > Authors

---

### Official Review · Reviewer_LSto · 2023-11-01

**Soundness:** 4 excellent
**Presentation:** 4 excellent
**Contribution:** 3 good
**Rating:** 8
**Confidence:** 3

**Summary:**

This work provides a theoretical analysis of the sample complexity of Convolutional Networks (CNNs), Local Connected Networks (LCNs), and Fully-Connected Networks (FCNs). This paper claims that CNNs are more sample efficient than LCNs, which in turn are more sample efficient than FCNs when the data has implicit locality and translation invariance properties, as is the case with visual data. The authors prove lower bounds on the sample complexity of FCNs and LCNs, and similarly demonstrate upper bounds on the sample complexities of LCNs and CNNs to draw firm conclusions under the assumptions made in their theory.

To carry out the analysis, the paper proposed a toy data model inspired from the concepts of locality and translation invariance in natural images, called the Dynamic Signal Distribution (DSD) task. Here the input is comprised of k consecutive patches of dimension d, and one of the k patches is randomly filled with a noisy signed signal, while remaining patches are filled with isotropic Gaussian noise. A binary label for the input is determined from the sign of the signal.

**Strengths:**

### Significance
* While the results agree with empirical intuition that inductive biases in the model reflecting properties of the data should improve sample complexity, it is interesting to see such a result made crisp for vision models, and with specific focus on the locality and translation invariance properties of visual data.

### Clarity
* The analysis is non-trivial and requires heavy notation; the paper does a good job at walking the reader through the intuition, proofs, claims, and findings, however much of the theoretical development is relegated to the appendix.

### Originality
* The authors theoretical analysis is quite unique in my opinion. To the best of my knowledge, the proposed DSD model is novel, and the derived lower bounds are enabled by taking the learning algorithm (gradient descent) into account. This is necessary since a learning algorithm can simulate CNNs with LCNs and FCNs. While previous works have established a sample complexity separation when assuming training with gradient descent, they used a data model which did not seek to capture translation invariance and locality.

**Weaknesses:**

The main weakness in my opinion is step connecting the theoretical findings to practice. Given the theoretical contribution, I do not expect an in-depth empirical analysis with deep networks; however, I would to see numerical results demonstrating the derived theory under the DSD model. It is fine to constrain to single hidden layer networks as is done in the theory.

** Minor point, but in notation it is stated that vectors are indexed at 1, but on page 4, $\mu_i[(i-1)d:id]$ is indexed at 0.

**Questions:**

Please provide numerical experiments with synthetic data under the DSD model and one-layer CNNs, LCNs, and FCNs, and examine the empirical sample complexity of this model as you vary the quantities of interest ($\sigma$, $k$, $d$).

---

> ### Author Response · Authors · 2023-11-21
>
> We genuinely thank the reviewer for their positive evaluation and feel encouraged by their recognition of our work's significance, originality, and clarity. We now address their comments below.
>
> **Comment 1: Can experiments be added that validate the theory?**
>
> Answer:
> We agree with the reviewer that empirical validation of our bounds can help us connect them to practice. We have added detailed experimental protocols and results in the supplementary material, and we plan to incorporate them more fully in the next version of our manuscript.
>
> **Comment 2: Much of the theoretical development is relegated to the appendix.**
>
> Answer:
> We agree with the reviewer's concern about the placement of our theoretical development. In response, we will include a more comprehensive proof outline in our paper's forthcoming revision. Additionally, we have provided a concise outline for the proof of Theorem 6.1 in an [official comment](https://openreview.net/forum?id=AfnsTnYphT&noteId=zesOXLpzPT), which maybe of interest to the reviewer.
>
> **Comment 3: Notational inconsistency in vectors indexing on page 4.**
>
> Answer:
> We are thankful to the reviewer for pointing out this notational inconsistency, and we will rectify it in our revised submission.
>
> We sincerely hope that our responses have addressed the concerns raised by the reviewer.

---

### Author Response · Authors · 2023-11-21
**Addition of Experimental Results in Supplementary Material**

Respected Reviewers,


We would like to express our sincere gratitude for the valuable feedback and constructive suggestions we received during the review process. In line with recommendations, we have included our experimental protocols and results as the supplementary material. We hope this will address some of the key questions raised by the reviewers.


Best Regards,
Authors

---

### Meta-Review · Area_Chair_YLhp · 2023-12-10

**Metareview:**

The paper studies the role of architectural elements such as locality and convolution in reducing the sample complexity for problems exhibiting locality and translation equivariance. It introduces a model theoretical problem, in which observations consist of k patches, k-1 of which consist of gaussian noise, and the k-th of which contains a certain (a-priori unknown) signal vector, or its negative, and the goal is to determine this sign. The paper analyzes three shallow neural networks — a fully connected model, a local model which is aware of the patch decomposition, and a convolutional model with weight sharing across patches. It proves sample complexity separations between the CNN, LCN and FCN — namely when trained with gradient descent, the CNN requires k+d samples, while the LCN requires at least kd samples and the FCN requires at least k^2d samples. These bounds are proved using a variant of Fano’s inequality for randomized algorithms, derived in the paper.

The main strength of the paper compared to previous work is (I) the more realistic task formulation, and (II) it provides sample complexity upper and lower bounds for gradient descent, rather than algorithm-independent empirical risk minimizers, or function classes. Like several previous works, the paper proves sample complexity separations by leveraging the rotation equivariance of FCN and the rotation/permutation invariance of CNN/LCN. The analysis involves a number of technical novelties, which help in proving lower bounds for gradient descent and coping with the fact that the input distribution is not rotationally invariant.

 The main limitations reside in the somewhat idealized nature of the model problem studied here: there is a gap between the gaussian noise model studied here and visual clutter, and the signal model is highly idealized. Nevertheless, the paper contributes in a significant way to the analytical understanding of the benefits of image-specific inductive biases.

**Justification For Why Not Higher Score:**

The paper makes a significant contribution to the study of inductive biases for vision tasks, establishing rigorous separations between convolutional networks, a class of local networks, and fully connected networks. The only concern is the somewhat idealized nature of the task and networks studied here.

**Justification For Why Not Lower Score:**

This is a strong paper, which helps to make rigorous the sample complexity advantages of image-specific inductive biases such as locality and translation invariance.

---

### Decision · Program_Chairs · 2024-01-16

Accept (spotlight)